

# Observed snow depth trends in the European Alps: 1971 to 2019

Michael Matiu[1], Alice Crespi[1], Giacomo Bertoldi[2], Carlo Maria Carmagnola[3], Christoph Marty[4], Samuel Morin[3], Wolfgang Schöner[5], Daniele Cat Berro[6], Gabriele Chiogna[7,8], Ludovica De Gregorio[1], Sven Kotlarski[9], Bruno Majone[10], Gernot Resch[5], Silvia Terzago[11], Mauro Valt[12], Walter Beozzo[13], Paola Cianfarra[14], Isabelle Gouttevin[3], Giorgia Marcolini[8], Claudia Notarnicola[1], Marcello Petitta[1,15], Simon C. Scherrer[9], Ulrich Strasser[8], Michael Winkler[16], Marc Zebisch[1], Andrea Cicogna[17], Roberto Cremonini[18], Andrea Debernardi[19], Mattia Faletto[18], Mauro Gaddo[13], Lorenzo Giovannini[10], Luca Mercalli[6], Jean-Michel Soubeyroux[20], Andrea Sušnik[21], Alberto Trenti[13], Stefano Urbani[22], and Viktor Weilguni[23]

[1]Institute for Earth Observation, Eurac Research, Bolzano, 39100, Italy
[2]Institute for Alpine Environment, Eurac Research, Bolzano, 39100, Italy
[3]Univ. Grenoble Alpes, Université de Toulouse, Météo-France, CNRS, CNRM,
Centre d'Etudes de la Neige, Grenoble, 38000, France TS1
[4]WSL Institute for Snow and Avalanche Research SLF, Davos, 7260, Switzerland
[5]Department of Geography and Regional Sciences, University of Graz, Graz, 8010, Austria
[6]Società Meteorologica Italiana, Moncalieri, 10024, Italy
[7]Chair of Hydrology and River Basin Management, Technical University Munich, Munich, 80333, Germany
[8]Department of Geography, University of Innsbruck, Innsbruck, 6020, Austria
[9]Federal Office of Meteorology and Climatology MeteoSwiss, Zurich-Airport, 8058, Switzerland
[10]Department of Civil, Environmental and Mechanical Engineering, University of Trento, Trento, 38123, Italy
[11]Institute of Atmospheric Sciences and Climate, National Research Council, (CNR-ISAC), Turin, 10133, Italy
[12]Centro Valanghe di Arabba, Arabba, 32020, Italy
[13]Meteotrentino, Provincia Autonoma di Trento, Trento, 38122, Italy
[14]Dipartimento di Scienze della Terra, dell'Ambiente e della Vita – DISTAV,
Università degli Studi di Genova, Genova, 16132, Italy
[15]SSPT-MET-CLIM, ENEA, Rome, 00123, Italy
[16]ZAMG, Innsbruck, 6020, Austria
[17]ARPA Friuli Venezia Giulia, Palmanova, 33057, Italy
[18]ARPA Piemonte, Torino, 10135, Italy
[19]Assetto idrogeologico dei bacini montani, Region Valle d'Aosta, Aosta, 11100 Italy/ TS2 Fondazione Montagna sicura,
Courmayeur, 11013, Italy
[20]Météo-France, Direction de la Climatologie et des Services Climatiques, Toulouse, 31057, France
[21]Meteorology Office, Slovenian Environment Agency, Ljubljana, 1000, Slovenia
[22]Centro Nivometeorologico, ARPA Lombardia, Bormio, 23032, Italy
[23]Abteilung I/3 – Wasserhaushalt (HZB), BMLRT, Vienna, 1010, Austria

**Correspondence:** Michael Matiu (michael.matiu@eurac.edu)

Received: 3 October 2020 – Discussion started: 12 October 2020
Revised: 30 January 2021 – Accepted: 31 January 2021 – Published:

**Abstract.** The ⟨CE1⟩European Alps stretch over a range of climate zones which affect the spatial distribution of snow. Previous analyses of station observations of snow were confined to regional analyses. Here, we present an Alpine-wide analysis of snow depth from six Alpine countries – Austria, France, Germany, Italy, Slovenia, and Switzerland – including altogether more than 2000 stations of which more than 800 were used for the trend assessment. Using a principal component analysis and $k$-means clustering, we identified five main modes of variability and five regions which match the climatic forcing zones: north and high Alpine, north-east, north-west, south-east, and south and high Alpine. Linear trends of monthly mean snow depth between 1971 and 2019 showed decreases in snow depth for most stations from November to May. The average trend among all stations for seasonal (November to May) mean snow depth was $-8.4\,\%\,\mathrm{decade}^{-1}$, for seasonal maximum snow depth $-5.6\,\%\,\mathrm{decade}^{-1}$, and for seasonal snow cover duration $-5.6\,\%\,\mathrm{decade}^{-1}$. Stronger and more significant trends were observed for periods and elevations where the transition from snow to snow-free occurs, which is consistent with an enhanced albedo feedback. Additionally, regional trends differed substantially at the same elevation, which challenges the notion of generalizing results from one region to another or to the whole Alps. This study presents an analysis of station snow depth series with the most comprehensive spatial coverage in the European Alps to date.

## 1 Introduction

In the European Alps, snow is pervasive throughout nature and human society. Snow is a major driver of Alpine hydrology by storing water during the winter season which gets released in spring and summer and which is used for water supply, agriculture, and hydropower generation. Water stored in the snow cover also feeds alpine aquifers through the network of fault and fracture systems. Ecologically, the mountain flora and fauna depend on the timing and abundance of snow cover (Esposito et al., 2016; Keller et al., 2005; Lencioni et al., 2011). Snow is tightly linked to human culture in the European Alps and has brought economic wealth to previously remote regions through tourism (Beniston, 2012a; Steiger and Stötter, 2013). Since snow cover depends on temperature and precipitation, ongoing climate change in the Alps and especially rising temperatures and changing precipitation patterns affect the abundance of snow (Beniston and Stoffel, 2014; Gobiet et al., 2014; Steger et al., 2013). Snow cover extent decreased globally, while for snow mass, some regions experienced increases (Pulliainen et al., 2020). Decreases are expected in the future, especially at low elevations with more uncertain trends in observations and future projections at higher elevation (Beniston et al., 2018; Hock et al., 2019; IPCC, 2019).

Observations are needed to assess ongoing changes in snow cover. The most widespread snow cover measurements are snow depth (HS), depth of snowfall (HN, also denoted as fresh snow or snowfall), snow water equivalent (SWE), snow cover area (SCA), and snow cover duration (SCD). Snow depth and depth of snowfall measurements have been scientifically documented in the European Alps since the late 18th century (Leporati and Mercalli, 1994). Such measurements indicating the height of the snow cover relative to the ground (snow depth) or a reference surface, usually a board (depth of snowfall), are performed each morning by observers and only require a graduated stake or rod and a metre stick. While automatic sensors have been developed in recent decades, most European weather and hydrological services continue with manual observations. Although there is a trend towards automatization, missing standards on the processing of the data (even at national level) impede their uptake (Haberkorn, 2019; Nitu et al., 2018). The main limitation of snow depth and depth of snowfall measurements is that their number decreases sharply with elevation with few stations available above 3000 m in the European Alps. SWE is the mass of snow per unit surface area, which corresponds to the amount of water stored in the snow cover and thus is a key hydrological variable. However, its measurement is far more complicated and available with lower temporal frequency than snow depth, and thus not as widely observed. SCA and SCD identify the spatial extent and temporal duration of snow on the ground. SCD can be inferred from snow depth measurements using a threshold or more recently from satellite observations which also allow SCA retrieval at different spatial scales from tens of metres to several kilometres. The main benefit of satellite observations is that they cover the whole elevational gradient and are also available in more data-scarce regions. Satellite observations can identify SCA and SCD at high spatial resolutions (1 to 5 km for decadal length time periods) and less accurately SWE at coarser resolution ($\sim$ 25 km) (Schwaizer et al., 2020). However, they typically cover a relatively short time period and are hampered by cloud cover and rugged topography (Bormann et al., 2018), and the satellite orbit might not provide a worldwide cover. An application of global satellite imagery for 2000–2018 has shown an SCD decline for 78 % of global mountain areas and only a few regions with increasing SCD (Notarnicola, 2020), although the short time span of 19 years is a limiting factor in interpreting these trends.

The European Alps are densely populated and have a long history of manual snow depth and depth of snowfall observations which makes them ideal to study long-term trends over a large spatial domain with complex topography and strong climate gradients. Not surprisingly, much literature on the topic exists (see Table B1 in Appendix B for an overview). However, most studies are limited in their spatial extent to regions or nations and restricted by a lack of data sharing, harmonized data portals, and joint projects or initiatives fostering such analyses (Beniston et al., 2018).

The most relevant findings of the latest literature on snow cover trends (Table B1) can be summarized as follows. Snow variables exhibited a strong temporal and spatial variability (e.g. Beniston, 2012b; Schöner et al., 2019). Long-term analyses identified periods of high snow cover in the 1940s/50s, as well as in the 1960s/70s, followed by absolute minima in the 1980s and early 1990s with some recovery afterwards but not to the pre-1980s values (Marty, 2008; Micheletti, 2008; Scherrer et al., 2013; Schöner et al., 2009; Valt and Cianfarra, 2010). Trends were strongly related to elevation (Laternser and Schneebeli, 2003; Marcolini et al., 2017b; Valt et al., 2008) and were mostly negative at low elevations (Bach et al., 2018), while higher elevations showed no change or even increases (Marty et al., 2017; Terzago et al., 2010). Snow melt was identified as the main contribution to the decreasing trends (Klein et al., 2016), which explains the pronounced trends at low elevations and in spring (Marty et al., 2017). Finally, after accounting for elevation, regional differences between trends were observed (Beniston, 2012b; Laternser and Schneebeli, 2003; Schöner et al., 2019; Terzago et al., 2013).

Quantitatively synthesizing all these studies into a common Alpine view is challenging, and thus the provision of quality-ensured information on snow cover climatology and trends at larger extents, such as the whole Alpine mountain range, is hampered (Hock et al., 2019). The challenge starts from the different definitions of the studied seasons, which range from December–February to October–May, and thus sometimes include the start, middle, and end of the season. Difficulties also arise in the selection of existing snow variables and indices, such as mean snow depth, maximum snow depth, snow days (based on thresholds from 1 to 50 cm), 3 d cumulative values, etc. Naturally, the station series are of different lengths, and the studied periods get longer for the more recent studies. Finally, the statistical methods differ from one study to another: linear regressions, Mann–Kendall tests, Sen slopes, moving window approaches (windows ranging from 5 to 20 years), breakpoint analysis, principal component analysis/empirical orthogonal function analysis (PCA/EOF), and more.

To overcome these limitations, we embarked on the effort to collect and analyse an Alpine-wide dataset of snow measurements from stations covering Austria, France, Germany, Italy, Slovenia, and Switzerland. The main aim is to understand how changes in snow cover vary over space and time by applying the same methods to an as homogenous as possible Alpine-wide dataset. This approach avoids sub-regional perspectives, inconsistencies from single data sources and different methods, and influences of artificial boundaries such as national borders. Since we wanted the data collection effort to be of use for the scientific community, we make as much of the data as possible openly accessible (as far as data policies allow us to). The remainder of the paper is structured as follows: Sect. 2 introduces the data and the statistical methods, Sect. 3 presents results and discusses them, and Sect. 4 provides conclusions.

## 2 Data and methods

### 2.1 Study region

The European Alps extend with their arc-shaped structure over more than 1000 km from the French and Italian Mediterranean coasts to the lowlands east of Vienna, covering southeastern France, Switzerland, northern Italy, southern Germany, Austria, and Slovenia (see Fig. 1a). The Alpine region is characterized by a very complex orography with large elevation gradients and deep valleys of different orientations intersecting the ridge and shaping numerous mountain massifs.

Regarding their climatic setting, the European Alps are located in a transitional area influenced by the intersection of three main climates: the zone impacted by the Atlantic Ocean with moderate wet climate, the zone linked to the Mediterranean Sea characterized by dry summers and wet and mild winters, and the zone characterized by European continental climate with dry and cold winters and warm summers. Elevational effects and very small-scale climatic features originating from the complex Alpine topography are superimposed on this large-scale climatic setting (Auer et al., 2005; Isotta et al., 2014).

The interaction of the three climate forcing zones, together with the topography of the Alps, results in climatic gradients along the north–south and west–east directions. The intersection of these two gradients can be characterized by four main climate regions, as shown by Auer et al. (2007). The first and sharpest climatic border is along the central main ridge separating the temperate westerly from the Mediterranean subtropical climate. The second climatic border separates the western oceanic from the eastern continental influences.

### 2.2 Data sources

The acquisition of snow observation data was performed by using open data portals and by directly contacting data providers (see Table 1 for an overview). For Austria, the Austrian Hydrographical Service (HZB, Hydrographisches Zentralbüro) offers free downloads of their data for recent decades, and additional historical data at the seasonal scale were kindly provided by the HZB. For France, data were kindly provided by the national weather service Météo-France. This includes data collected as part of the collaborative network (réseau nivo-météorologique) between Météo-France and mountain stakeholders (in particular Domaines Skiables de France, Association Nationale des Maires des Stations de Montagne, and l'Association Nationale des Directeurs de Pistes et de la Sécurité de Stations de Sports d'Hiver). For Germany, data were downloaded from the national weather service's (DWD, Deutscher Wetterdienst)

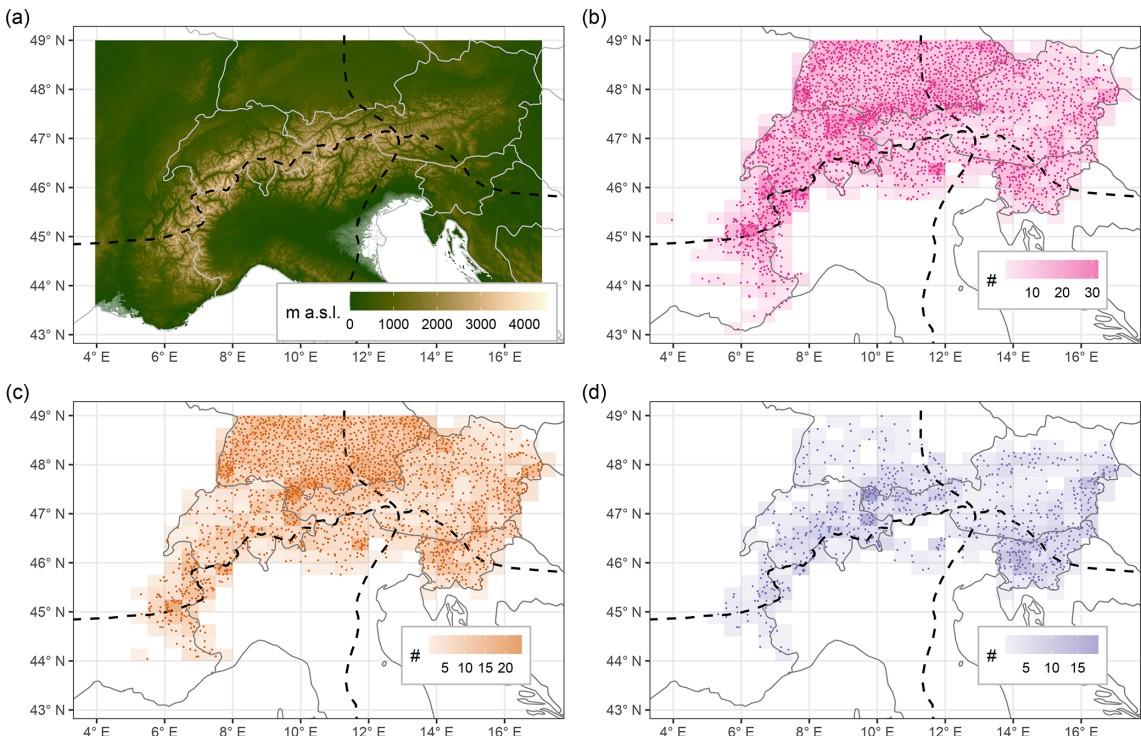

**Figure 1.** Topography of the European Alps **(a)** and overview of station locations **(b–d)**. Panel **(a)** shows the SRTM30 DEM (Shuttle Radar Topography Mission digital elevation model) with ∼ 1 km resolution. Panel **(b)** shows the location of snow depth measurement locations that were available (provided). Panel **(c)** shows the locations of stations used in the regionalization analysis. Panel **(d)** shows the stations used for the long-term trend analysis. The station density for a 0.5° × 0.25° grid is shown underneath the points in **(b–d)**. The main climatic divides from Auer et al. (2007) are shown as dashed lines in **(a–d)**. See also Appendix A and Sects. 2.4 and 2.5 for selection criteria. TS3

**Table 1.** Overview of the number of stations with daily data provided by the different data sources. The data source consists of a country abbreviation, followed by the data source. Country abbreviations are AT for Austria, CH for Switzerland, DE for Germany, FR for France, IT for Italy, and SI for Slovenia. For source abbreviations, please see Sect. 2.2. Station numbers are shown for depth of snowfall (HN) and snow depth (HS) time series. See Appendix A and Sects. 2.4 and 2.5 for more details on station selection procedures associated with the different types of analyses. HN was not analysed but was used for checking HS.

| Data source | HN | HS | HS used (regionalization) | HS used (trend analysis) |
|---|---|---|---|---|
| AT_HZB | 653 | 652 | 588 | 335 |
| CH_METEOSWISS | 505 | 501 | 142 | 79 |
| CH_SLF | 96 | 96 | 94 | 84 |
| DE_DWD | 956 | 964 | 830 | 104 |
| FR_METEOFRANCE | 239 | 286 | 145 | 45 |
| IT_BZ | 60 | 64 | 48 | 0 |
| IT_FVG | 30 | 30 | 18 | 8 |
| IT_LOMBARDIA | 11 | 11 | 11 | 0 |
| IT_PIEMONTE | 34 | 34 | 24 | 15 |
| IT_SMI | 6 | 8 | 8 | 7 |
| IT_TN | 52 | 52 | 29 | 8 |
| IT_TN_TUM | 0 | 5 | 1 | 0 |
| IT_VDA_AIBM | 57 | 57 | 17 | 5 |
| IT_VDA_CF | 0 | 17 | 11 | 3 |
| IT_VENETO | 10 | 11 | 11 | 9 |
| SI_ARSO | 130 | 172 | 172 | 152 |
| Total sum | 2839 | 2960 | 2149 | 854 |

open data portal using the R package rdwd. For Germany, only stations below 49° N were downloaded. For Italy, the data were kindly provided by many regional authorities:

- for the province of Bolzano, from the hydrographical office of Bolzano (BZ);

- for Friuli Venezia Giulia (FVG), from the regional weather observatory (OSMER, Osservatorio meteorologico regionale), which is part of the ARPA (Agenzia regionale per la protezione dell'ambiente) FVG and from where the data were collected and cleaned by the Servizio foreste e corpo forestale struttura stabile centrale per l'attività di prevenzione del rischio da valanga;

- for Lombardy, from the ARPA Lombardia;

- for Piedmont, from the ARPA Piemonte;

- for the province of Trento, from Meteotrentino (TN) with some additional long-term series previously analysed (TN_TUM; Marcolini et al., 2017a);

- for the Aosta Valley (VDA), from the civil protection office (CF: Centro funzionale, Regione Valle d'Aosta) and from the avalanche office (AIBM: Assetto idrogeologico dei bacini montani, Regione Valle d'Aosta);

- for Veneto, from the avalanche office (Centro valanghe di Arabba), which is part of the ARPA Veneto;

- and, finally, additional data for Piedmont and Aosta Valley from the Italian meteorological society (SMI, Società Meteorologica Italiana).

For Slovenia, data were kindly provided by the Slovenian Environmental Agency (ARSO, Agencija Republike Slovenije za okolje). For Switzerland, data were downloaded from the IDAWEB portal of the national weather service MeteoSwiss, and additional data were kindly provided by the WSL Institute for Snow and Avalanche Research SLF. This dataset comprises the entire geographical range of the European Alps, yet we are aware of the existence of additional datasets (such as in the private sector or public but not yet digitized) which unfortunately were not included in this analysis and whose inclusion would be beneficial for even more robust results.

The data consist of daily measurements of snow depth (HS) and depth of snowfall (HN). The largest part of the data is manual measurements. Some automatic measurements were included in the dataset provided for France. For a few sites in the Aosta Valley in Italy, manual series were merged with automatic series. This was done in order to extend up to the present some records that were dismissed at the beginning of the last decade, and this was performed in close communication with the operating office. While the observers follow slightly different guidelines in each country

or network, the observation modalities are remarkably similar, thus allowing a combination of the different sources. For more detailed information on the measuring modalities, we refer to the European Snow Booklet (Haberkorn, 2019). Values of HS and HN were rounded to full centimetres. The further processing, quality checking, and gap filling are described in Appendix A. For all the following statistical analyses, the quality was checked, and gap-filled data were used.

The fraction of stations used from the MeteoSwiss data is very low compared to the other networks. The MeteoSwiss data contain a large number of stations from the manual precipitation network which is not dedicated to snow. Many stations contain an important data gap for the 1981–1997 period that rendered a large fraction of the stations unusable for this study.

The homogenization of series, which is the removal of non-climatic parts in the time series, such as, for example, those caused by instrumentation changes or station relocations, is a standard practice in long-term temperature and precipitation records (Auer et al., 2007). Applying the same tools to snow depth is not straightforward. There is an ongoing discussion on the appropriate homogeneity tests and suitable observation frequency, such as daily, monthly, or seasonally (Marcolini et al., 2017a, 2019; Schöner et al., 2019). An analysis of a dataset with parallel snow measurements indicates that snow cover duration and maximum snow depth are amongst the indicators least affected by inhomogeneities (Buchmann et al., 2021). Current research has tried to extend existing approaches with new innovations (Resch et al., 2020). Homogenization could improve the robustness of estimated trends, and be especially useful for areas with sparse observations, such as for elevations above 2000 m. Given the large extent of our dataset, it was not possible to apply a common homogenization framework for our study, and we leave this for future studies.

## 2.3 Data overview

The locations of the stations are shown in Fig. 1b–d, the availability of stations in time in Fig. 2a, and the elevational distribution in absolute terms in Fig. 2b and in relative terms in Fig. 2c. The stations cover the whole Alpine arc, but they are distributed with different station densities arising from the different national and regional networks. As expected, most stations were found at lower elevations, the maximum number was at ∼ 500 m, and the number sharply declined at higher elevations. Above 2000 m, the number was low, and no stations above 3200 m were available for this study. The longest series dates back to the late 19th century for HS (Passau_Maierhof CE2 in Germany, starting in 1879). The total number of available HS stations depended on the availability of digitized data. It slowly started increasing around ∼ 1900 with significant jumps in the 1960s and 1970s when the French, Slovenian, and Austrian series started and in the 1980s when Germany had a large network increase. The

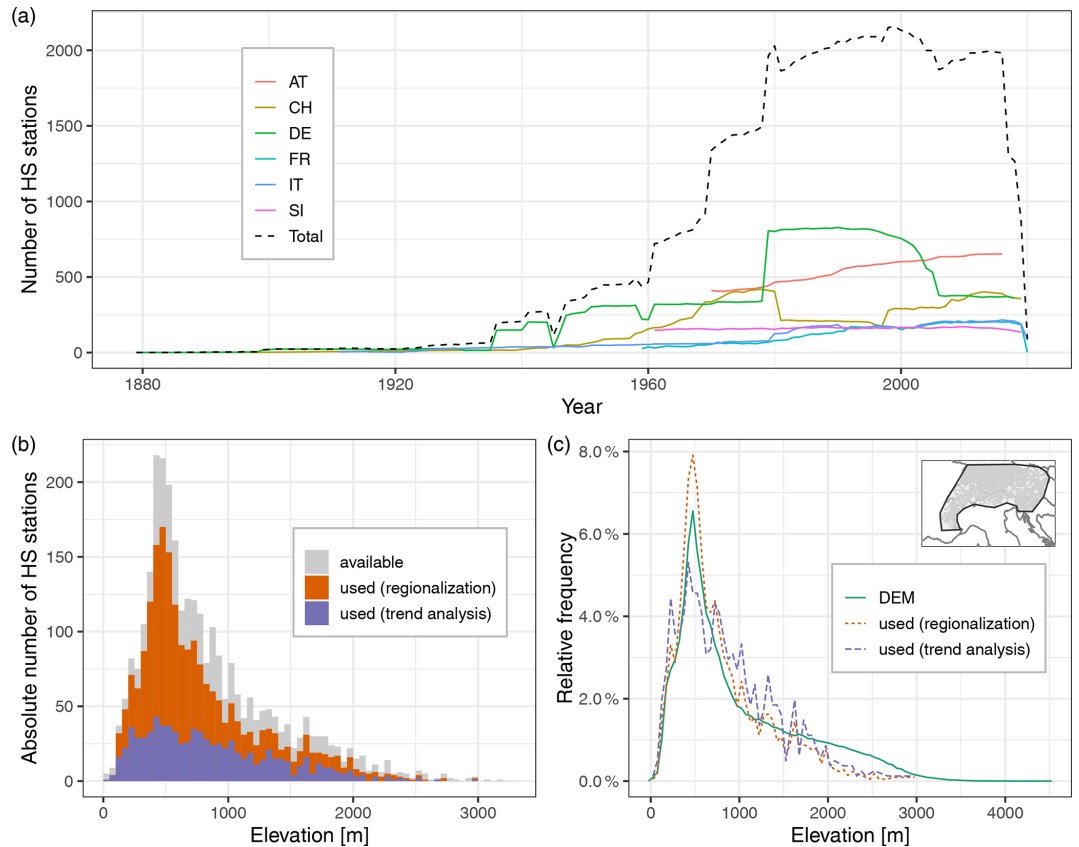

**Figure 2.** Overview of temporal data availability and station elevation. **(a)** The number of stations with daily data (before gap filling) is shown per year and country, as well as a total sum for the whole Alpine region. Stations are included in the count if they have at least one non-missing observation in the respective calendar year. This simple threshold was chosen because the aim of this figure is to show the availability and network abundance. Country abbreviations are like in Table 1. **(b)** The elevational distribution of snow depth (HS) stations in absolute numbers. For the histogram, 50 m bins were used. **(c)** Comparison of the relative elevational distribution of the station locations vs. a digital elevation model (DEM). The distribution of the stations is shown in relative terms using the same bin width (50 m) as in the histogram in **(b)** but normalized to show the relative frequency instead of absolute numbers and displayed as lines instead of bars. This is compared to the elevation for the whole area spanned by the stations (see polygon in inset map; area was outlined manually along the stations), which is extracted from the SRTM30 DEM (Shuttle Radar Topography Mission; ∼ 1 km resolution).

highest number of stations was available after the 1980s with approximately 2000 stations. The total number of stations dropped significantly after 2017 because the data for Austria were only available until 2016 due to the delays caused by the data provider performing quality checks. Moreover, the data collection was performed between 2019 and 2020, thus some sources ended in between. We used two different periods for the two analyses that we performed. For the regionalization, we aimed to have the largest possible spatial extent and density of the stations; so the period 1981 to 2010 was chosen because it is the period with the highest number of stations. For the trend analysis, we aimed to have trends as long as possible that sample the whole region; so the period 1971 to 2019 was chosen because it offered the best tradeoff between station coverage and period length.

## 2.4 Regionalization

An empirical orthogonal function (EOF) analysis, also called principal component analysis (PCA), was conducted to determine the common modes of spatial variability. PCAs are widely employed in climatological studies to evaluate spatial modes of variability (Storch and Zwiers, 1999). They have been employed for meteorological records in the European Alps (Auer et al., 2007) and also for snow variables (López-Moreno et al., 2020; Scherrer and Appenzeller, 2006; Schöner et al., 2019; Valt and Cianfarra, 2010). For the PCA, we used daily quality-checked and gap-filled data. However, the gap filling was only employed when enough confidence in the filled value could be expected (see Appendix A for a detailed description), so some of the series still had gaps. Because the aim of this regionalization was to have a large spatial coverage, we did not want to exclude series with only

a few missing values. Consequently, we used a modification of the PCA algorithm that allows for the use of data with gaps to estimate the principal components (Taylor et al., 2013).

The PCA was applied to the daily data from December to April for the hydrological years 1981 to 2010. The period was chosen because it is long enough to provide a climatological reference (30 years), and it is the period that has the largest number of stations available. A hydrological year is defined here as starting in October, and it is designated as the calendar year of the ending month (e.g. December 1998 to April 1999 belong to the hydrological year 1999). Only those stations were selected that had at least 70 % of daily data available in this period. Each series was scaled to zero mean and unit variance before applying the PCA.

In order to identify spatially homogeneous regions within the Alpine domain, we performed a $k$-means clustering on the estimated PCA matrix. We tested configurations with two to eight clusters CE3 with the PCA matrix and with two to eight principal components (PCs) as input. We also applied $k$-means clustering directly on scaled daily observations of snow depth for comparison. To identify the best number of clusters, we used the "elbow method", average silhouette coefficients, and visual interpretation. For the elbow method, the fraction of explained variance is plotted against the number of clusters, and the elbow of this curve is the point where the increase in explained variance becomes marginal. This is a semi-objective method because an elbow cannot always be clearly identified. The silhouette is a measure of how well an observation fits into its own cluster vs. the others. For an observation $i$ in cluster $C_i$, the silhouette coefficient is $1 - a(i)/b(i)$ if $a(i) < b(i)$, $b(i)/a(i) - 1$ if $a(i) > b(i)$ and 0 if $a(i) = b(i)$, where $a(i)$ is the mean distance between $i$ and all other points in the same cluster, and $b(i)$ is the smallest mean distance of observation $i$ to all other clusters. Specifically, $a(i) = \frac{1}{|C_i| - 1} \sum_{j \in C_i, i \neq j} d(i, j)$ and $b(i) = \min_{k \neq i} \frac{1}{|C_k|} \sum_{j \in C_k} d(i, j)$, where $d(ij)$ is the Euclidean distance between observations.

The optimal number of clusters varied between two and five depending on the input (observations or PCA matrix) and depending on the metric (elbow in variance explained or average silhouette coefficients). Additional PCs only explained less than 2.6 % of the variance. After looking at the clustering results on maps (see Fig. S1 in the Supplement), all two to five clusters are meaningful. They simply highlight different aspects of the snow depth spatial variability, such as the gradients along elevation, north–south, and west–east. Finally, five clusters based on the PCA matrix were chosen because they provide the best trade-off between the semi-objective metrics and the patterns expected from the climatic drivers.

## 2.5 Trend analysis

For the trend analysis, monthly and seasonal indices were used which are indicative for different aspects and times of the snow season: monthly mean HS for November to May (NDJFMAM), mean winter HS (December to February, DJF), mean spring HS (March to May, MAM), mean seasonal HS (November to May), maximum HS from November to May (maxHS), early season snow cover duration (SCD, November to February), late season SCD (March to May), and full season SCD (November to May). SCD was the number of days with HS above 1 cm (Brown and Petkova, 2007). Indices were calculated from the quality-checked and gap-filled daily snow depth observations if more than 90 % of the daily values in the respective period were available. Trends of all indices were calculated for the period 1971 to 2019 for stations with complete data in the period. For the monthly mean HS analysis only, April and May series displaying mean HS less than 1 cm in all years were discarded because these are insignificant snow amounts which divert attention from the other sites; series of the other months at the site were still included. The number of series available for each snow variable differs; the largest number of series is available for the monthly mean HS, less for the half-seasonal (3 to 4 months) indices, and the fewest for the full-season indices.

Trend analysis was performed using two generalized least squares (GLS) regression. GLS was used because it allows for changes in the variance to be accounted for (Pinheiro and Bates, 2000). This was employed because the monthly snow depth series exhibited a change in the interannual variability, especially at the end of the season when monthly snow depths approached zero. The regression formula was $y_t = \beta_0 + \beta_1 t + \epsilon_t$, where $y_t$ is the value of the respective snow variable in year $t$ (centred such that year 1971 becomes year 0), $\beta_0$ and $\beta_1$ are the estimated regression coefficients, and $\epsilon_t$ is the normally distributed errors with mean zero. GLS allows for the variance to depend on the year $t$ with $\mathrm{Var}(\epsilon_t) = \sigma^2 \cdot \exp(2 \cdot \gamma \cdot t)$, where $\gamma$ is a coefficient to be estimated in the inference procedure that indicates the change in variance associated to $t$. The GLS regressions for monthly mean HS showed a significantly improved goodness of fit ($p < 0.05$, likelihood ratio test) for 40 % of all cases and, specifically for November, April, and May, even for more than 60 % when compared to ordinary least squares (OLS) that assumes a constant error variance. The significance of trends was assessed using a 95 % confidence level. For the fraction of the variance explained by the trend, we used the R squared statistic. To determine the magnitude of the interannual variability after accounting for the trend, we used the standard deviation (SD) CE4 of the model residuals.

An alternative for dealing with such heteroscedastic data is to use the robust nonparametric Theil–Sen trend estimator with the Mann–Kendall test for significance assessment. We systematically evaluated the differences in the estimated trend magnitudes and trend significance of the Theil–Sen approach vs. the GLS model and found only negligible differences (Fig. S13 and Table S10 in the Supplement); the mean difference between trend estimates was 0.02 cm per decade, the correlation between trend estimates was 0.96, and the

agreement of significance based on a *p* value threshold of 0.05 was 86 %.

The SCD variables are bounded counts which can pose problems to the assumption of standard linear regression with normally distributed errors. This was only problematic for very low- and very high-elevation sites which display many SCD values at the minimum or maximum. For MAM, this concerns series below 500 m and above 2000 m, while for November–February (NDJF) and November–May (NDJF-MAM), this is problematic below 250 m and above 2500 m. Instead, for such count data, a probability distribution such as negative binomial would be more appropriate (Venables and Ripley, 2002). Compared to the Poisson distribution, the negative binomial family accounts for overdispersion. We evaluated the differences in trend estimates and trend significance between the negative binomial linear model and the GLS model. Since the negative binomial linear model gives relative estimates of trends, these were transformed to absolute decadal trends for comparison. Again, differences were negligible on average (Fig. S13 and Table S10). Consequently, we applied the GLS model for all snow variables.

## 2.6 Air temperature and precipitation data

In order to study the relationship of snow depth with temperature and precipitation, we extracted temperature and precipitation series for each station from available gridded products. While gridded datasets clearly have some shortcomings, e.g. comparisons with point observations need a cautious interpretation (Salzmann and Mearns, 2011), their strength is the spatial and temporal coverage.

Two types of products were considered. The first is a reanalysis, and the second is an observation-based spatial analysis. For the reanalysis, we used temperature and precipitation from the MESCAN-SURFEX dataset (Bazile et al., 2017) which was produced during the UERRA (Uncertainties in Ensembles of Regional Reanalyses) project and which is available via the Copernicus Data Store (CDS). It covers the period from January 1961 to July 2019 on a 5.5 km grid. Precipitation is available as total daily sum and temperature at 6 h intervals (00:00, 06:00, 12:00, 18:00 UTC). For the observation-based data, we chose E-OBS v20.0e for mean daily temperature (Cornes et al., 2018) and the Alpine precipitation grid dataset (EURO4M-APGD) for total daily precipitation (Isotta et al., 2014; Isotta and Frei, 2013). E-OBS v20.0e spans the period from January 1950 to July 2019 on a 0.1° grid. APGD covers the period from January 1971 to December 2008 on a 5 km grid. It should be noted that the observation-based precipitation grids do not account for undercatch, which can lead to uncertainties at high elevations and in winter (Prein and Gobiet, 2017).

In order to assign grid cells to stations for temperature and precipitation, we selected those grid cells which contain the stations. Consequently, some nearby stations could have the same series of temperature and precipitation. The daily (or 6 h for temperature from MESCAN-SURFEX) series were aggregated to monthly means for temperature and monthly sums for precipitation.

The gridded products have a reference orography that, in complex mountain terrain, can differ significantly from the elevation of the point observation, thus, for example, introducing biases in temperature. Hence, temperatures were adjusted using a constant lapse rate of $6.5\,°\mathrm{C\,km^{-1}}$.

Monthly temperature and precipitation can be considered largely independent from one month to the next, while snow cover is a cumulative process across the snow season. Because of this, seasonal comparisons were performed with average seasonal temperature and precipitation for winter (December to February), spring (March to May), and the whole snow season (November to May). The time period 1981 to 2010 was used, which had the densest station coverage. Climatological averages were computed for all seasons using the quality-checked and gap-filled snow depth data. Since EURO4M-APGD ends in 2008, the time period 1981 to 2008 was used for the observation-based products. The paper contains results from the comparison with the reanalysis product (MESCAN-SURFEX), and the results from the observation-based products are shown in the Supplement as a sensitivity analysis.

## 3 Results and discussion

### 3.1 Regionalization of daily snow depths 1981 to 2010

The PCA of daily snow depth series yielded five main modes of spatial variability which explained in total 84 % of the variance in the period December to April from 1981 to 2010 (Fig. 3). The first PC explained 54.3 % of the variance and distinguished between high- to middle- and low-elevation stations (approximate threshold 500–1000 m; Fig. 4). It explained the variability in snow depth for stations above 1000 m and was probably also partly linked to the permanence (or permanent absence) of snow cover, which is also why some low-elevation sites presented similar loading to the high sites (a PC loading can be considered the correlation of the original series with the principal component). The second PC explained 11.9 % of the variance and was also linked to elevation, but it captured the variability below 1000–1500 m (Fig. 4). Consequently, PC1 and PC2 together captured the variability across the whole elevation range. The third PC explained 8.1 % of the variance and separated the stations into north and south of the main ridge. The fourth PC explained 6.0 % of the variance and separated the stations into east from west. The fifth PC explained 3.7 % of the variance and separated the south-eastern and north-western stations from the rest.

Some gradients in the PC loading map (Fig. 3) could give the impression that data artefacts between the different data providers exist, such as at the Austrian–German border in

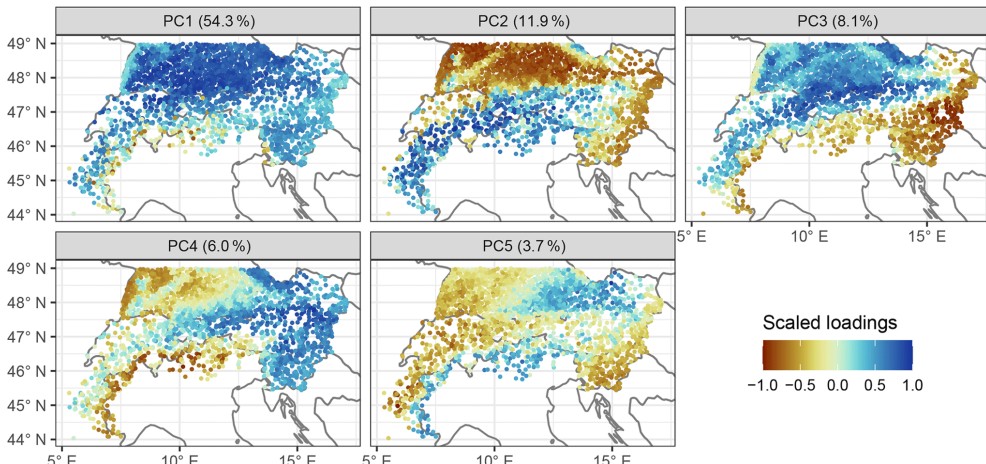

**Figure 3.** Main modes of variability in daily snow depth series. The plots show scaled loadings for the first five principal components (PCs), which can be considered the correlation of the original series with the respective PC. The title in each panel contains the amount of the variance explained by the respective PC. The principal component analysis was applied to daily snow depth data from December to April for the hydrological years 1981 to 2010 for stations that had at least 70 % of available data.

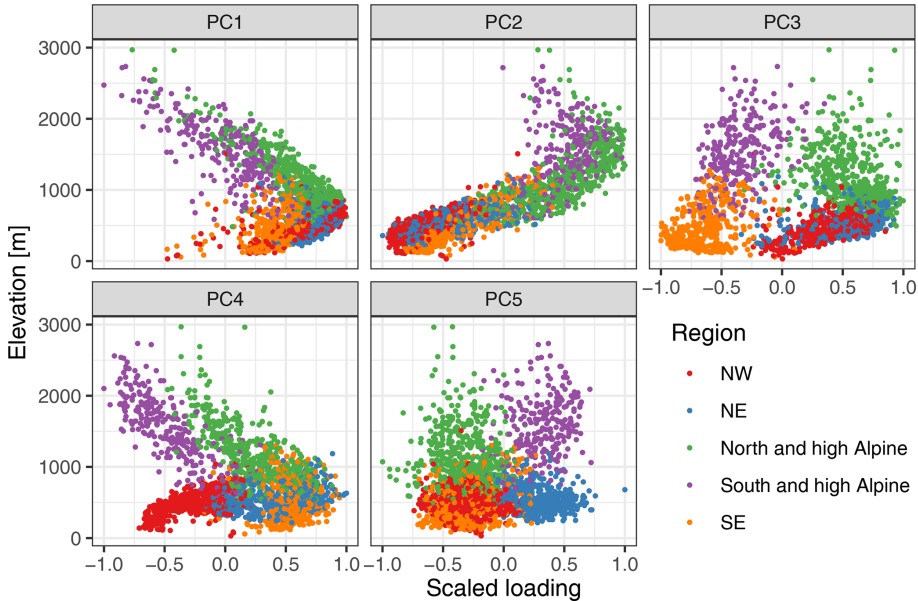

**Figure 4.** Scatterplots of principal component (PC) loading vs. elevation and region. The PC loading can be considered the correlation of the original series with the respective PC. See Fig. 3 for a map of the PC loadings and Fig. 5 for a map of the regions.

PC2 and PC5 or at the French–Italian border in PC3-5. However, this is caused by the fact that the administrative borders in the Alps are tied to topography and thus closely located near elevational borders (Fig. 1a). A version of Fig. 3 subdivided by data provider highlights clearly that the gradients were not associated with the administrative borders (Fig. S2 in the Supplement).

The PCA loadings from the five PCs were used as input for a clustering algorithm (*k* means) which divided the stations into five clusters or regions (Fig. 5). This yielded three regions in the north: north-west (NW) with a median elevation

of 472 m (min–max: 30–1510 m), which contained stations from south-western Germany, north-western Switzerland, a few from France, and a few from eastern Austria; north-east (NE) with a median elevation of 515 m (215–1188 m), which contained stations from south-eastern Germany and northern Austria; and north and high Alpine with a median elevation of 1050 m (482–2970 m), which contained stations mainly located in France, Switzerland, and Austria but also includes the high-elevation sites in Germany, such as in the Black and Bavarian forests. Two regions emerged south of the main ridge: south and high Alpine with a median el-

https://doi.org/10.5194/tc-15-1-2021 The Cryosphere, 15, 1–40, 2021

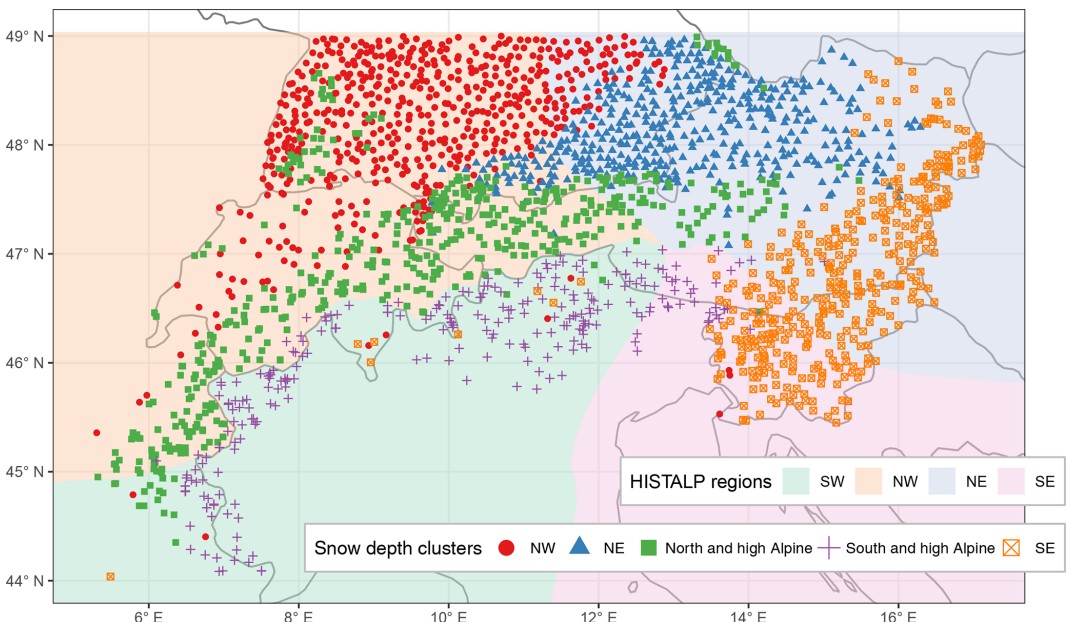

**Figure 5.** Clustering of stations based on daily snow depth data. Map of regions from applying a *k*-means clustering on the first five principal components. Underlaid are the HISTALP coarse-resolution subregions (Auer et al., 2007) which were derived using a semi-automatic principal component analysis of climate variables (temperature, precipitation, air pressure, sunshine, and cloudiness).

evation of 1530 m (588–2735 m), which contained stations from the southern French Alps, almost all of Italy, a few in southern Switzerland, and some in southern Austria and eastern Slovenia; and south-east (SE) with a median elevation of 420 m (55–1300 m), which contained almost all stations from Slovenia and parts of eastern Austria.

Consequently, clusters NW, NE, and SE contained lower-elevation sites, while north and high Alpine and south and high Alpine contained the higher elevations. The spatial coverage of the stations in this study included low-elevation sites for Switzerland, Germany, Austria, and Slovenia but not for France and Italy where the available stations were mostly high-elevation sites. For future analysis, it would be interesting to include more low-elevation sites from France and Italy and see whether a third cluster would emerge (as in the north) because the division into south and high Alpine and SE is surely also caused by the different station elevations.

The results from the clustering were obtained automatically, and no manual post-processing or modification of the cluster assignments was performed. Additionally, the only input into the clustering algorithm was daily snow depth series, and no information on location or elevation was included. Given this absence of location information in the clustering process, the estimated modes of variability and the resulting regions were very homogenous in space. However, in the clustering, some stations seemed off, such as the few north-west stations around Lugano in Switzerland, northern Italy, and on the Adriatic coast in Slovenia, as well as the SE stations in France, Switzerland, and northern Italy. This was not related to the PCA algorithm used that allowed gaps in

data since the results looked almost identical to a standard PCA (see Figs. S3 and S4 in the Supplement), in which the clustering agreed in 98.5 % of the stations, and the same stations seemed mis-clustered. Instead, this might be related to special local climatic conditions affecting snow cover or to the fact that these stations did not have any similar neighbours in the estimated clusters. For example, the five stations in Ticino, located in Switzerland south of the main ridge, are low-elevation stations that had no correspondence in the south and high Alpine cluster which is comprised of middle to high elevations. Thus, the next best clusters were the SE and NW which, however, did not fit well; these sites and all other seemingly mis-clustered stations had low silhouette values (Fig. B1), which is a measure of how well a point matches its cluster compared to the others. Low silhouette values were also found along the borders of the different clusters, especially between NW and NE, which implies a smoother transition between NW and NE compared to the north–south boundary.

The estimated modes of variability in snow are similar to previous estimates on climatic subregions in the Alps, as identified in the HISTALP project (Auer et al., 2007) and which are underlaid in Fig. 5. The HISTALP regions were based on temperature, precipitation, air pressure, sunshine and cloudiness, and the division into north, south, east, and west matches what we found for snow depth. Since the four regions were a compromise between all variables, they do not match perfectly to what we found for snow depth because the individual atmospheric variables exert different controls on surface snow cover. While the north–south boundary is al-

most identical in the central-western part, the eastern part has large mismatches. However, if the single element boundary for precipitation was considered as a main factor (cf. Fig. 8 from Auer et al., 2007), then the agreement with snow depth would be almost perfect. This finding confirms a consistent picture of the Alpine climate in which snow depth is strongly related to precipitation and air temperature patterns.

The amount of the variance explained in the PCA with five PCs (84 %) might seem surprisingly high given that snow cover is hypothesized to have a high spatial and temporal variability. The value is higher than recent estimates for the Swiss Alps, where the first three PCs explained 78 % of the snow cover (Scherrer and Appenzeller, 2006), or for Austria and Switzerland, where the first three PCs explained 70 % of the snow cover (Schöner et al., 2019). However, since we included here more stations and also stations from regions with different climatic influences, such as south of the main ridge, an increase in the amount of the explained variance could be expected.

## 3.2 Snow depth climatology 1981 to 2010 and links to temperature and precipitation

Besides differences in the patterns of daily variability in the snow depth series, the regions also demonstrated different snow depth climatologies (Fig. 6). Looking at average winter (December to February) snow depth from 1981 to 2010, the northern regions had higher snow depths than their southern counterparts. These differences became larger with increasing elevation. While below 750 m no substantial differences were observed, southern stations had ≈ 30 % less snow than northern stations until 1750 m and ≈ 20 % less until 2250 m; above this the number of stations is too low to obtain robust results (Table S1).

Average winter temperatures were higher in the NW compared to the NE and SE, and the latter two were similar. In north and high Alpine and south and high Alpine, temperatures were also comparable, although northern sites were colder at 1500–2000 m. However, precipitation amounts were significantly lower in the south than north, and south and high Alpine sites received ≈ 100 mm less winter precipitation than north and high Alpine sites up to 2000 m, which amounts to ∼ 1/3 of the precipitation in the north. These results suggest that the difference in the December to February snow amounts in the north vs. south is predominantly driven by precipitation differences and not temperature.

Seasonal snow depth was correlated to temperature and precipitation extracted from a gridded reanalysis (MESCAN-SURFEX). Results indicated negative correlations with temperature decreasing strongly with elevation and positive correlations with precipitation mildly increasing with elevation (Fig. S5 in the Supplement). The magnitude of temperature correlations was between −0.8 and −0.5 below 1000 m, and the correlation decreased to about −0.2 up to 2000 m. For precipitation, correlations were between −0.2 and 0.7 with

much higher variability than temperature. Correlations of snow depth with temperature did not differ by region. However, the stations in the SE exhibited stronger (more positive) correlations with precipitation than the NE and NW regions.

The findings on the correlations agree with previous estimates for Swiss and Austrian stations (Schöner et al., 2019) in terms of signs and elevation patterns. However, our estimates are of higher magnitude for both temperature and precipitation. As a sensitivity analysis, we repeated the climatology and correlational analysis using observation-based spatial analyses instead of reanalysis for extracting temperature and precipitation (Figs. S6 and S7 and Tables S3 and S4 in the Supplement), but results did not differ substantially from above.

## 3.3 Long-term trends for the period 1971 to 2019

Trends of monthly mean snow depth from November to May were mainly negative with some exceptions (Fig. 7 and Table 2). Over all stations and all months, 85 % of the trends were negative and 15 % positive; 23 % were significantly negative and 0 % (only four station–month combinations) significantly positive (for significance, $p$ values had to be less than 0.05). The percentage of significant negative trends was substantially higher in the spring months (March to May) and at lower elevations irrespective of region, and it could reach 40 %–70 % (see also Table 2).

In the low-elevation regions (NE, NW, SE), snow depth was decreasing much stronger in the SE than in the NE or NW across all months. The mean trend of December snow depth below 1000 m in the NE was −0.7 cm per decade (all further trends in the same unit) and −0.8 in the SE, while in January, it was −0.5 in the NW, −0.6 in the NE, and −1.6 in the SE (Table 2). In February, NE stations even had an increasing snow depth at +0.8, while in the NW and SE it decreased. At middle elevations (1000 to 2000 m), differences between north and south were even stronger and variable in amplitude during the snow season; in December, the mean trend in north and high Alpine (N&hA) stations was more strongly negative (−1.9) compared to south and high Alpine (S&hA) stations (−0.8), but for January and February, we observe the opposite behaviour with a less pronounced negative trend in N&hA (−1.6 and −2.2) compared to S&hA (−3.9 and −5.1).

In the spring months of March and April, trends in snow depth were again more negative in the south than north. For example, at middle elevations (1000 to 2000 m), the mean March snow depth trend was −3.9 in N&hA compared to −7.0 in S&hA, in April −5.7 compared to −6.6, and in May −1.4 compared to −2.7. Notably, stations in S&hA above 2000 m exhibited strong variability in trends, and there were stations with increasing snow depth in all months (November to May). While mean trends were positive until January (November 2.7, December 4.0, January 0.0), mean trends

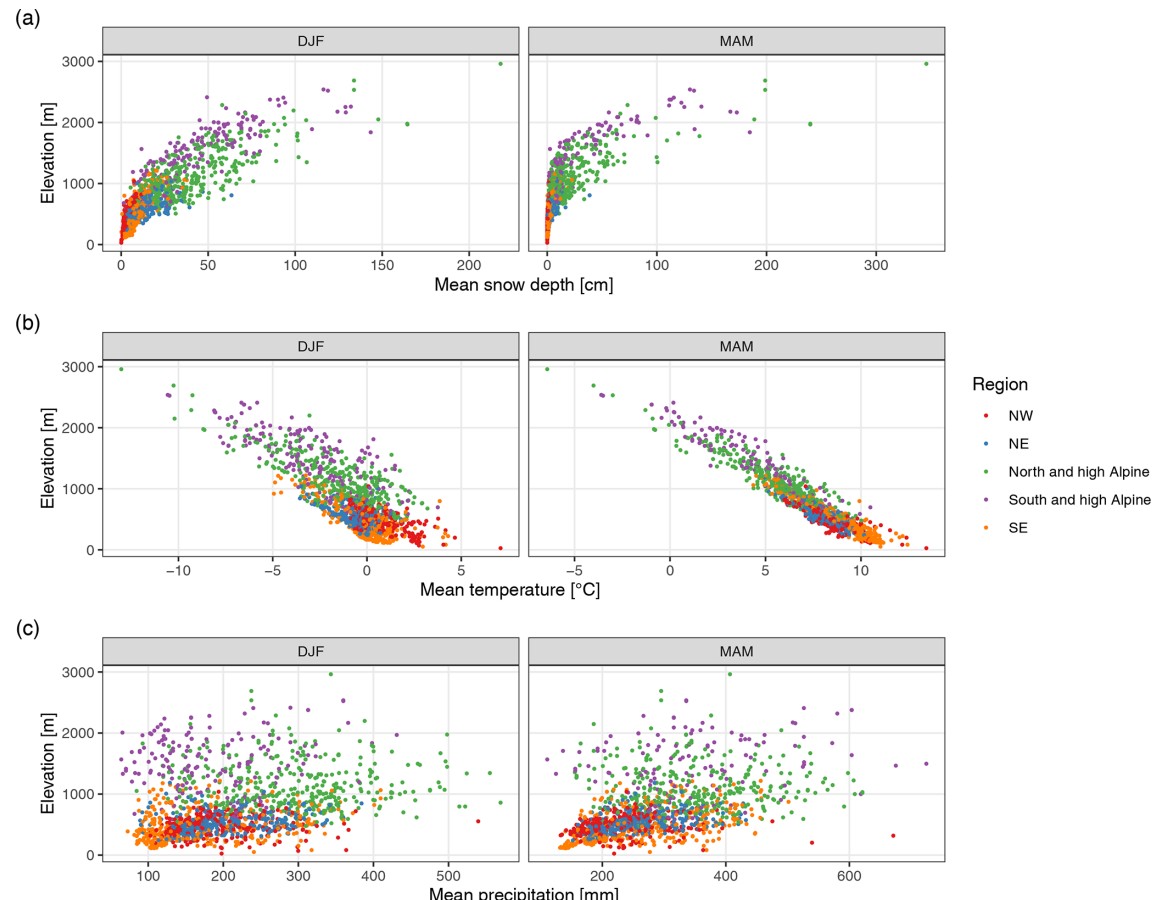

**Figure 6.** Climatology of **(a)** snow depth, **(b)** temperature, and **(c)** precipitation across regions and elevations for the winter season (December to February, DJF) and spring season (March to May, MAM). Average values are for the period 1981–2010. Each point represents one station. The temperature and precipitation values were extracted from MESCAN-SURFEX reanalysis, while the snow depths are based on station data. See also Tables S1 and S2 in the Supplement for summary values.

were negative otherwise (February −1.9, March −2.6, April −8.3, May −9.5).

## 3.4 Interannual variability from 1971 to 2019

Complementing the trend analysis, this section presents an evaluation of the interannual variability in snow depth series. Figure 8 highlights that mean snow depth exhibited a strong interannual variability in the analysed period. Because of the large number of stations, only time series that average over all stations in 500 m elevation bands are shown; however, individual station behaviour was well represented by the 500 m averages; see also auxiliary plots in the repository (Matiu et al., 2020). In the 1970s and 1980s, high snow depths were observed, followed by a period of extreme low snow depth in the 1990s. Since the 1990s, snow depths in winter have partly recovered, while in spring, snow depths have continued to decline. At the end of the snow season and for lower elevations, average snow depths approached zero, such as in April for 500 to 1000 m or in May for 1000 to 1500 m. The

different regions showed similar large-scale patterns, and, for example, the 1990s drop can be seen across the whole Alps. Particular years, especially extreme ones, show concurrent behaviour, for example February 1986 or 2009. Otherwise, there is mixed coherence across regions, as can also be seen from looking at standardized anomalies (Fig. B2) instead of raw snow depth.

These patterns are generally in line with those presented in previous studies which showed high snow amounts in the 1960s and 1980s and negative anomalies in the 1970s and 1990s, i.e. snow-scarce winters, regime shifts, or breakpoints in that period in France, Switzerland, Italy, and the western and southern part of Austria, and a recovery afterwards (Durand et al., 2009; Laternser and Schneebeli, 2003; Mallucci et al., 2019, 2019; Marcolini et al., 2017b; Marty, 2008; Micheletti, 2008; Scherrer et al., 2013; Schöner et al., 2019; Valt and Cianfarra, 2010). In an Alpine-wide view, this temporal variability is also accompanied by a strong regional variability.

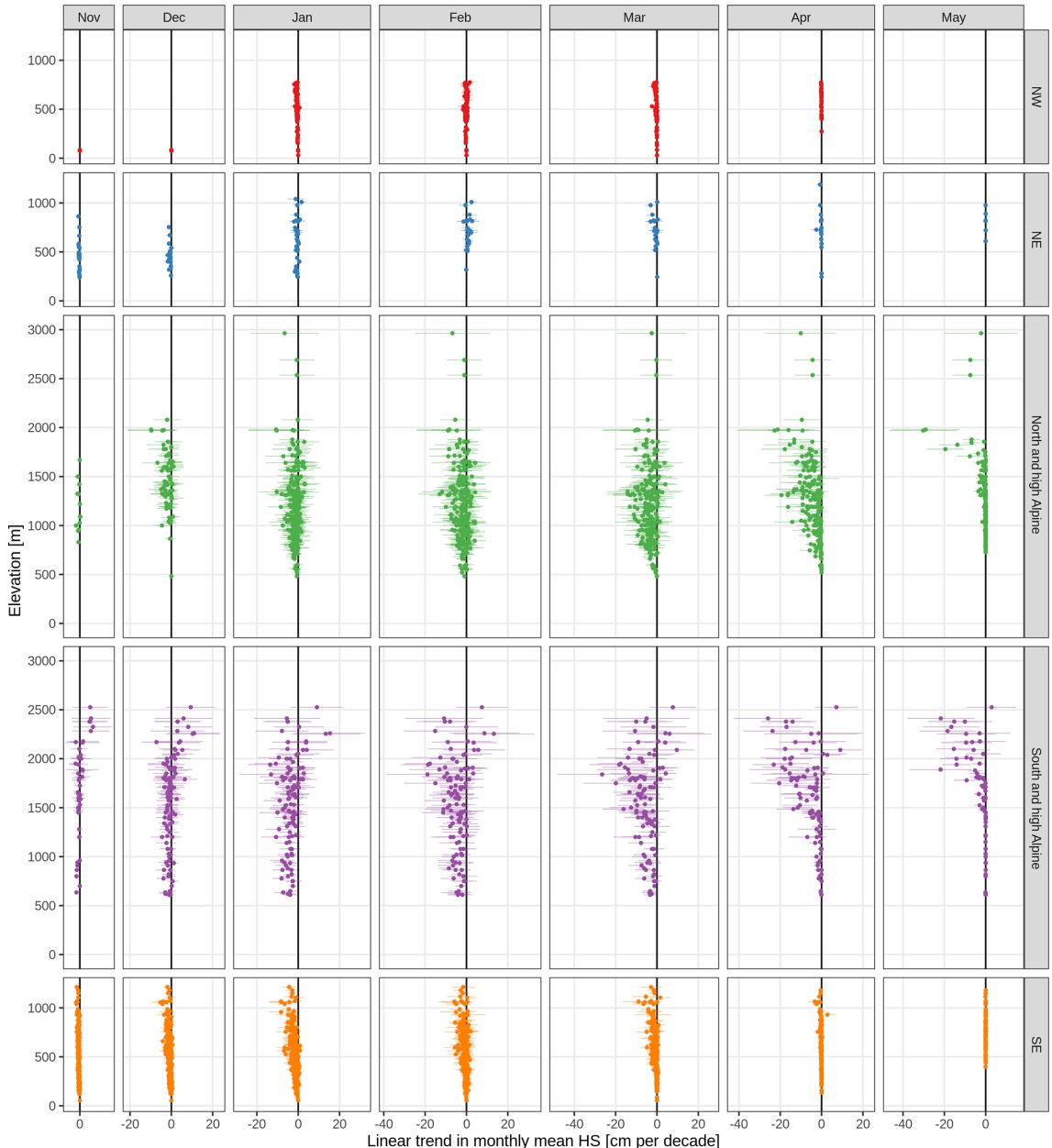

**Figure 7.** Long-term (1971 to 2019) linear trends in mean monthly snow depth (HS). Trends are shown separately by month (columns) and region (rows). Each point is one station. The points indicate the trend and the lines the associated 95 % confidence interval.

In order to put the trends from Sect. 3.3 in the context of interannual variability, we examined their relationship by looking at the ratio between the 1971 to 2019 trend and the SD of residuals (Fig. B3a). This gives an indication of the relative contribution of the trend to interannual variability. The highest ratios were observed in November to January below 1000 m, in March between 500 and 2000 m, in April between 0 and 2500 m, and in May between 1500 and 2500 m.

As expected from the high temporal variability in the snow depth series, the fraction of explained variance from the linear trends was low. The average $R^2$ over models with signifi-

cant trends ($p < 0.05$) was 10 %. However, $R^2$ increased with elevation and in the last months of the snow season reached up to 32 %.

From Fig. 8, a decrease in the variability in the snow depth series can be observed, especially at the end of the season and for lower elevations. This is confirmed by the large fraction of negative time coefficients for the error variance in April and May (Table B2) in which approximately 40 %–80 % of the stations presented significantly decreasing variability depending on the region. Notable decreases in variability were also observed in November and in January for the NE, NW,

**Table 2.** Overview of long-term (1971 to 2019) trends in mean monthly snow depth. Summaries are shown by month, region, and 1000 m elevation bands (0 to 1000, 1000 to 2000, and 2000 to 3000 m). Cell values are the number of stations (#), the mean trend (mean, in centimetres per decade), and percentages of significant negative (sig−) and positive (sig+) trends; the remaining percentage (not shown) corresponds to the total of non-significant negative and positive trends. Empty cells denote no station available (for # and mean) and no stations with significant negative or positive trends (sig− and sig+). Trends were considered significant if $p < 0.05$. See also Fig. 7. A version of the table with 500 m bands instead of 1000 m is available in the Supplement (Table S5).

| Month | Region | Elevation: (0,1000] m | | | | Elevation: (1000,2000] m | | | | Elevation: (2000,3000] m | | | |
|---|---|---|---|---|---|---|---|---|---|---|---|---|---|
| | | # | mean | sig− | sig+ | # | mean | sig− | sig+ | # | mean | sig− | sig+ |
| Nov | NW | 2 | −0.01 | 50.0 % | | | | | | | | | |
| | NE | 34 | −0.32 | 41.2 % | | | | | | | | | |
| | N&hA | 4 | −0.93 | 50.0 % | | 9 | −0.31 | | | | | | |
| | S&hA | 7 | −1.02 | 71.4 % | | 23 | −0.22 | | | 12 | 2.68 | | |
| | SE | 218 | −0.50 | 52.3 % | | 8 | −1.21 | 50.0 % | | | | | |
| Dec | NW | 2 | −0.01 | | | | | | | | | | |
| | NE | 24 | −0.68 | 29.2 % | | | | | | | | | |
| | N&hA | 3 | −1.72 | 33.3 % | | 67 | −1.91 | 1.5 % | | 1 | −2.02 | | |
| | S&hA | 17 | −1.34 | 5.9 % | | 67 | −0.89 | 1.5 % | 1.5 % | 17 | 3.98 | | |
| | SE | 221 | −0.77 | 24.9 % | | 9 | −2.38 | 44.4 % | | | | | |
| Jan | NW | 81 | −0.51 | 12.3 % | | | | | | | | | |
| | NE | 32 | −0.55 | 3.1 % | | 2 | 0.23 | | | | | | |
| | N&hA | 83 | −1.59 | 3.6 % | | 154 | −1.59 | 4.5 % | | 4 | −2.02 | | |
| | S&hA | 19 | −4.91 | 73.7 % | | 76 | −3.94 | 21.1 % | | 17 | 0.50 | | |
| | SE | 243 | −1.59 | 29.6 % | | 10 | −4.32 | 70.0 % | | | | | |
| Feb | NW | 78 | −0.09 | 11.5 % | | | | | | | | | |
| | NE | 24 | 0.75 | | | 1 | 2.44 | | | | | | |
| | N&hA | 84 | −1.36 | 4.8 % | | 153 | −2.24 | 7.2 % | | 4 | −3.56 | | |
| | S&hA | 19 | −4.10 | 10.5 % | | 78 | −5.09 | 15.4 % | | 17 | −1.91 | 5.9 % | |
| | SE | 228 | −0.63 | 4.4 % | | 12 | −2.50 | | | | | | |
| Mar | NW | 65 | −0.33 | 4.6 % | | | | | | | | | |
| | NE | 20 | −0.93 | 10.0 % | | 1 | 0.10 | | | | | | |
| | N&hA | 75 | −3.10 | 30.7 % | | 151 | −3.94 | 21.9 % | | 4 | −1.91 | | |
| | S&hA | 18 | −3.52 | 33.3 % | | 73 | −7.00 | 46.6 % | | 17 | −2.55 | 11.8 % | 5.9 % |
| | SE | 212 | −0.65 | 5.2 % | | 12 | −3.22 | 16.7 % | | | | | |
| Apr | NW | 34 | −0.08 | 23.5 % | | | | | | | | | |
| | NE | 18 | −0.33 | 38.9 % | | 1 | −0.73 | | | | | | |
| | N&hA | 69 | −1.48 | 68.1 % | | 133 | −5.70 | 65.4 % | | 4 | −7.07 | 25.0 % | |
| | S&hA | 14 | −0.92 | 50.0 % | | 65 | −6.63 | 56.9 % | | 17 | −8.28 | 41.2 % | |
| | SE | 136 | −0.13 | 38.2 % | 0.7 % | 7 | −1.42 | 14.3 % | | | | | |
| May | NE | 7 | −0.01 | | | | | | | | | | |
| | N&hA | 36 | −0.03 | 5.6 % | | 114 | −1.42 | 28.1 % | | 3 | −5.69 | | |
| | S&hA | 9 | −0.01 | 11.1 % | | 41 | −2.68 | 39.0 % | | 15 | −9.46 | 40.0 % | |
| | SE | 52 | −0.02 | | 1.9 % | 7 | −0.02 | | | | | | |

and SE. Considerable significant increases in variability, on the other hand, were only observed in December for 27 % of the south and high Alpine series.

## 3.5 Seasonal snow indices of snow depth and snow cover duration

In addition to the analysis of monthly mean snow depth from Sects. 3.3 and 3.4, this section gives a summary of trends in seasonal indices of mean and maximum snow depth, as well as snow cover duration (Table 3; Appendix C). The results of seasonal mean HS agree with the monthly analysis and show generally decreasing snow depths in winter up to 2000 m and in spring for all elevations. Maximum snow depth across the whole season (November to May) decreased stronger than mean snow depth; e.g. the average trend of mean HS for stations in the north (N&hA, NE, NW) between 1000 and 2000 m was −2.8 and −5.2 cm decade$^{-1}$ TS4 for maxi-

**Table 3.** Summary CE5 of 1971 to 2019 trends in seasonal snow indices. The five regions were collapsed into two (north and south). The number of stations differs by season, and the range of available series is indicated in the third column. Average trends (with minimum and maximum in parentheses) are given for seasonal indices of mean snow depth (meanHS), maximum snow depth (maxHS), and snow cover duration (SCD). The season is indicated in the second row with the first letter of the included months (e.g. NDFJ is November, December, January, and February). Absolute trends are in centimetres per decade for meanHS and maxHS and in days per decade ($d\,decade^{-1}$) for SCD. Relative trends are expressed as percent per decade ($\%\,decade^{-1}$). A few stations in the south below 1000 m were removed because their low and insignificant snow amounts caused unlikely high relative trends.

| Elevation (m) | Region | # series (range) | meanHS DJF | meanHS MAM | meanHS NDJFMAM | maxHS NDJFMAM | SCD NDJF | SCD MAM | SCD NDJFMAM |
|---|---|---|---|---|---|---|---|---|---|
| Absolute changes | | | $cm\,decade^{-1}$ | | | | $d\,decade^{-1}$ | | |
| (0,1000] | North | 141–190 | −0.9 (−5.3, 1.0) | −0.8 (−6.4, 0.1) | −0.8 (−4.7, 0.4) | −2.4 (−11.2, 3.1) | −2.8 (−11.5, 2.7) | −1.7 (−5.6, 0.1) | −4.5 (−13.6, 2.9) |
| | South | 224–241 | −1.2 (−6.0, 0.9) | −0.3 (−3.2, 0.3) | −0.7 (−3.6, 0.2) | −3.2 (−15.3, 3.1) | −3.7 (−10.7, 1.4) | −1.1 (−5.5, 1.2) | −4.8 (−14.6, 0.6) |
| (1000,2000] | North | 122–155 | −2.1 (−11.0, 3.1) | −3.7 (−21.9, 0.8) | −2.8 (−15.6, 1.6) | −5.2 (−19.9, 3.0) | −2.1 (−8.0, 5.0) | −3.0 (−7.5, 0.7) | −5.3 (−13.9, 0.7) |
| | South | 61–84 | −3.5 (−12.6, 2.3) | −4.9 (−18.7, −0.3) | −4.1 (−14.0, 1.6) | −9.8 (−29.2, 2.6) | −2.5 (−7.3, 1.7) | −4.1 (−8.3, 1.2) | −7.0 (−13.9, −0.2) |
| (2000,3000] | North | 3–4 | −4.3 (−9.9, −2.2) | −4.5 (−5.2, −4.1) | −5.0 (−8.2, −3.3) | −8.1 (−15.8, −4.2) | 0.1 (−0.1, 0.2) | | 0.1 (−0.1, 0.2) |
| | South | 16–17 | −0.1 (−9.2, 11.3) | −6.7 (−18.2, 6.6) | −2.9 (−11.5, 6.8) | −9.4 (−29.2, 6.1) | −0.2 (−2.1, 1.8) | −0.6 (−4.5, 1.9) | −1.0 (−4.7, 1.7) |
| Relative changes | | | $\%\,decade^{-1}$ | | | | | | |
| (0,1000] | North | 141–190 | −7.2 (−20.4, 12.1) | −11.2 (−20.6, 18.0) | −8.7 (−20.4, 10.0) | −4.7 (−19.4, 8.6) | −5.2 (−18.1, 7.8) | −9.7 (−28.5, 9.7) | −6.1 (−16.6, 6.9) |
| | South | 220–238 | −8.7 (−18.6, 22.7) | −7.5 (−21.7, 28.0) | −10.0 (−19.0, 10.6) | −6.8 (−16.7, 10.4) | −6.8 (−14.7, 8.2) | −8.0 (−19.5, 15.0) | −7.3 (−14.3, 4.4) |
| (1000,2000] | North | 122–155 | −3.6 (−17.6, 23.3) | −9.5 (−20.0, 3.1) | −6.2 (−18.3, 5.1) | −4.2 (−13.8, 3.9) | −2.0 (−8.5, 10.6) | −6.0 (−18.3, 4.0) | −3.5 (−11.8, 0.7) |
| | South | 61–84 | −6.5 (−14.1, 5.1) | −11.4 (−17.2, −1.0) | −8.9 (−14.8, 4.9) | −7.1 (−12.0, 3.0) | −2.6 (−8.8, 1.9) | −7.8 (−16.8, 4.2) | −4.7 (−10.7, −0.1) |
| (2000,3000] | North | 3–4 | −2.5 (−4.0, −1.5) | −1.8 (−2.0, −1.4) | −2.4 (−2.9, −2.1) | −2.1 (−3.1, −1.6) | 0.1 (−0.1, 0.1) | | 0.0 (−0.0, 0.1) |
| | South | 16–17 | 0.2 (−8.0, 11.7) | −4.2 (−11.9, 13.2) | −2.4 (−10.1, 6.7) | −3.4 (−9.1, 4.6) | −0.2 (−1.8, 1.6) | −0.7 (−5.3, 2.4) | −0.6 (−2.4, 0.8) |

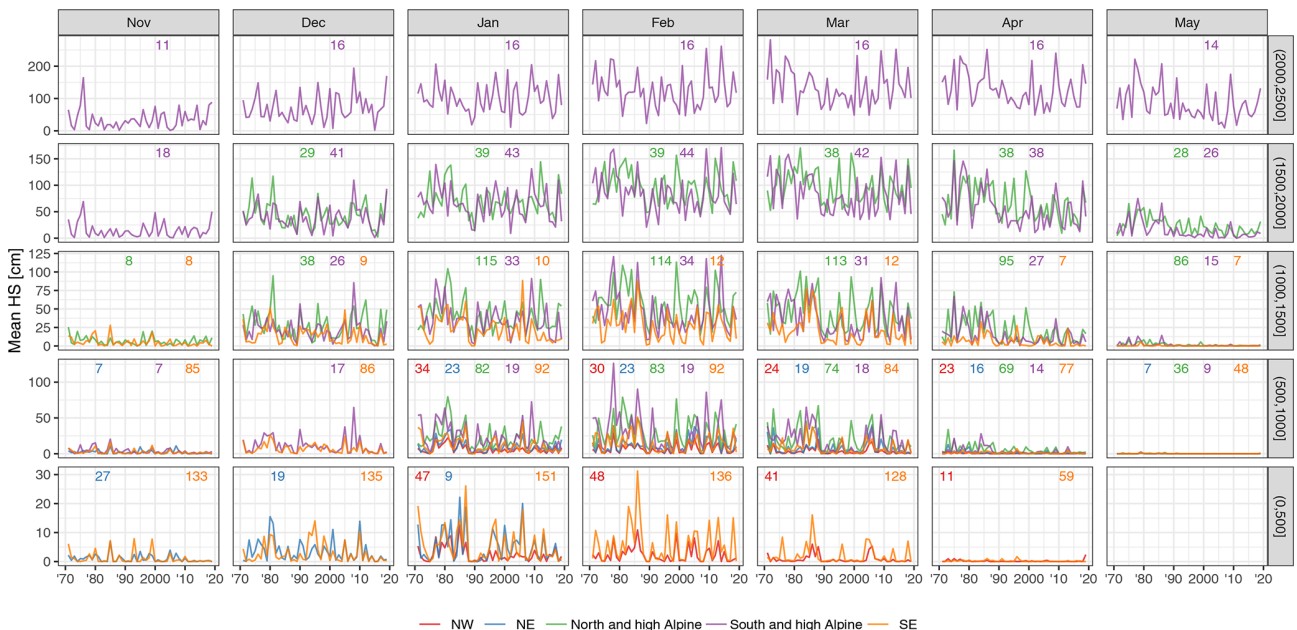

**Figure 8.** Time series of mean monthly snow depth averaged by 500 m elevation bands. The rows indicate elevation band and the columns the months. The small numbers at the top of each panel denote the number of stations included in the average. Lines are only shown if more than five stations were available. Time series of all single stations are available in the repository (Matiu et al., 2020).

mum HS, which corresponds to −6.2 and −4.2 % decade⁻¹, respectively. Again, stations in the south (S&hA, SE) had more negative trends, e.g. −4.1 cm decade⁻¹ for mean HS and −9.8 cm decade⁻¹ for maximum HS for the same elevations (1000 to 2000 m), which corresponds to −8.9 and −7.1 % decade⁻¹, respectively. Average relative trends below 1000 m were more negative than average trends between 1000 and 2000 m for meanHS (DJF and NDJFMAM) and all SCD indices but not that obviously for meanHS in MAM and maxHS.

Seasonal SCD also decreased for almost all stations below 2000 m, while above no consistent or significant changes were observed. The average trend in November to May SCD over all stations below 1000 m was −4.5 d decade⁻¹ in the north and −4.8 in the south, and over all stations between 1000 and 2000 m, it was −5.3 in the north and −7.0 in the south, respectively. The fact that above 2000 m no changes in SCD were observed might also be caused by our season definition (November to May), which is not always enough to capture the full season above 2000 m. In terms of relative changes, mean HS decreased stronger than maximum HS in our study, which is consistent with previous findings (Bach et al., 2018). However, in terms of absolute trends, the opposite was true for our study; mean HS decreased less than maximum HS. Another potential explanation for these differences might be the fact that the study period in Bach et al. (2018) starts 20 years earlier than in our study, and their maxHS trends are influences by some extreme events at the start of their study period.

In addition, our changes per decade for maximum HS and SCD are clearly smaller than the ones found by Klein et al. (2016) for a similar time period but a smaller number of stations in Switzerland. We were able to reproduce the exact estimates from Klein et al. (2016) for the same sites and hereby found that the differences were caused mostly by the different period (1970–2015 vs. 1971–2019), which makes sense since 1970 was a snow-abundant year, as were the years after 2015 compared to before 2015. In the case of SCD, the different season length (Klein et al., 2016, **TS5** used the whole year, and we used only November to May) also had an impact, especially for the higher-elevation sites. This supports our introductory statement on the challenge of synthesizing different studies and on the requirement of a unified analysis.

### 3.6 Representativeness of the stations in an Alpine-wide context

Since we aimed to give an Alpine-wide assessment, the horizontal and elevational coverage of the station observations is crucial in determining the confidence in the results. For this, we compared the elevation distribution of our station set with a digital elevation model (DEM) at 1 km resolution for the area spanned by the stations (Fig. 2c). In relative terms, the elevations of the stations used in this study oversampled the elevations up to 1000 m, were similar from 1000 to 2000 m, significantly underrepresented 2000 to 3000 m, and did not cover elevations above 3000 m.

If the absolute number of stations used in this study is deemed sufficient to describe the spatial coverage, then the

confidence of statements would be high for elevations up to 2000 m, while between 2000 and 3000 m, the results should be taken more cautiously. While the elevations above 3000 m only cover a minimal area (0.7 % of the area studied here; see Fig. 2c), they store large amounts of snow; Fig. 6a gives an indication of the expected increase in HS with elevation. Long-term monitoring is extremely challenging at elevations above 3000 m, and the snow cover at these elevations is relevant for hydrology, mountain ecosystems, glacier dynamics, and mountain (ski) tourism.

Spatial variability in snow increases with elevation (see also Fig. 6), and thus the absolute number of stations required for comparative assessments would be even higher for high elevations compared to low elevations. This limitation could be tackled with automatic snow depth sensors which better sample high elevations; however, their historical time series are yet too short for assessing long-term trends, besides their issue of harmonized data processing (see also Sect. 1).

An alternative method to derive spatially representative results is to transform the station point observations into a gridded product, by, for example, deterministic or geostatistical interpolation (i.e. kriging). However, in the complex topography of the European Alps with strong elevation gradients, it is challenging to determine an appropriate horizontal resolution that represents elevation well. Moreover, a high enough station density would be needed to perform interpolation. An observation-based grid of snow depth for the Alps would have many potential uses, from hydrological applications to the evaluation of remote sensing and climate models, but it is beyond the scope of this study.

## 3.7 Outlook

The scope of this study was primarily the detection of snow depth trends, thereby contributing to the better understanding and quantification of the state and evolution of the mountain cryosphere in the European Alps (Beniston et al., 2018; Hock et al., 2019). The formal attribution of the trends to climatic drivers, such as temperature and precipitation, as well as the influence of anthropogenic climate change on snow trends (Najafi et al., 2017; Pierce et al., 2008), is not explicitly addressed, although the collation of this unique dataset allows the scientific community to develop such studies in the future. The correlational analysis from Sect. 3.2 suggests that temperature and precipitation are important drivers of temporal and spatial variability in snow depth across the whole Alps.

Besides snow depth, observations of the depth of snowfall (HN) were also collected for this study which were used partly for quality checking the snow depth series. However, analysing the HN series and comparing results to those obtained for snow depth would have exceeded the scope of this study. In the future, we plan to continue with the analysis of HN series, for which we are also aware of other data sources, in particular for low-elevation sites in Italy (Pifferetti et al., 2017).

## 4 Conclusions

We presented the first Alpine-wide assessment of snow depth trends based on in situ measurements in the European Alps. This enabled the identification of five distinct snow regions whose spatial gradients are related to the known diverse climatic influences for the Alps.

The trend analysis, based on measurements from 1971 to 2019, highlighted the overall reduction in snow cover. Decreases in monthly mean snow depth from November to May were observed for 85 % of the station–month combinations (of which 26 % were significant, $p < 0.05$), while only 15 % showed increases (of which $< 1$ % were significant). Stronger negative trends with higher significance were observed in spring and in the case of low elevations during the whole season (Table 2). These are the times and elevations at which the transition from snow to snow-free occurs. The observed changes are thus consistent with the expectations from the snow albedo feedback (Thackeray et al., 2019) and highlight its importance for mountain climates (Pepin et al., 2015). Seasonal maximum snow depth decreased stronger than seasonal mean snow depth in absolute terms, while in relative terms, the opposite was true (Table 3). Snow cover duration decreased below 2000 m, while above no consistent change was observed partly due to our choice of snow season (November to May).

The different regions showed good agreement in the interannual variability for snow cover duration indices (Fig. C6) and less for snow depth variables (Fig. C4). The magnitude of trends differed by region, and the decreases in the south were on average stronger than in the north (Tables 2 and 3). Combined with the lower snow depths in the south than north (Fig. 6; Tables S1 and S2), this resulted in an even stronger relative decrease in the south than north. The number of stations analysed here gives high confidence to the changes up to 2000 m, while above this elevation, the changes have to be interpreted more carefully, especially in the north, where only a few stations were available.

The orography of the Alps clearly manifests as the main impact on the snow climatology. It defines boundaries for subregions in the north vs. south, followed by the west vs. east. The location of a station with respect to the climatic forcing zones defines the snow depth climatology and impacts the variability in snow depth at a daily scale. Additionally, it can result in different trend magnitudes and also trend signs. Besides these larger-scale features, substantial variability exists at higher elevations within the estimated snow depth regions. In summary, the assumption that results from one region are valid in another or for the whole European Alps needs to be evaluated cautiously.

This study provides a clear and harmonized picture for the detection of observed snow depth trends across the European Alps. Thereby it contributes to bridge a scientific gap which exists for many mountain areas in the world (Hock et al., 2019). We anticipate that the dataset developed for this study, a large part of which is made available to the broader scientific community, will provide support for further studies, in particular to formal attribution studies which quantify the anthropogenic component in the physical drivers of change and which remain extremely limited regarding snow cover trends (Najafi et al., 2017; Pierce et al., 2008).

A large community effort and open data sharing for research purposes has made this study possible. We have shown the benefits of a dataset that spans many nations and institutions. We expect this dataset to be used for further studies addressing various sectoral applications or for the evaluation of remote sensing or reanalysis products. Perhaps it might be expanded in the future thanks to additional contributing organizations. However, we currently lack the opportunity to have a continuously updated version. With ECA&D (European Climate Assessment & Dataset), a harmonized station data collection portal exists at the European scale for many meteorological variables. But while the coverage of, for example, temperature and precipitation is balanced across Europe, snow depth is only limited or not at all available for many European mountain regions, such as the European Alps, Carpathians, Balkan Mountains, or Dinaric Alps. It would be desirable to have an updated harmonized station dataset for snow cover given its importance in mountains and further downstream. This would enable a better monitoring of the changes and their consequences and impacts and contribute more quantitatively to climate change, ecosystem, and environmental assessments than is possible at the moment. However, such an endeavour requires a more formal umbrella and long-term commitment, e.g. in the framework of the Copernicus Earth monitoring programme of the European Commission.

## Appendix A: Data processing

After collecting the data, the series from different data providers were harmonized and put into a common data format. This included converting all station coordinates into latitude and longitude. In a few cases when only station name and elevation were available but no coordinates, the missing coordinates were extracted from Google Maps using the approximate location (with correct elevation) based on the station name. Most data providers used station identifiers along with station names. We chose to have unique identifiers for all stations based on the station name. Station names were standardized by replacing blanks and apostrophes with underscores and by removing accents. If multiple stations had the same name within one data source, i.e. by data provider, the names were suffixed with the station identifier from the data provider. If multiple stations had the same name across data providers, the names were suffixed with the data provider identifier.

### A1 Merging of records

The final database included several cases in which snow measurements for the same location were stored as separate records since they covered different periods and/or a slight relocation of the same station site occurred. In some cases, different records were available at very close locations where snow data were collected at the same time or over partially overlapping periods for different operative or research purposes. In order to maximize the temporal continuity and extent of available HS and HN series, the records referring to the same site or to very close locations were merged; one series was created from the multiple series by replacing missing values or missing periods. In particular, the merging was performed only if the sites were closer than 3 km and their vertical distance was less than 200 m. In the case of overlapping periods, the data from the series with the fewest gaps were retained. The merging was evaluated and performed on HS series first. In the case that HN series for the same sites were also available, the data were merged by following the same criteria used for HS in order to preserve consistency between HS and HN measurements. The metadata of the most recent series included in the merging were assigned to the resulting record. About 60 merged series were obtained in total, and the duplicate records for the same site were discarded.

### A2 Quality control

The series were quality checked in order to remove recording errors. First, below zero HS or HN values were replaced with missing values. Then a temporal consistency check was applied to HS to identify recording errors. Series were screened for jumps larger than 50 cm (up and down on 2 consecutive days, or vice versa). This criterion identified 680 values from the daily observations from all series, which were checked manually, and recording errors were replaced with missing values. Another issue with HS series is that missing observations might falsely be recorded as 0 cm. To identify suspicious series, mean winter (December to February) HS and the fraction of 0 cm values were calculated per station. Then, looking at a surrounding elevation band per station (200 to 500 m, depending on the elevation and station availability), series were marked if the mean HS was less than the 5th percentile or the fraction of 0 cm values was higher than the 95th percentile of all stations in the elevation band. Given the climatological nature of this pre-screening and the stronger dependence on elevation, we did not consider horizontal distance for this step. This resulted in 181 suspicious series which were checked manually. For 32 stations, there were periods when 0 cm was obviously a missing value, and in these periods, the 0 cm values were replaced with missing values; the remaining 149 stations had no missing values denoted as 0 cm. Finally, during all previous manual checks, series that showed "dubious" behaviour were marked, which were in total 48 series. Dubious behaviour was, for example, an inconsistency between HN and HS, unlikely values, improbable temporal variability, multiple seasons with no snow, or excessive gaps. From these 48 series, 29 were considered usable, 11 had some periods removed, and 8 were completely removed.

These procedures could identify some errors but definitely not all. Because of the large number of series, it was not feasible to manually quality check all of them, and fully automatic checks are often not feasible. Instead, a spatial consistency check was applied (see Appendix A.4), and the rest of remaining errors could be considered noise given the large amount of data.

### A3 Gap filling

Most series contained gaps ranging from some days up to whole seasons. In order to conduct climatological or trend analyses, gaps in the series needed to be filled. For this, we employed a spatial interpolation approach which is similar to the one used for temperature and precipitation records (see, e.g., Brunetti et al., 2006; Crespi et al., 2018; Golzio et al., 2018). The approach is based on correlations between the series, and because snow strongly depends on elevation, we first performed a spatial analysis to identify which correlations can be expected depending on horizontal and vertical distances between stations. For this, pairwise correlations (Pearson) between the daily HS series were performed for December to April from 1981 to 2010 only if the series had at least 70 % valid data and only if each pair had at least 50 % of data in common. As expected, correlations decreased with both horizontal and vertical distance (Fig. A1), but correlations remained high even for large distances; e.g. correlations higher than 0.7 were found for vertical distances of up to 500 m (with less than 100 km horizontal distance) or

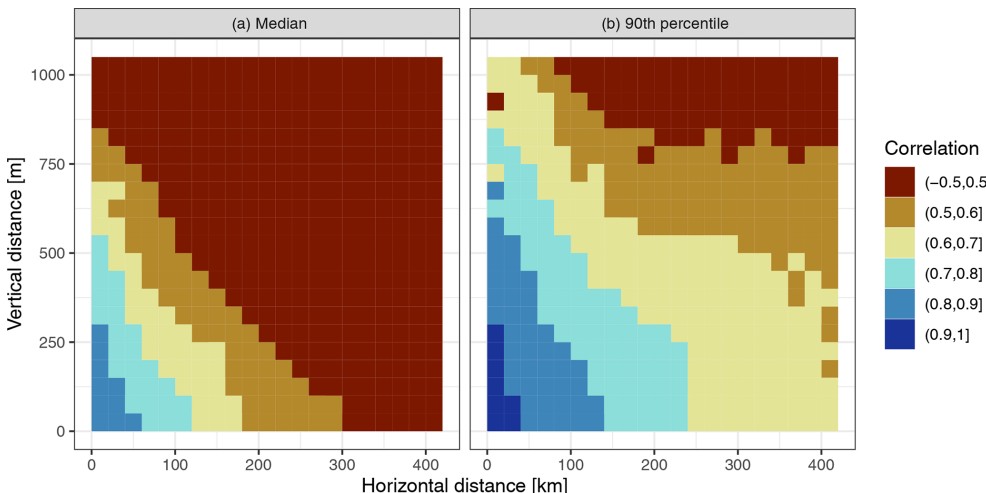

**Figure A1.** Summary of pairwise correlations between HS series for December to April, 1981 to 2010. The average (median, **a**) and 90th percentile **(b)** are shown of all pairwise correlations in bins of 20 km horizontal distance by bins of 50 m vertical distance. The correlations were only calculated if each series had at least 70 % valid data in the period and if each pair had at least 50 % of data in common.

horizontal distances of up to 200 km (with less than 250 m vertical distance). It should be noted that correlations can be high even if there are large differences in amounts or ratios between the series as long as the differences and ratios are constant across the range of values.

The chosen approach fills a gap based on finding neighbouring series that are highly correlated to the one with gaps. The gap-filling algorithm works as follows for each gap.

1. Find temporally surrounding non-missing values in the gap series around the gap date ("window data"); see also Fig. A2a.

   1.1. Take 15 d before and after the gap. This results in 31 d of the year; e.g. for Jan 15, this would be 1 to 31 January and for Jan 01, this would be 16 December to 16 January. TS6

   1.2. Repeat step 1.1 for 10 years before and after the gap. This results in 21 years; e.g. for 1996, this would be 1986 to 2006.

   1.3. This window data potentially contains 651 values ($21 \cdot 31$) but likely has missing values. If there are more than 150 non-missing values, continue to step 2. If there are less than 150 non-missing values, increase the day window by 5 d in both directions, and repeat from 1.1. If the day window has reached 45 in one direction (i.e. a total of 91 d) and still there are less than 150 non-missing dates, stop. Note that only the day window is increased, the year window from 1.2 stays constant at 10 years before and after.

2. Pre-select potential reference series (Fig. A2b) based on the following criteria: vertical distance to gap series is below 500 m, horizontal distance is below 200 km, and the value at the date of the gap is not missing.

3. For each potential reference series, do the following.

   3.1. Identify dates with values available for both gap and reference series in the window identified in step 1 (Fig. A2c). Continue only if more than 80 % of the minimum 150 non-missing values (i.e. 120) are available in common.

   3.2. For the common dates, calculate mean of gap series and mean of reference series, and calculate correlation between gap series and reference series. If all values of gap and reference series are zero, set the correlation to the minimum threshold (see step 4) plus 0.001 (in order to be able to fill also zero periods). If only one of the series has all zero values, i.e. either gap or reference but not both, set the correlation to zero.

   3.3. Calculate ratio between mean of gap series divided by mean of reference series. If the mean of the reference series (divisor) is zero, set the ratio to zero (in order to be able to fill also zero periods).

4. Sort potential reference series by correlation with gap series (from step 3.1). Remove all candidates with a correlation below 0.7. This threshold was chosen as it is used, for example, in the homogenization of snow depth (Marcolini et al., 2017a).

5. Select the first five best correlated reference series or up to five, depending on how many are available.

6. Calculate weights based on vertical distance. The weights are based on exponential decay with a halving distance of 250 m ("half-time" transformation of decay constant). This implies that the weights are halved every 250 m.

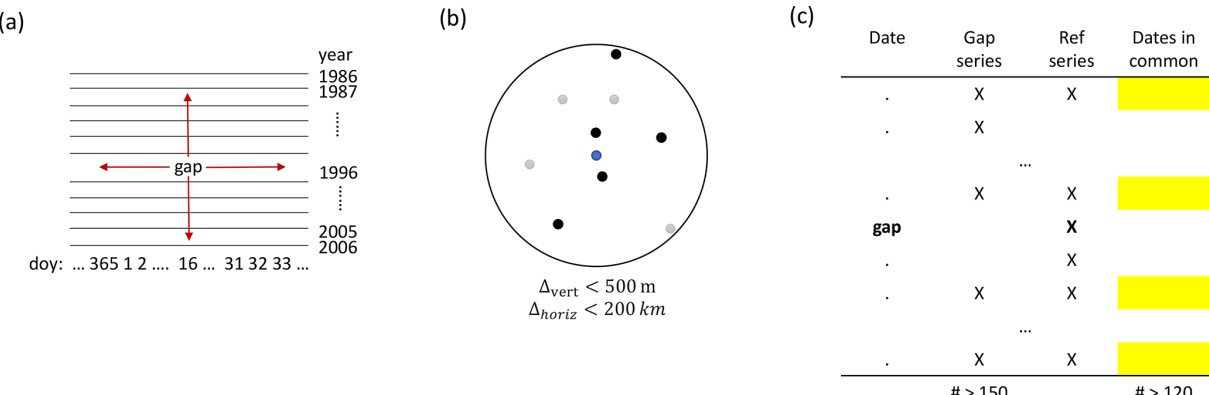

**Figure A2.** Visualization of some steps of the gap-filling algorithm. Panel **(a)** shows how the window data in the gap series around the gap is determined (step 1); doy is day of the year. Panel **(b)** shows the selection of potential reference series by horizontal and vertical distance (step 2). Panel **(c)** shows how common dates for gap and reference series are identified (step 3); the dates come from the window in **(a)**.

7. Fill the gap value with a weighted (step 6) average of the reference series values adjusted by the ratios between gap and reference series (step 3.3): $\mathrm{HS}_t^{\mathrm{gap}} = \frac{1}{n}\sum_{i=1}^{n} w_i \cdot \mathrm{HS}_t^{\mathrm{ref}_i} \cdot \frac{\mathrm{HS}_{\mathrm{mean}}^{\mathrm{gap}}}{\mathrm{HS}_{\mathrm{mean}}^{\mathrm{ref}_i}}$, where $t$ is the date of the gap, $i$ is the index of the reference series, $n$ is the number of the reference series $1\ldots5$, and $w_i$ are the weights with $\sum_i w_i = 1$.

The filled value was rounded to the nearest integer value in centimetres. Since the method requires finding suitable reference stations, it was only performed for the period 1961 to 2020 because the station density was too low before. The gap filling was applied to all gaps in all series considering all available data; afterwards, thresholds were applied to select usable series (see end of this section).

The chosen limits of 200 km horizontal distance and 500 m vertical distance might seem very high in the Alpine context with the complex topography. Since we were interested in larger-scale snow patterns and not local snow peculiarities, such large distances are justified. Moreover, the correlation threshold should exert control on selecting only stations that share the same snow cover evolution, and high correlations were found up to these horizontal and vertical limits (Fig. A1). On the other hand, a nearby station might also be a worse predictor than a more distant one, if, for example, it differs in its local climate.

Since this gap-filling approach has not yet been used for snow depth, we performed a cross-validation analysis to identify the gap-filling errors. For this, we used data from November to May from the period 1981 to 2010. For each station and each year, 1 month at a time was held out but only if at least 10 d were available. Thus, for each month, a maximum potential of $\sim 900$ values were cross-validated; however, the effective number was lower because of missing values and because not all gaps could be filled if no suitable reference stations were available. In order to test the effect of shorter period gaps, we also applied the cross-validation to subsets (to reduce computation time): (1) 100 random samples of 1 d and (2) 20 random samples of 5 consecutive days. Then, the held-out values were filled using the above approach, and metrics were calculated based on the filled and held-out values. Metrics include the bias, the MAE (mean absolute error), the MAE for non-zero held-out values only, and a modified version of relative MAE. The relative MAE is based on the MAE for non-zero values only, and this non-zero MAE is divided by the average of the held-out non-zero values. This is then not a "true" relative error which would divide each error by the true value, i.e. $\frac{1}{n}\sum_{i=1}^{n}|\frac{y_i-x_i}{x_i}|$, but our modification is $\frac{1}{n}\sum_{i=1}^{n}\frac{|y_i-x_i|}{|\underline{x}|}$, where $\underline{x}$ is the average of all $x_i$. This was done to remove the large influence of errors close to zero which are not that relevant in this case. The metrics were only calculated if more than 50 values were available per month and station (out of potentially $\sim 900$ for the month-long gaps and 100 for the 1 and 5 d gaps) in order to provide robust estimates.

The cross-validation showed that the gap filling has extremely little bias (Table A1), with the overall average daily bias for the month-long gaps being $-0.04$ cm. Average daily MAE for filling whole months was 1.6 cm (averaged over stations located at 0–1000 m), 7.7 cm (1000–2000 m), and 22.0 cm (2000–3000 m). MAE was lower for 1 and 5 d long gaps compared to month-long gaps, but almost no differences were observed comparing 1 or 5 d, e.g. for the 1000–2000 m band. MAE for 1 d gaps was 6.2 cm and for 5 d gaps 6.4 cm compared to 7.7 cm for 1 month gaps. The relative MAE of month-long gaps decreased with elevation from 39.4 % (0–1000 m) to 32.7 % (1000–2000 m) to 22.8 % (2000–3000 m). Additionally, there was also a seasonal dependence of MAE, while the bias remained largely constant across the season (Fig. A3). MAE below 2000 m peaked in February, while above 2000 m, MAE increased throughout the season. Relative MAE decreased with higher snow depths both temporally and with elevation; that is, relative MAE was lowest in

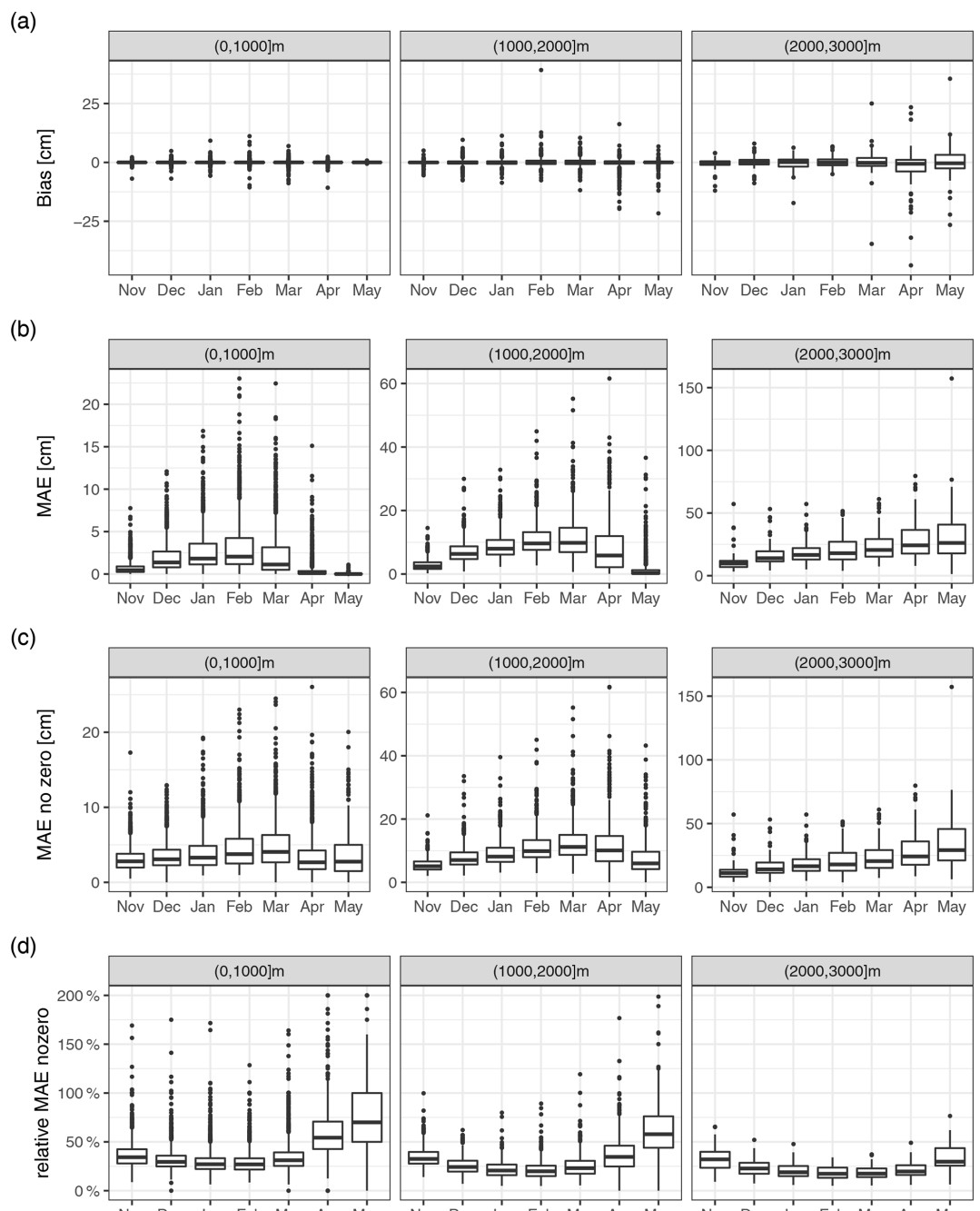

**Figure A3.** Cross-validation metrics for the gap-filling approach: (**a**) bias, (**b**) mean absolute error (MAE), (**c**) mean absolute error for non-zero values (MAE no zero), and (**d**) non-zero MAE divided by the true non-zero mean (relative MAE no zero). Panels show the 1000 m elevation bands indicated in the title. The boxplots represent statistical quantities. The box indicates the first and third quartile, the bold line inside the box is the median, the vertical lines outside the box extend up to the most extreme point but at most 1.5 times the interquartile range (IQR; height of the box), and, finally, points below and above 1.5 · IQR of the first and third quartile are shown as separate points.

February and at high elevations. It is to be expected that errors at the end of the season are related to the ablation scheme (i.e. local climatic and topographic characteristics that influence ablation) of the different stations; however, at this stage, we did not check this issue further.

Moreover, we compared our proposed gap-filling approach to results from gap-filling snow depth series using simulations of the Crocus snow model for the French Alps. The Crocus simulations with meteorological forcing were performed independently of this study, but we found it useful

**Table A1.** Cross-validation (CV) metrics for the gap-filling approach: bias (the difference between gap-filled and observed values), the mean absolute error (MAE), mean absolute error only for non-zero observed values (MAE no zero), and MAE no zero divided by the average of all true non-zero values (Rel. MAE no zero).

| Elevation band (m) | CV period | Bias (cm) | MAE (cm) | MAE no zero (cm) | Rel. MAE no zero |
|---|---|---|---|---|---|
| (0,1000] | 1 d | −0.0 | 1.3 | 3.1 | 30.1 % |
|  | 5 d | −0.0 | 1.4 | 3.3 | 34.0 % |
|  | 1 month | −0.0 | 1.6 | 3.9 | 39.4 % |
| (1000,2000] | 1 d | −0.1 | 6.2 | 7.9 | 26.1 % |
|  | 5 d | −0.1 | 6.4 | 8.2 | 28.5 % |
|  | 1 month | −0.1 | 7.7 | 9.7 | 32.7 % |
| (2000,3000] | 1 d | −0.6 | 18.2 | 18.6 | 18.9 % |
|  | 5 d | −0.8 | 18.3 | 18.7 | 19.2 % |
|  | 1 month | −0.4 | 22.0 | 22.5 | 22.8 % |

to compare the two approaches – albeit only exploratively. The observed snow depths with gaps were assimilated into the Crocus modelling scheme using SAFRAN reanalysis data as forcing (López-Moreno et al., 2020). The two gap-filling approaches were compared only for existing gaps in the French Alps. This was intended as a preliminary companion evaluation, and no cross-validation was performed. Thus, there was no ground truth to evaluate the two gap-filling approaches with formal metrics, and we only performed a visual assessment (figures for comparison available in Matiu et al., 2020). Time series of both gap-filling procedures looked remarkably similar even for reconstructions of complete missing seasons; the different snowfall events were visible in both, and snow depths averaged over multiple days were comparable. Differences emerged in the snow settling behaviour and for the spring snow melting periods. More information on this exercise is available from the authors on request.

For Switzerland, a comparison of gap-filling methods for HS was performed which aimed at reconstructing complete missing seasons and which included regression-based methods and snow models (Aschauer et al., 2020). While our proposed method was not explicitly used in that comparison, it can be assumed to be similar to the regression-based and distance-weighted methods used there. The errors reported in their study (root mean squared error less than 20 cm) are in the same order of magnitude as those found in our cross-validation.

Altogether, the above-mentioned points (the cross-validation results, the comparison to Crocus, and the preliminary findings of the Swiss study) convinced us that the gap-filling procedure is also suitable for reconstructing whole seasons and not only some intermediate gaps, considering the fact that we only used it to derive monthly means (see below) and did not use the daily values directly. Further research would be required to check the suitability of the daily reconstructions, in our opinion, also considering the temporal distance to the last existing observations. For the final analy-

sis, all gap-filled data within the recording period was used, and we also allowed the period to be extended by up to 5 years before the start or after the end of the recordings – but only if the total number of gap-filled observations was less than the number of observations without gap filling. The main reason for this extension was to have series covering the complete period until 2019 because some series stopped just a few years earlier. As a sensitivity analysis, we repeated most of the statistical analysis also for the original data without gap filling and provide results in the Supplement; the estimated modes of variability matched (Fig. S12 in the Supplement), the magnitude and variability in monthly trends were similar, although significantly fewer stations were available (Fig. S10 in the Supplement), and finally the time series of 500 m average HS also showed similar behaviour (Fig. S11 in the Supplement). The gap filling was able to significantly increase the temporal availability, but its aim was not to fill all gaps. Gaps were not filled, for example, if no suitable reference station was found or if not enough common data were available.

## A4  Aggregation and spatial consistency

The daily snow depth (HS) values were aggregated to mean monthly HS if at least 90 % of the daily values were available in the respective months after the gap filling (monthly time series plots available at Matiu et al., 2020).

Based on the monthly series, a consistency check was performed (Crespi et al., 2018) which identified dubious values and/or series (but can also identify series with strong local influences on snow depth). Each monthly HS series of the tested station was reconstructed from up to five reference stations by a spatial interpolation approach. The reference series were selected if the monthly record was available and if at least 10 monthly records were in common with the tested station. If more than five neighbours were available, the ones with the highest weights were selected with weights being derived from the horizontal distance and eleva-

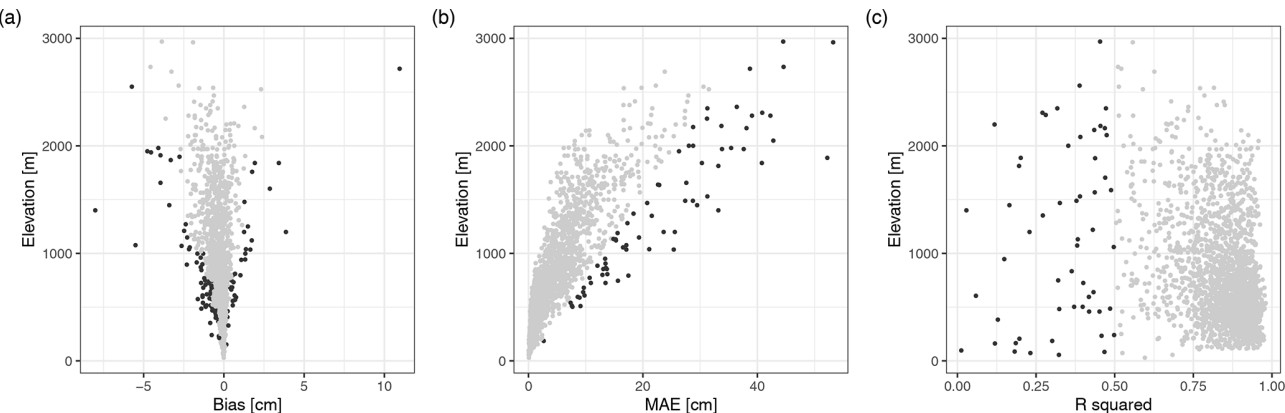

**Figure A4.** Metrics for spatial consistency: **(a)** bias, **(b)** mean absolute error (MAE), and **(c)** $R^2$ TS7 (squared correlation). Metrics were derived from statistical simulations of the monthly series from December to February using spatial neighbours. Black points indicate stations which were further analysed with manual checks.

tion difference, which is similar to the gap-filling procedure described above. Each reference station value was rescaled by the ratio between tested and reference mean HS for the month under reconstruction. Finally, the monthly simulation of the tested series was defined as the median of up to five rescaled neighbouring values. The comparison between simulated and observed monthly HS series for each station was evaluated by computing bias, MAE, and $R^2$ (squared correlation) from December to February in order to avoid unreliable low error values due to zeros in HS records outside of winter.

The mean bias over all stations was $-0.3$ cm (min, max: $-8.0$, 10.9 cm), average MAE was 4.8 cm (0.1, 61.3 cm), and average $R^2$ was 0.83 (0.0, 0.98). However, there was a strong elevational dependency, and station metrics deteriorated with elevation (Fig. A4). A semi-automatic approach was considered to look for suspicious series. The following criteria were used to screen stations: bias outside the 95 % confidence interval per elevation band (250 m bands up to 1500 m, then 1500 to 2000 m, and 2000 to 3000 m), MAE above a manually defined threshold line (see Fig. A4b), $R^2$ below 0.5, or simulation not successful because of too many gaps. This yielded 225 stations which were checked manually by looking at monthly simulated and observed series and daily series. Only 14 stations were found suspicious and 18 partly suspicious; all of these 32 series were removed from the statistical analyses. More detailed results and time series comparing simulated with observed snow depths are available as auxiliary material (Matiu et al., 2020).

## Appendix B: Additional figures and tables

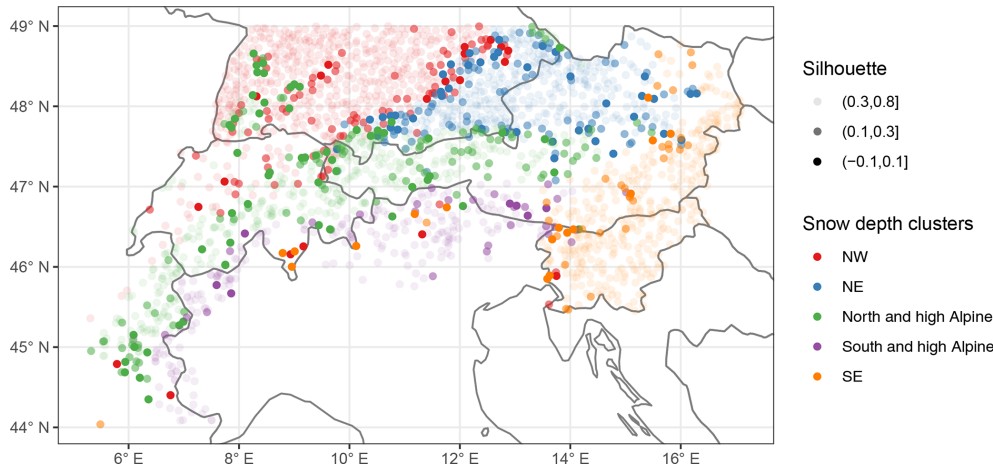

**Figure B1.** Silhouette values of the stations which show the consistency of clustering. The silhouette is a measure of how similar the station is to its own cluster compared to the other clusters (see methods CE6 for formula). High values indicate a good match, while low and negative values indicate a poor match.

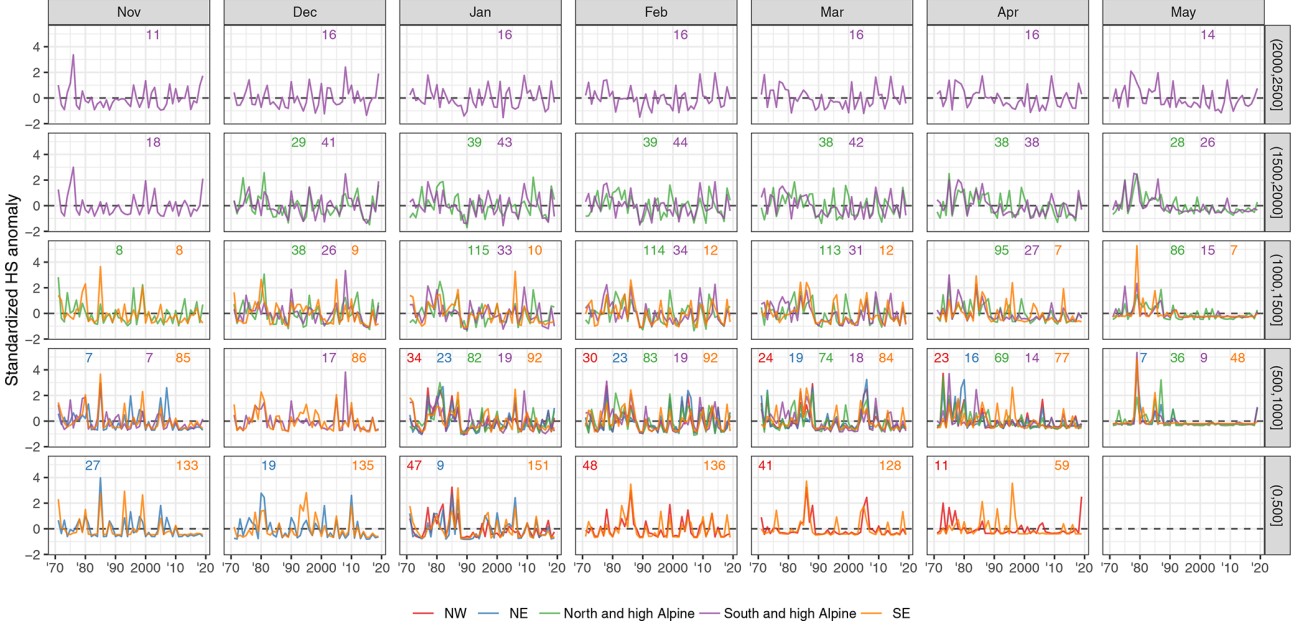

**Figure B2.** Same as Fig. 8 but using standardized anomalies.

**Table B1.** Overview of the literature on snow cover trends in the European Alps.

| Area | Number of stations | Time period | Season | Snow variable | Methods |
|---|---|---|---|---|---|
| (Bach et al., 2018)<br>Mean HS $-12.2\,\%/10\,\mathrm{yr}$ [TS8] ($40\,\%$ station [CE7] $p < 0.05$); max HS $-11.4\,\%/10\,\mathrm{yr}$ [TS9] ($36\,\%$ station $p < 0.05$); except coldest climates<br><br>Pan-Europe: mostly Germany, Benelux, AT-Tirol, *Czech* Republic, Slovakia, Finland; partly UK, Balkan, partly east [CE8] of Baltic Sea | (not specified) | 1951–2017 | DJF | Mean HS; Max HS (95 pctl) | OLS if trend pos.; OLS (exp) if trend negative; significance: Mann–Kendall |
| (Beniston, 2012b)<br>$10\,\%$–$50\,\%$ decline in DJF HS (less of a decline in the moist north vs. dry south)<br><br>Switzerland | 10 | 1930–2010 | DJF; NDJFMA | Mean HS; snow days (10 cm) | Visual: 5 yr moving window |
| (Durand et al., 2009)<br>HS no trend at 2700 m, decreases below; $n0$ (number of days with snow) negative trends<br><br>French Alps | Modelling:<br>ERA-40,<br>SAFRAN,<br>Crocus | 1959–2005 | DJF | Mean HS<br>$n0$<br>HS100d (minimum 100 d snow depth) | 300 m elevation steps (1500–2700); Spearman correlation (year $- n0$); step-year ($n0$); linear trend ($n0$) |
| (Klein et al., 2016)<br>SCD shorter $8.9\,\mathrm{d\,decade^{-1}}$; more because of earlier snow melt ($5.8\,\mathrm{d\,decade^{-1}}$); decrease in maxHS and earlier date of maxHS<br><br>Switzerland | 11 | 1970–2015 | Sep–Aug | maxHS; date of snow onset, snowmelt, maxHS; SCD; snow days (1, 20, 50, 100) | Theil–Sen, Mann–Kendall; stepwise regression |
| (Kreyling and Henry, 2011)<br>150 stations showed a decrease ($p < 0.05$ for 69) and 22 an increase ($p < 0.05$ for 1); decrease accelerated over the last 15 yr ($-0.48$ to $-0.89\,\mathrm{d\,yr^{-1}}$)<br><br>Germany | 177 | 1950–2000 | Aug–Jul | SD (1 cm) | OLS; random effects with stations |
| (Laternser and Schneebeli, 2003)<br>All variables show an increase until 1980, followed by a significant decrease; trends more pronounced at middle and low elevations; south = north; shorter SCD because earlier melt in spring<br><br>Swiss Alps | 140 (HS)<br>120 (HN) | 1931–1999 | NDJFMA;<br>2 month splits | Mean seasonal HS; SCD (start, end, length); days with HN $> 0$, 10, …; HN3max (max 3 d HN) | Trend analysis; relative to long-term mean; trend of short period equal to long period |

| Area | Number of stations | Time period | Season | Snow variable | Methods |
|---|---|---|---|---|---|
| (Lejeune et al., 2019) 39 cm less in 1990–2017 vs. 1960–1990 | | | | | |
| France | 1 (Col de Porte) | 1960–2017 | DJFMA | mean HS | Moving window (15 yr) and comparison 30 yr |
| (Marcolini et al., 2017b) Different dynamics above and below 1650 m; larger reductions at lower elevations; strong change late 1980s | | | | | |
| Italy (BZ + TN) | 37 | 1980–2009 | NDJFMA | SCD (> 30 cm); seasonal HS | Homogenization; Hovmöller plots; wavelet analysis |
| (Marty, 2008) Regime shift at end of 1980s, no clear trend since then | | | | | |
| Switzerland | 34 | ~ 1931–2008 | DJFM | Snow days (5, 30, 50 cm) | Mann–Kendall; shift detection |
| (Marty and Blanchet, 2012) 44 % of stations show significant decrease in HSmax, 32 % for HN3max; decrease in spread of HSmax | | | | | |
| Switzerland | 18 (HSmax) 25 (HN3max) | 1931–2010 | Annual | HSmax (annual max HS); HN3max (annual max sum HN 3 d) | GEV CE9 with time-dependent location and shape |
| (Marty et al., 2017) SWE decline (independent of lat or long); stronger and more significant decrease in spring (−80 % to −10 % low to high elevations/60 years) than winter; winter: some positive non-significant at high elevation | | | | | |
| alpine-wide (AT, FR, DE, IT, CH) | 54 | 1968–2012 | Index values (spring and winter) | SWE (not continuously measured) | Mann–Kendall; Theil–Sen |
| (Micheletti, 2008) Positive anomalies until the end of the 1980s then shift to low snow amounts until beginning of 2000; some recovery but still below level of 1980s | | | | | |
| Italy (FVG) | 8 | 1972–2007 | Seasonal | sumHN, max of monthly meanHS | Time series (only descriptive); % anomalies w.r.t. 1972–2007 |
| (Scherrer et al., 2013) Strong decadal variability; high values 1900–1920 and 1960–1970/80; lowest values end 1980/1990; increases/plateau in 2000s linked to temperature evolution | | | | | |
| Switzerland | 9 | 1864–2009 | Annual | MAXNS (max annual HN); NSS (sum annual HN); DWSF (days with snowfall) | Plots; 20 yr smooth; comparison to 71 other stations |

| Area | Number of stations | Time period | Season | Snow variable | Methods |
|---|---|---|---|---|---|
| (Schöner et al., 2009) Largest HS in 1940s/50s; summer snow decreasing; interannual variability in winter precipitation closely related to HS (highest in 40s/50s TS10; strong decreases since → TS11; less extremes) | | | | | |
| Austria | 1 (Sonnblick) | 1928–2005 | Monthly | HS | (Visual) CE10 |
| (Schöner et al., 2019) EOF groups AT–CH in seven regions; trend analysis based on first PC; strong trends in the south at ~2000 m: up to −12 cm 10 yr⁻¹ TS12; strongest trends at highest elevations; regional dependence of trends | | | | | |
| Switzerland and Austria | 196 (139 passed QC) | 1961–2012 | NDJFMA | Seasonal HS and HN | MK test with lag1 pre-whitening; running trend approach; Sen slope; EOF for regionalization |
| (Terzago et al., 2010) More snow Nov–Dec, less Jan–Apr, disappeared in May | | | | | |
| Italy (Piemonte) | 3 | 1971–2009 | Monthly | HN, HS, snowy days (HN ≥ 1 cm) | 1971–2000 vs. 2000–2009 |
| (Terzago et al., 2013) Some maxima in 1940, 1950, 1960, 1970, absolute minima 1990, then recovery; significant decrease in seasonal HS of 2–14 cm decade⁻¹; stronger decreases in north (considering elevation); changes not driven by precipitation changes; snowfall anticorrelated to NAO CE11 | | | | | |
| Italy (west) | 6 | 1926/1951–2010 | DIF, MAM, NDJFMAM | Precipitation, days with precipitation, solid precipitation fraction, HN, snowy days (HN > 0), HS | Trend analysis; Mann–Kendall; spectral analysis |
| (Valt et al., 2008) Snow cover decreased 14 d (1991–2007 vs. 1960–1990), stronger < 1600 m (16 d) vs. > 1600 m (11 d); fresh snow decreased 1990–2000, then stationary (for all altitudes and months) | | | | | |
| Italy (east and west) | 5 (west); 6 (east) | ~1920/1960–2007 | Oct–May | Snow days (≥ 1 cm); sumHN | (Visual) |
| (Valt and Cianfarra, 2010) NDJFMA CSF CE12 shows −3 to −40 cm 10 yr⁻¹ TS13 for all 18 stations from 1960 to 2009, SCD also all negative; breakpoint ~1990 before decrease, after increase; strongest negative trend in spring and below 1500 m; negative trend related to precipitation decrease; PCA shows long-term negative trend | | | | | |
| Italy (east and west) | 18 | 1950–2009 | DJFMA; DJF; MA | SCD (> 1 cm); CSF (sum of new snow) | Split by 1500 m alt; OLS, Mann–Kendall; changepoint; PCA |

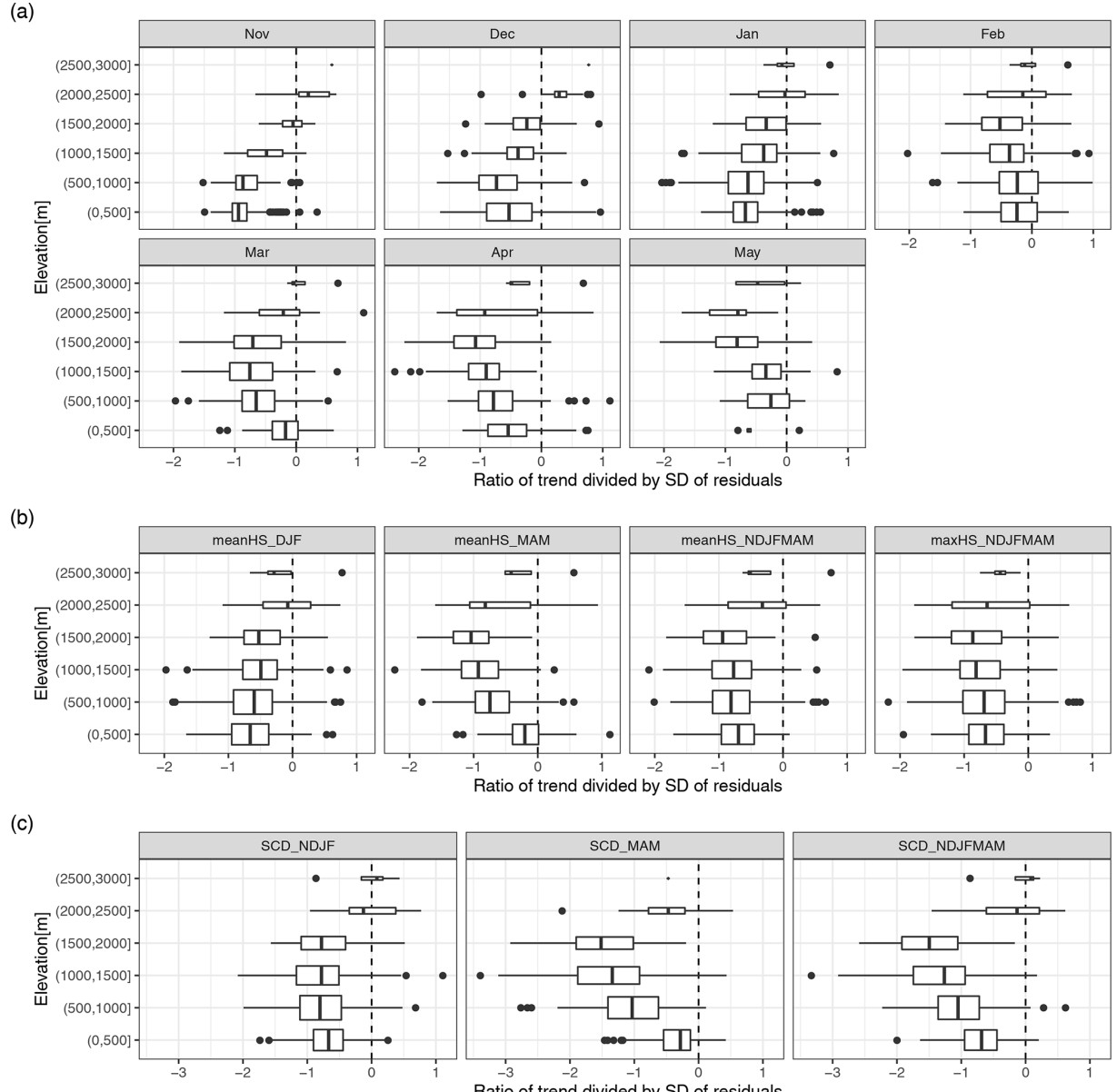

**Figure B3.** Ratio between the trend over the full period (1971 to 2019) and interannual variability (SD of the residuals). **(a)** The values for monthly mean HS (snow depth), **(b)** for seasonal indices of HS, and **(c)** for seasonal indices of SCD (snow cover duration). The boxplots represent statistical quantities. The box indicates the first and third quartile, the bold line inside the box is the median, the vertical lines outside the box extend up to the most extreme point but at most 1.5 times the interquartile range (IQR; width of the box), and, finally, points below and above 1.5 · IQR of the first and third quartile are shown as separate points. The height of the box is proportional to the number of observations in each group.

**Table B2.** Fraction of models with significantly positive or negative changes in the error variance by time. The remaining percentage (not shown) corresponds to the total of non-significant negative and positive changes. Empty cells indicate no stations with significant negative or positive trends (sig− and sig+). Changes were considered significant if the GLS model with a time coefficient for the error variance showed significantly improved goodness of fit compared to the OLS model with constant error variance ($p < 0.05$).

| Region | Nov | | Dec | | Jan | | Feb | | Mar | | Apr | | May | |
|---|---|---|---|---|---|---|---|---|---|---|---|---|---|---|
| | sig− | sig+ | sig− | sig+ | sig− | sig+ | sig− | sig+ | sig− | sig+ | sig− | sig+ | sig− | sig+ |
| NW | | | | | 86.4 % | | 24.4 % | 5.1 % | 30.8 % | 3.1 % | 76.5 % | 11.8 % | | |
| NE | 47.1 % | 2.9 % | 16.7 % | | 47.1 % | 2.9 % | 4.0 % | 8.0 % | 28.6 % | 4.8 % | 78.9 % | | 80.0 % | |
| N&hA | 53.8 % | | 4.3 % | | 28.9 % | 0.4 % | 5.5 % | 0.4 % | 22.4 % | | 72.8 % | | 75.3 % | 4.7 % |
| S&hA | 43.9 % | 4.9 % | | 26.7 % | 8.0 % | 8.0 % | 9.1 % | 6.4 % | 14.0 % | 6.5 % | 41.1 % | 1.1 % | 76.6 % | 1.6 % |
| SE | 72.4 % | 0.4 % | 22.6 % | 2.2 % | 40.5 % | 2.4 % | 18.4 % | 2.9 % | 29.1 % | 2.7 % | 55.2 % | 6.3 % | 100.0 % | |

**Table B3.** Overview of shareable data. The column "daily" indicates if the original daily data can be shared and "monthly" if the derived monthly data can be shared.

| Code | Country | Data provider | Daily | Monthly |
|---|---|---|---|---|
| AT_HZB | Austria | HZB | No | Yes |
| CH_METEOSWISS | Switzerland | MeteoSwiss | No | Yes |
| CH_SLF | Switzerland | SLF | No | Yes |
| DE_DWD | Germany | DWD | Yes | Yes |
| FR_METEOFRANCE | France | Météo-France | Yes | Yes |
| IT_BZ | Italy | Bolzano | Yes | Yes |
| IT_FVG | Italy | Friuli Venezia Giulia | Yes | Yes |
| IT_LOMBARDIA | Italy | Lombardia | Yes | Yes |
| IT_PIEMONTE | Italy | Piemonte | No | No |
| IT_SMI | Italy | SMI | No | No |
| IT_TN | Italy | Trentino | Yes | Yes |
| IT_TN_TUM | Italy | Trentino (TUM) | No | No |
| IT_VDA_AIBM | Italy | Valle D'Aosta (AIBM) | No | No |
| IT_VDA_CF | Italy | Valle D'Aosta (CF) | Yes | Yes |
| IT_VENETO | Italy | Veneto | No | Yes |
| SI_ARSO | Slovenia | ARSO | No | Yes |

**Appendix C:  Seasonal snow indices**

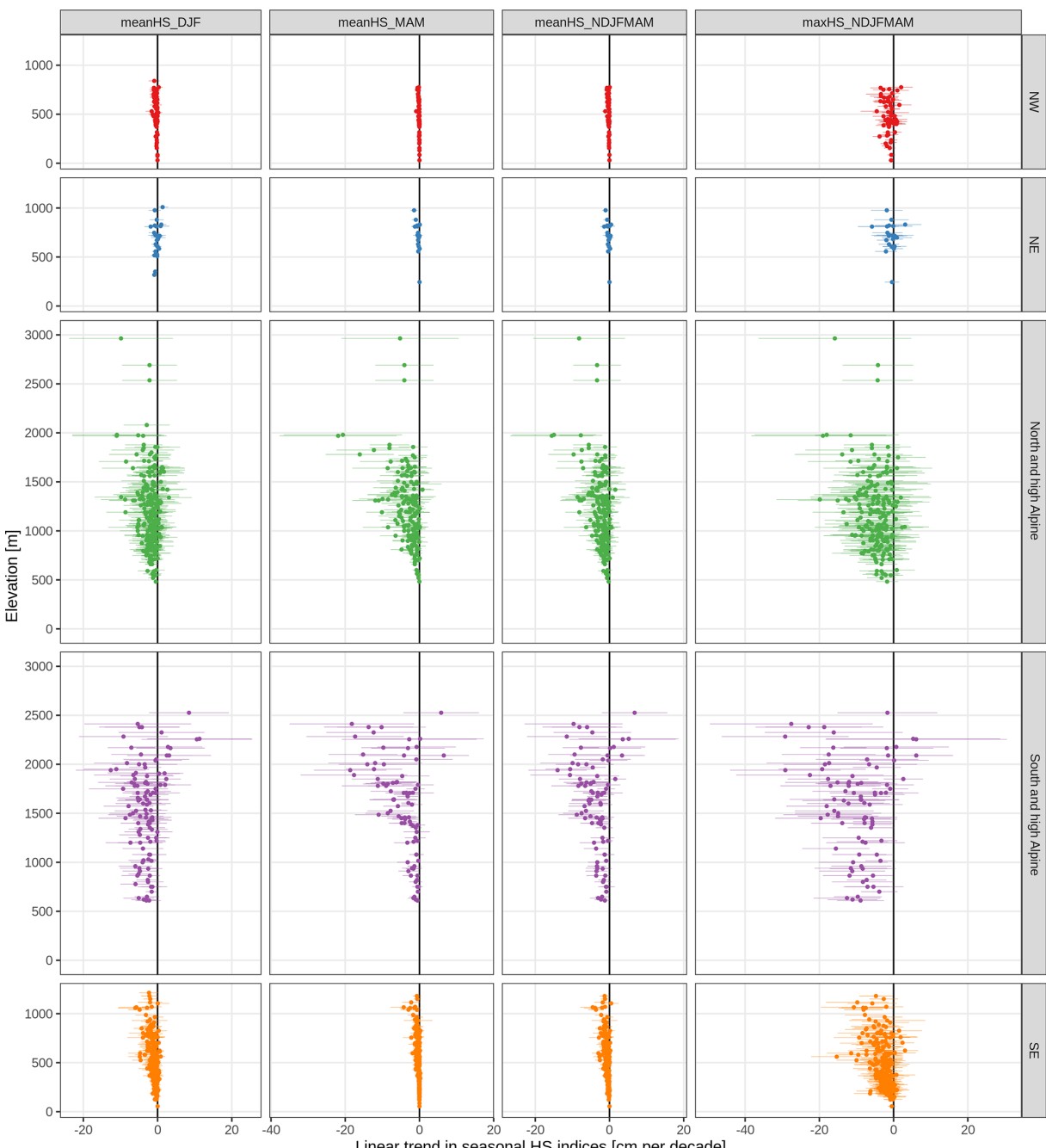

**Figure C1.** Long-term (1971 to 2019) linear trends in seasonal snow depth (HS) indices. Trends are shown separately by index (columns) and region (rows). The season is indicated in the columns with the first letter of the included months (e.g. DFJ is December, January, and February). Each point is one station. The points indicate the trend and the lines the associated 95 % confidence interval.

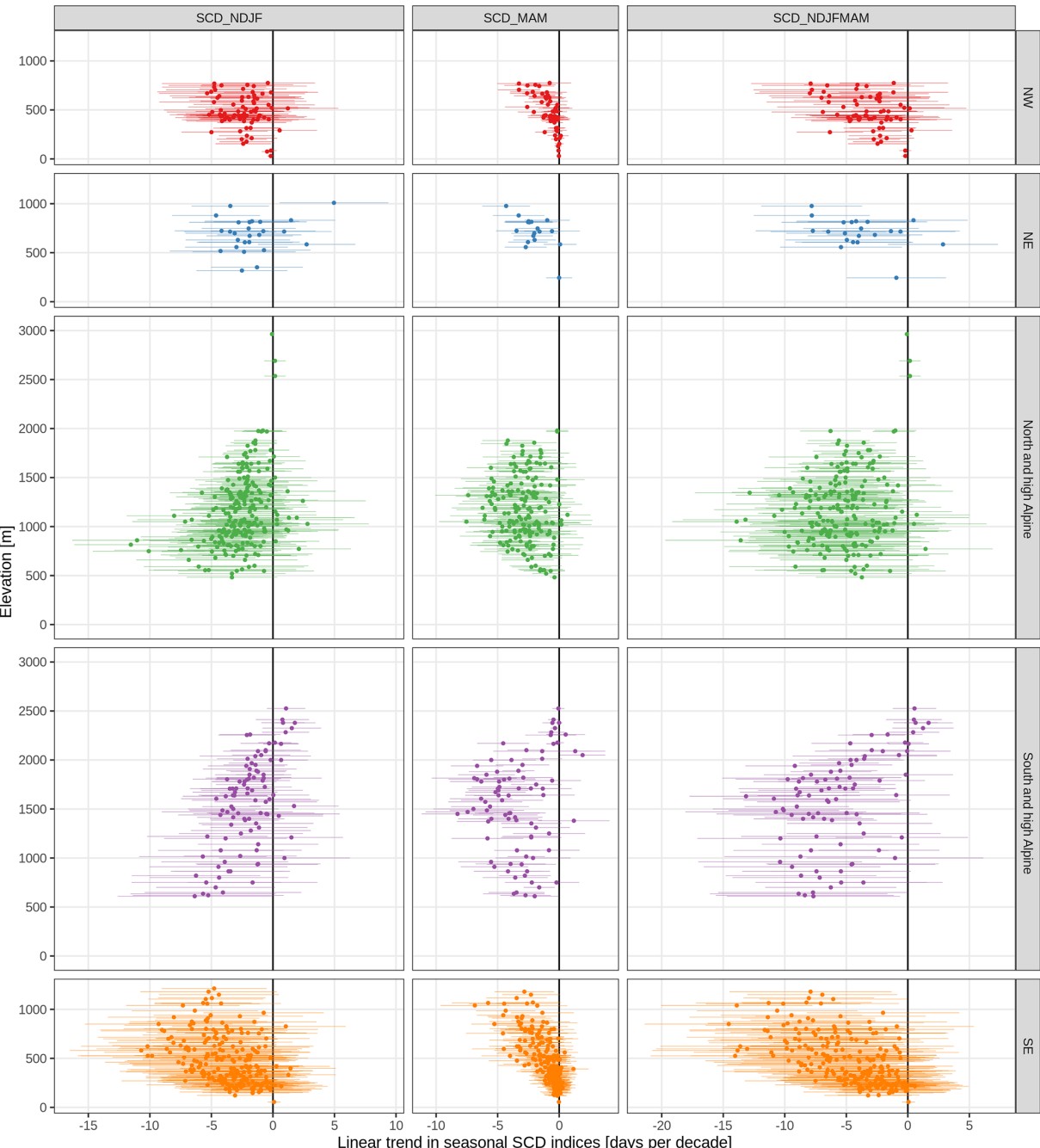

**Figure C2.** Long-term (1971 to 2019) linear trends in seasonal snow cover duration (SCD) indices. Trends are shown separately by index (columns) and region (rows). The season is indicated in the columns with the first letter of the included months (e.g. NDFJ is November, December, January, and February). Each point is one station. The points indicate the trend and the lines the associated 95 % confidence interval.

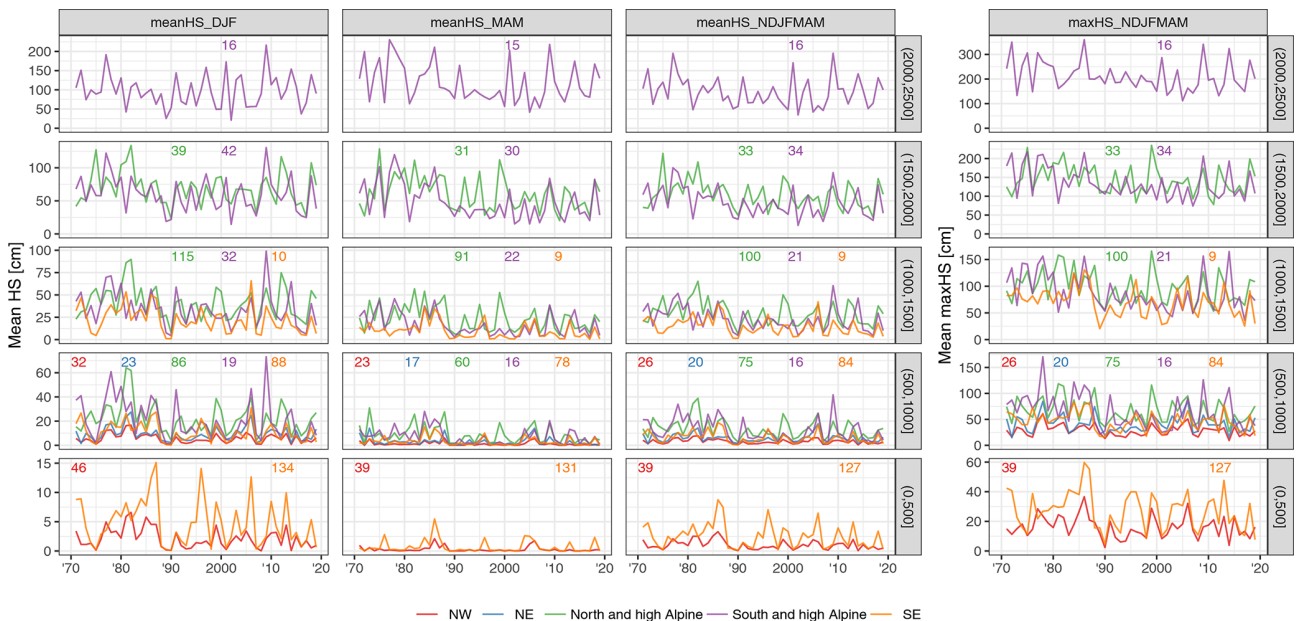

**Figure C3.** Time series of mean seasonal snow depth (HS) indices averaged by 500 m elevation bands. The rows indicate elevation band and the columns the index. The season is indicated in the columns with the first letter of the included months (e.g. DFJ is December, January, and February). The small numbers at the top of each panel denote the number of stations included in the average. Lines are only shown if more than five stations were available.

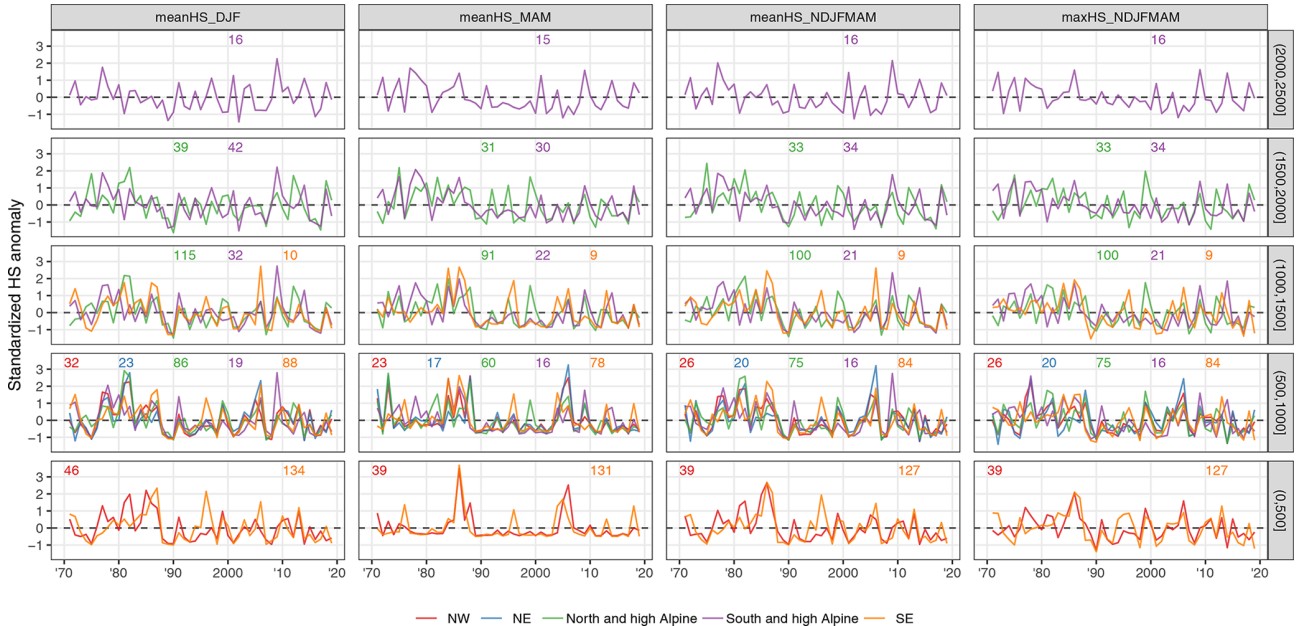

**Figure C4.** Same as Fig. C3 but for standardized anomalies.

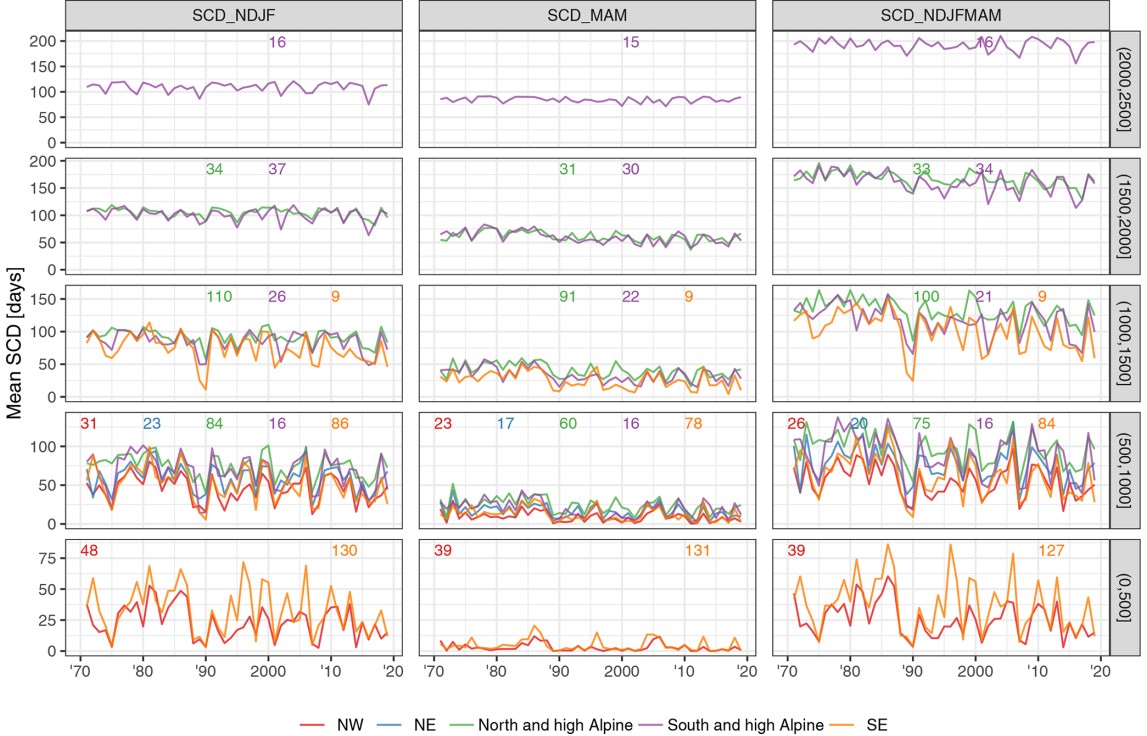

**Figure C5.** Time series of mean seasonal snow cover duration (SCD) indices averaged by 500 m elevation bands. The rows indicate elevation band and the columns the index. The season is indicated in the columns with the first letter of the included months (e.g. NDFJ is November, December, January, and February). The small numbers at the top of each panel denote the number of stations included in the average. Lines are only shown if more than five stations were available.

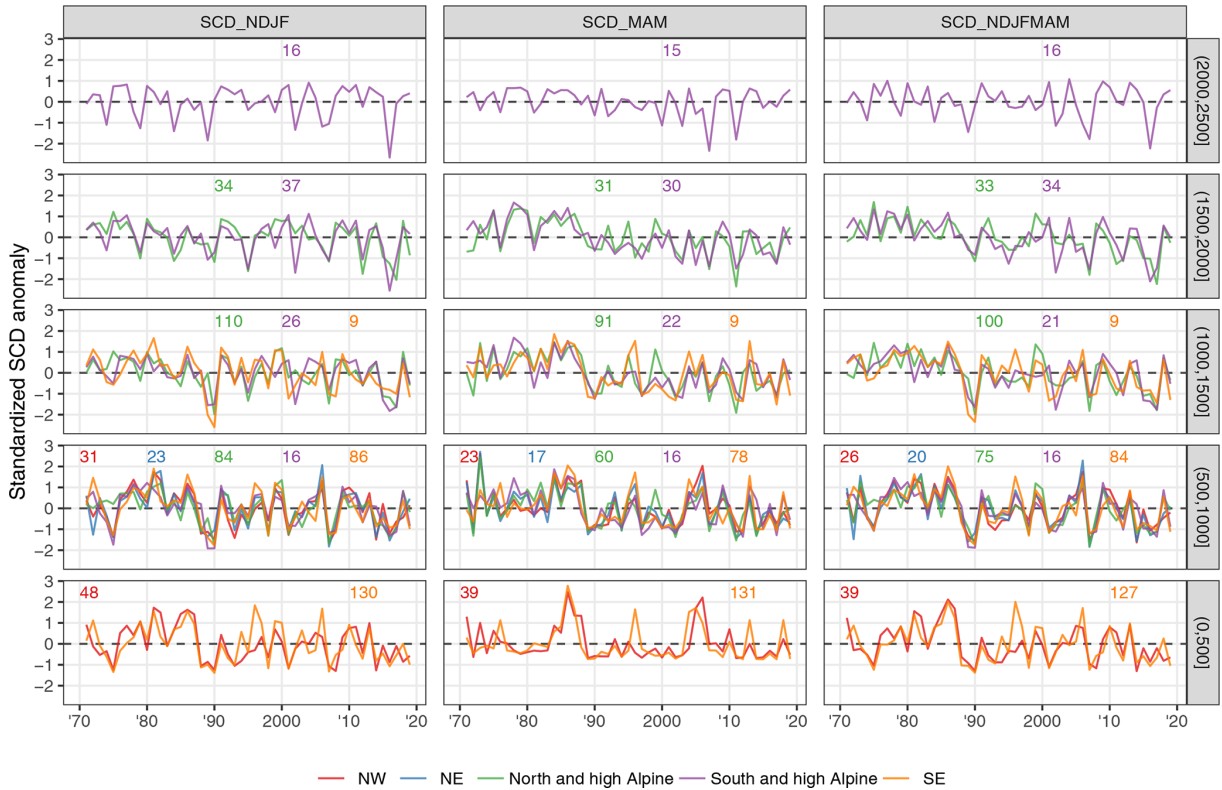

**Figure C6.** Same as Fig. C5 but for standardized anomalies.

**Table C1.** Overview of long-term (1971 to 2019) trends in mean seasonal snow depth indices. Summaries are shown by index, region, and 1000 m elevation bands (0 to 1000, 1000 to 2000, and 2000 to 3000 m). Cell values are the number of stations (#), the mean trend (mean, in cm decade$^{-1}$), and percentages of significant negative (sig−) and positive (sig+) trends; the remaining percentage (not shown) corresponds to the total of non-significant negative and positive trends. Empty cells denote no station available (for # and mean) and no stations with significant negative or positive trends (sig− and sig+). Trends were considered significant if $p < 0.05$. See also Fig. C1. A version of the table with 500 m bands instead of 1000 m is available in the Supplement (Table S6).

| Index | Region | Elevation: (0,1000] m | | | | Elevation: (1000,2000] m | | | | Elevation: (2000,3000] m | | | |
|---|---|---|---|---|---|---|---|---|---|---|---|---|---|
| | | # | mean | sig− | sig+ | # | mean | sig− | sig+ | # | mean | sig− | sig+ |
| meanHS_DJF | NW | 78 | −0.38 | 26.9 % | | | | | | | | | |
| | NE | 25 | −0.26 | 8.0 % | | 1 | 1.36 | | | | | | |
| | N&hA | 87 | −1.64 | 14.9 % | | 154 | −2.09 | 7.8 % | | 4 | −4.28 | | |
| | S&hA | 19 | −3.57 | 42.1 % | | 74 | −3.56 | 17.6 % | | 17 | −0.07 | 5.9 % | |
| | SE | 222 | −0.95 | 22.1 % | | 10 | −2.94 | 50.0 % | | | | | |
| meanHS_MAM | NW | 62 | −0.12 | 9.7 % | | | | | | | | | |
| | NE | 18 | −0.45 | 11.1 % | | | | | | | | | |
| | N&hA | 61 | −1.56 | 47.5 % | | 122 | −3.74 | 42.6 % | | 3 | −4.45 | | |
| | S&hA | 16 | −1.34 | 43.8 % | | 52 | −5.38 | 69.2 % | | 16 | −6.73 | 31.2 % | |
| | SE | 209 | −0.24 | 7.2 % | 0.5 % | 9 | −1.82 | 33.3 % | | | | | |
| meanHS_NDJFMAM | NW | 65 | −0.23 | 41.5 % | | | | | | | | | |
| | NE | 21 | −0.31 | 9.5 % | | | | | | | | | |
| | N&hA | 76 | −1.44 | 32.9 % | | 133 | −2.77 | 27.1 % | | 3 | −4.96 | | |
| | S&hA | 16 | −2.15 | 56.2 % | | 55 | −4.38 | 50.9 % | | 17 | −2.91 | 23.5 % | |
| | SE | 211 | −0.60 | 27.0 % | | 9 | −2.13 | 55.6 % | | | | | |
| maxHS_NDJFMAM | NW | 65 | −1.15 | 16.9 % | | | | | | | | | |
| | NE | 21 | −0.82 | 4.8 % | | | | | | | | | |
| | N&hA | 76 | −3.99 | 19.7 % | | 133 | −5.19 | 20.3 % | | 3 | −8.11 | | |
| | S&hA | 16 | −8.87 | 75.0 % | | 55 | −10.33 | 56.4 % | | 17 | −9.37 | 41.2 % | |
| | SE | 211 | −2.78 | 27.0 % | | 9 | −6.59 | 55.6 % | | | | | |

**Table C2.** Overview of long-term (1971 to 2019) trends in mean seasonal snow cover duration indices. Summaries are shown by index, region, and 1000 m elevation bands (0 to 1000, 1000 to 2000, and 2000 to 3000 m). Cell values are the number of stations (#), the mean trend (mean, in d decade$^{-1}$), and percentages of significant negative (sig−) and positive (sig+) trends; the remaining percentage (not shown) corresponds to the total of non-significant negative and positive trends. Empty cells denote no station available (for # and mean) and no stations with significant negative or positive trends (sig− and sig+). Trends were considered significant if $p < 0.05$. See also Fig. C2. A version of the table with 500 m bands instead of 1000 m is available in the Supplement (Table S7).

| Index | Region | Elevation: (0,1000] m | | | | Elevation: (1000,2000] m | | | | Elevation: (2000,3000] m | | | |
|---|---|---|---|---|---|---|---|---|---|---|---|---|---|
| | | # | mean | sig− | sig+ | # | mean | sig− | sig+ | # | mean | sig− | sig+ |
| SCD_NDJF | NW | 79 | −2.47 | 30.4 % | | | | | | | | | |
| | NE | 25 | −1.92 | 16.0 % | | 1 | 4.96 | | 100.0 % | | | | |
| | N&hA | 85 | −3.33 | 38.8 % | | 144 | −2.14 | 36.1 % | | 3 | 0.09 | | |
| | S&hA | 16 | −3.79 | 6.2 % | | 63 | −2.08 | 28.6 % | | 17 | −0.20 | | 5.9 % |
| | SE | 216 | −3.67 | 28.2 % | | 9 | −5.26 | 66.7 % | | | | | |
| SCD_MAM | NW | 62 | −0.82 | 22.6 % | | | | | | | | | |
| | NE | 18 | −2.05 | 66.7 % | | | | | | | | | |
| | N&hA | 61 | −2.56 | 59.0 % | | 122 | −3.03 | 66.4 % | | | | | |
| | S&hA | 16 | −3.06 | 75.0 % | | 52 | −4.16 | 78.8 % | | 16 | −0.60 | 18.8 % | 6.2 % |
| | SE | 208 | −0.99 | 20.7 % | | 9 | −3.58 | 66.7 % | | | | | |
| SCD_NDJFMAM | NW | 65 | −3.33 | 40.0 % | | | | | | | | | |
| | NE | 21 | −3.86 | 33.3 % | | | | | | | | | |
| | N&hA | 76 | −5.58 | 57.9 % | | 133 | −5.28 | 73.7 % | | 3 | 0.09 | | |
| | S&hA | 16 | −6.66 | 50.0 % | | 55 | −6.67 | 80.0 % | | 17 | −1.01 | 17.6 % | |
| | SE | 211 | −4.70 | 34.6 % | | 9 | −8.84 | 88.9 % | | | | | |

*Code and data availability.* All computations were performed with R statistical software version 4.0.2 (RCoreTeam, 2008). Colours for the figures were taken from scientific colour scales (Crameri, 2019) and colorBrewer. The code is available from a repository (https://doi.org/10.5281/zenodo.4064128, Matiu et al., 2020) which includes scripts for the following tasks: reading the different data sources, performing all data preprocessing, quality checking, gap filling, and statistical analyses.

Most data providers agreed to share their data; see Table B3 for the availability of daily and monthly values. For the full dataset, please contact the main authors (Michael Matiu or Alice Crespi); the usage is generally free for research purposes, although explicit consent is required from some data providers which want to keep track of the usage of the data. The shareable data are available from an open repository (https://doi.org/10.5281/zenodo.4064128, Matiu et al., 2020).

*Supplement.* The supplement related to this article is available online at: https://doi.org/10.5194/tc-15-1-2021-supplement.

*Author contributions.* Conceptualization: MM, AC TS14, GB, CM, SM, WS; Data curation: MM, AC, GB, CMC, CM, DCB, GC, MV, WB, PC, GM, SCS, AC, RC, AD, MF, MG, LM, JMS, AS, AT, SU, VW; Formal analysis: MM, AC; Funding acquisition: MM; Investigation: MM; Methodology: MM, AC, CM, SM, WS; Resources: MZ; Software: MM, AC, CMC; Supervision: MM, MZ; Validation: MM, AC; Visualization: MM; Writing – original draft preparation: MM, AC; Writing – review and editing: MM, AC, GB, CMC, CM, SM, WS, GC, LDG, SK, BM, GR, ST, MV, PC, IG, GM, CN, SCS, US, MW, LG. TS15

*Competing interests.* The authors declare that they have no conflict of interest.

*Acknowledgements.* We thank the reviewers (two anonymous and Ross Brown) for their comments which have greatly improved the paper.

This project has received funding from the European Union's Horizon 2020 research and innovation programme under the Marie Sklodowska-Curie grant agreement no. 795310. This work has benefited from funding from the European Union's Horizon 2020 research and innovation programme under grant agreement no. 730203. CNRM/CEN is a member of LabEX OSUG@2020. Gabriele Chiogna acknowledges the support from the Stiftungsfonds für Umweltökonomie und Nachhaltigkeit GmbH (SUN) and likewise the support from the DFG (Deutsche Forschungsgemeinschaft) research group FOR2793/1 "Sensitivity of High Alpine Geosystems to Climate Change since 1850 (SEHAG)" grant CH981/3.

We acknowledge the E-OBS dataset from the EU-FP6 project UERRA (https://www.uerra.eu TS16) and the Copernicus Climate Change Service and the data providers for the ECA&D project (https://www.ecad.eu TS17). For providing us with station data, we are grateful to Günther Geier from the meteorological office and avalanche warning centre CE13 from the province of Bolzano, to Sara Ratto from the Centro Funzionale della Regione Autonoma Valle d'Aosta, and to Gregor Vertačnik from the meteorological office of the Slovenian Environmental Agency.

*Financial support.* This research has been supported by the European Commission Horizon 2020 Framework Programme CliRSnow (grant no. 795310) and PROSNOW (grant no. 730203) and the Deutsche Forschungsgemeinschaft (grant no. CH981/3). TS18

*Review statement.* This paper was edited by Guillaume Chambon and reviewed by Ross Brown and two anonymous referees.

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

**Remarks from the language copy-editor**

CE1   Please verify the short summary. Do you mean the whole winter season or all seasons?
CE2   Please verify the name. Should there be an underscore in the name?
CE3   Please verify the word.
CE4   Please verify the abbreviation.
CE5   Please verify the table which has been adapted to our standards.
CE6   Do you mean Sect. 2?
CE7   Please verify that stn signifies station.
CE8   Please verify.
CE9   Please define.
CE10   Does visual need to be in brackets here and below?
CE11   Please define. Do you mean North Atlantic Oscillation?
CE12   Please define.
CE13   Please verify. Is this a centre, a group, or something else?

**Remarks from the typesetter**

TS1   Are these separate affiliations? In this case, they need to be separated. Please check.
TS2   Are these two different affiliations? In this case, they need to be separated. Please check.
TS3   The composition of Figs. 1–8, A1–A3, B1–B2, and C1–C5 has been adjusted to our standards. This also includes language adjustments to Figs. 4–8, B1–B2, and C1–C5.
TS4   Please check.
TS5   Please verify the year for this source. Is it the same as the ones above?
TS6   Please check.
TS7   Please check.
TS8   Please check.
TS9   Please check.
TS10   Please check.
TS11   Please check.
TS12   Please check.
TS13   Please check.
TS14   Please differentiate between AlC for Alice Crespi and AnC for Andrea Cicogna throughout this section.
TS15   Please use complete sentences in this section.
TS16   Please provide date of last access.
TS17   Please provide date of last access.
TS18   Please note that the funding information has been added to this paper. Please check if it is correct. Please also double-check your acknowledgements to see whether repeated information can be removed or changed accordingly. Thanks.
TS19   Please submit name of publisher/publishing institution and place of publication.
TS20   Please provide last access date.
TS21   Please update.
TS22   Please update.
TS23   Please submit name of publisher/publishing institution and place of publication.
TS24   Please check additions.
TS25   Please check journal name. Abbreviation not found.
TS26   Please provide article number or page range.
TS27   Please check journal name. Abbreviation not found.
TS28   Please provide page range or article number and DOI.
TS29   Please provide last access date.
TS30   Please provide last access date.
TS31   Please provide page range or article number and DOI.
TS32   Please check journal name. Abbreviation not found.
TS33   Please check journal name. Abbreviation not found.
TS34   Please provide page range or article number and DOI.

TS35   Please provide last access date.