# Peer review of "Observed snow depth trends in the European Alps 1971 to 2019"

_The Cryosphere, 2020_

## Referee Comment (RC1) · Anonymous Referee #1 · 29 Oct 2020

An extensive set of snow depth measurements from the European Alps is evaluated to explore snow climatology regions and temporal trends. Raw data is collected from various sources and subsequently harmonized, quality controlled and temporal gaps were filled. Based on the snow depth time series, five regions with different snow climates are derived using Principal Component Analysis and k-means clustering. Linear long-term trends and short-term trend variability are computed from the snow depth time series for the five derived regions. Finally, snow depth observations are compared with gridded air temperature and precipitation data, which is either reanalysis based or inferred from spatially interpolating observational data. The authors find decreasing trends of snow depth for the majority of the stations and substantial differences in trends between regions.

This study addressed a very relevant topic: the evolution of snow depth in the European Alps during the last half century. The authors quantify these changes based on an extensive compilation of in-situ time series of snow depth from different regions. Additionally, the authors provide the unified dataset as an online resource, which can be very useful for further applications. The manuscript is well written and applied data and methods are sufficiently explained. My comments concern therefore primarily smaller ambiguities and (potential) errors.

**General/major comments**

- Some sentences are extremely long and should be shortened to increase readability.
- **Selection of different time periods**
  I'm confused about the different time periods that were used for this study: The PCA and k-means clustering was performed for 1981 – 2010, long-term trends were computed for 1971 – 2019 and short-term trends for 1961 – 2019. Could you elaborate in more detail why you use (these) three periods?
- **Which data (raw, gap-filled) was used for which analysis/plot?**
  In the manuscript, I was sometimes a bit confused which data was used for which application. E.g. in section 2.4 you use the raw data without any gap filling, right? But for the subsequent analyses you always use the gap-filled data (as explained in appendix section A.3)? This point should be more explicitly stated in my opinion.
- **Gap filling method**
  I struggled to follow the explanation of the gap filling method – particularly from line 641 to 654:
  - I do not understand what "crossing a calendar day window with a year window" means.
  - Shouldn't the "window of 31 days" be "window of gap length + 30 days"?
  - "mean of the daily values" → does this refer to the daily climatology?
  - "and the weights were based on the vertical distance between candidate and reference station." → why are horizonal distances not considered?
  - Finally, I wondered if reconstructed values have a smaller temporal variability (on smaller scales) because you apply climatological values in your method. But this is not the case, right? Because you only compute the ratios from climatological values (daily means)?

**Point comments**

**L150-151:** I don't understand this sentence – could you rephrase it?
**L157-158:** this sentence is a bit odd: why do three different climate forcing zones create four main climate regions?
**L179-189:** I struggled to read this sentence (because it is so long). To increase readability, the providers from Italy could e.g. be listed with bullet points.
**L195-196:** "with a few expectations of monthly/seasonal data form the HZB and SMI." → was the monthly/seasonal data also used in this study?
**L196-199:** I'm confused by this sentence: automatic measurements are used both from France and the Aosta Valley, right? Then I would change "only for France or" to "only for France and". Anyway, I think the sentence should be rephrased to improve comprehensibility.
**L205:** "(see also Fig. 2b)" should be "(see also Fig. 2c)", right?
**L245:** why is the data for Austria only available until 2016?
**L253:** "The frequency by elevation (50 m bins were used to calculate to proportion)…" → I don't understand this part.
**L257-258:** why is this criterion applied? Wouldn't e.g. a threshold of 50% make more sense?
**L273:** did you consider to use the "elbow method" to find the optimal number of clusters?
**L273-274:** what do you mean by "as well as clustering directly on the daily observations."?
**L287-288:** remove "using assessed"
**L337:** "and is highly correlated to elevation up to 1000-1500 m, and mostly constant above." → I don't understand this part.
**L348:** the clustering was performed on the 5-dimensional PCA loadings, right?

**L367-368:** please provide a reference here for the HISTALP subregions

**L370:** what you do mean by "estimated data-driven"?

**L377:** what do you mean by "unique stations"?

**L386:** I would not write "matches" here because there are 4 vs. 5 regions. I would rather write that the obtained regions are similar.

**L404:** does this statement refer to a specific month? Or the entire winter?

**L408-409:** "mean North & high Alpine" is odd. Do you mean: "While in December, the mean negative trend was stronger in North & high Alpine"?

**L422:** "Points with lines indicate" → "Lines indicate" (or are there points with no lines?)

**L467-468:** "Moreover, we assume that most of this seasonal imbalance is because there is no or no significant snow cover in that month" → I don't understand this part

**L493:** I guess "~100m" should be "~100 mm"

**L515-516:** "because this implies less chances that precipitation falls at the "right" time." → I don't understand this part

**L532-533:** I'm not sure if I understand this sentence correctly. Do you mean homogenization is not so important because such a large number of stations is used?

**L589:** how do you define a network? A country?

**L612:** how is this surrounding band defined in terms of horizontal distance?

**L673-674:** I don't' understand this sentence.

**L686:** what is meant by "ablation scheme of the different stations"?

**L701-702:** "gap filling snow depth series using simulations of the Crocus snow model for the French Alps" → I'm a bit confused by this part. Does it state that gap filling was performed by running the Crocus snow model with meteorological forcing?

**L721:** what is meant by "original observations"? Available observations?

**Figure 1b:** this panel is hard to read. Could you enlarge it? To increase readability, station density could be plotted instead (i.e. the number of stations per a certain area).

**Figure 2b:** how was the polygon for the DEM generated? With a convex hull?

**Table 2:** there are typos in the first row (e.g. "(0,1000] m").

**Table 3:** the spacing between the columns should be improved (→ it is currently confusing that the columns "DJF #" and "MAM mean (min, max)" are so closed together)

**Figure A1:** I guess a subset of stations was used to produce this figure, right?

**Figure A2:** which statistical quantities (percentiles, outliers, etc.) do the points, lines and box edges represent?

**Figure B4:** how is the numerical quantity "silhouette width" computed?

**Figure C4:** For completeness, the table for MESCAN-SURFEX (March to May) should also be shown

**Stylistic comments and typos**

**L120:** "1960–1990. (Lejeune et al., 2019)." → "1960–1990 (Lejeune et al., 2019)."

**L144:** "while Section 4 concludes." → "while Section 4 convers conclusions."

**L147:** "with their typical arc-shaped" → "with their arc-shaped"

**L231-232:** change to "Station numbers are shown for fresh snow (HN) and snow depth (HS) time series."

**L279-280:** I would remove the line break here.

**L312:** "significantly to the" → "significantly from the"

**L348:** "There were" → "This yielded"

**L353-354:** "South of the main ridge, there were two regions:" → "Two regions emerge south of the main ridge:"

**L362:** "as has in the north" → "analogue to the north"

**L376:** "the station in common, and the same common stations" → "the stations, and the same stations"

**L400:** I would rather use present tense here (and in the following lines).

**L469:** remove "supposed to be"

**L561:** there seems to be a space in the word "scientific"

**L661:** "were useful" → "are justified"

**L664:** "has not been yet used" → "has not yet been used"

---

## Referee Comment (RC2) · Ross Brown (Referee) · 2 Nov 2020

General comments: This paper provides an analysis of snow cover regional variability and trends over the European Alps based on a new in situ daily snow depth dataset developed through the collaboration of more than 20 institutions from six countries. The dataset covers the entire European Alps with more than 2000 surface snow depth observations, and represents an important contribution for research and development. The authors are to be congratulated for their efforts to develop this dataset and in particular, to make it available to the research community. The creation of a pan-Italian snow depth dataset from various agencies is a noteworthy achievement.

The paper presents the results of a PC and cluster analysis to characterize the re-

gional snow climate, along with trend analysis to document trends by climate region and elevation over almost 50 years (1971-2019). The paper is in general well-written and clearly explained, and is close to publishable quality once some issues with overly-long sentence construction are rectified. I have three main comments concerning the methodology. First (comment #6 below), I question the need for the moving window analysis for trend variability, and recommend it be removed from the paper. Second (comment #10), the PC results reflect an uneven spatial distribution of stations with oversampling of elevations below 1000 m and undersampling of elevations above 2000 m. It is unclear to what extent this distorts the analysis results compared to those obtained based on a gridded representation of the station data that evenly samples the full spatial and elevation domain. Third, the paper provides no insights into interannual variability of snow cover and its relative magnitude compared to the long-term trend. The authors may feel this is beyond the scope of the current paper, but presenting trends without discussing the interannual and multi-decadal variability is a major oversight in my opinion.

I look forward to seeing the revised paper and congratulate the authors again for their significant contribution.

Ross D. Brown, Canada (ross.brown@canada.ca)

Detailed comments: 1. Lines 74-75: Suggest rewording as "The main limitation .... that their number decreases sharply with elevation, with few stations available above 3000 m in the European Alps."

2. Lines 90-126: There is a lot of useful material presented here on published snow cover trends in the various countries, but it is difficult to read with very long sentences joined with rather unwieldy constructions like "which, however". I recommend you organize this material in a summary table, and provide a few lines that capture the common elements. This would lead very nicely into the paragraph starting on line 128.

3. Lines 155-160: It would be instructive to show the main climatic divides on Figure 1.

[Figure]

4. Lines 204-206: consider rewording as "Many stations contain an important data gap for the 1981–1997 period that rendered a large fraction of the stations unusable for this study."

5. Line 279: Suggest deleting the following "The predicted variable was the mean monthly HS and the only predictor the year (shifted to 0)" and replacing the previous sentence with "Linear trends in monthly mean HS were computed separately for each month from November to May for stations with complete data in the period."

6. Line 280: "The second approach was a moving window approach that aims at identifying the short-term changes in trends." I think it would be clearer to say "A second moving window approach was used to examine the variability in 30-year trends over the period from 1961". I would consider removing this analysis for the following reasons: (1) the lack of a clear rationale for the analysis, (2) the inconsistency introduced by the different start period (1961 vs 1971), (3) the fact that overlapping windows are not independent, (4) the use of what is essentially an arbitrary 30-year period for the trend, and (5) the fact that the network density changes over time. I think it would be more instructive to look at the signal-to-noise properties of the 1971-2019 trend, by breaking it up into the amount of variance explained by the trend vs the amount of variance explained by interannual variability. Mapping the two quantities would highlight areas where trend was stronger relative to natural variability and vice versa.

7. Line 297: Not clear what you mean here… the homogeneity of the data used in a gridded dataset is the key issue. Several reanalyses have well documented discontinuities related to changes in data input streams.

8. Line 317: There is no season dedicated to the snow-cover onset period (Nov-Dec?), but there is one (March-May) for the spring season. Any reason for this? In my work documenting snow cover variability over Bulgaria (Brown and Petkova, 2007, Int. J. Climatol. 27, 1215–1229) and Quebec (Brown, 2010, Hydrol. Process., 24, 1929–1954) we found different trends in the fall and spring periods as well as different modes

of atmospheric variability influencing snow cover variability in each season.

9. Line 400: Can you please include the variable(s) the trend is computed for to remind the reader what the results refer to.

10. Lines 473-474: "In relative terms, the elevations of the stations used in this study oversample the elevations up to 1000 m, are similar from 1000 to 2000 m, significantly underrepresent 2000 to 3000 m, and do not cover elevations above 3000 m". This begs the question of why you did not attempt to transform the observations to an equal area grid (e.g. by kriging, or pseudo obs from modelling) to force the spatial coverage to be representative prior to the PC analysis?

11. Conclusions: This study provided useful new insights into snow-climate regions and trends for the European Alps, but did not look at interannual variability in snow cover e.g. PC analysis of annual time series of snow cover duration and maximum accumulation. Is there a particular reason why you chose to ignore this? Documenting and understanding interannual variability is a key component of interpreting long-term changes (e.g. the signal-to-noise ratio of climate heating induced changes).
* * *

---

## Referee Comment (RC3) · Anonymous Referee #3 · 17 Nov 2020

General comment. I was delighted to see this compilation and analysis of snow records from the whole span of the European alps. Previous country-based studies have used different methods that prevented aggregate conclusions, and the efforts the authors have undertaken to compile this comprehensive dataset represents an important breakthrough that paves the way for a much improved understanding of the consequences of warming for snow in the European alps. Having assembled three datasets (with more similarities than those here) from different jurisdictions for some of my work, I can appreciate the magnitude of the task.

Two referees have provided some technical corrections, to which I add the following.

Abstract - lines 49-51 are an attempt to represent much of the information in table 3 in a line of text, but the result is insufficiently specified and confusing. I suggest reducing the

amount of detail and focusing on the key numerical message, and delivering it clearly. Perhaps one number for the DJF all-station average and one for the MAM all-station average. The next level of detail would be to list the average trends by elevation bands, but it's less confusing to put the elevation band first: "for 0-1000m, -1.1cm/decade; for 1000-200m, …" Including the ranges is too much detail for an abstract, and places undue emphasis on outliers.

IPCC 2019 - follow the citation convention specified at the beginning of the report

There almost seems to be a straight line through the loadings of PC2-5 (Fig 3) at about 47.5°N, straighter than the topography would suggest. It's suspiciously close to the Germany-Austria border. Can you convince me that it's not a data artefact?

Fig B1 is very important for the interpretation of the loadings; I strongly suggest moving it to the main paper

Line 401: state the p-value of significance

Section 3.3 - I see no real reason to shorten the record and present 30-year trends, except to calibrate the variability of shorter trends. I see another reviewer provided extensive comments on this.

---

## Author Comment (AC1) · 14 Dec 2020

An extensive set of snow depth measurements from the European Alps is evaluated to explore snow climatology regions and temporal trends. Raw data is collected from various sources and subsequently harmonized, quality controlled and temporal gaps were filled. Based on the snow depth time series, five regions with different snow climates are derived using Principal Component Analysis and k-means clustering. Linear long-term trends and short-term trend variability are computed from the snow depth time series for the five derived regions. Finally, snow depth observations are compared with gridded air temperature and precipitation data, which is either reanalysis based or inferred from spatially interpolating observational data. The authors find decreasing trends of snow depth for the majority of the stations and substantial differences in trends between regions.

This study addressed a very relevant topic: the evolution of snow depth in the European Alps during the last half century. The authors quantify these changes based on an extensive compilation of in-situ time series of snow depth from different regions. Additionally, the authors provide the unified dataset as an online resource, which can be very useful for further applications. The manuscript is well written and applied data and methods are sufficiently explained. My comments concern therefore primarily smaller ambiguities and (potential) errors.

We thank you for the detailed review of our manuscript, the appreciation of our work, and the constructive comments, which we shall address in a revised version.

**General/major comments**

• Some sentences are extremely long and should be shortened to increase readability.

We shortened sentences as much as possible throughout the manuscript.

• **Selection of different time periods**

I'm confused about the different time periods that were used for this study: The PCA and k-means clustering was performed for 1981 – 2010, long-term trends were computed for 1971 – 2019 and shortterm trends for 1961 – 2019. Could you elaborate in more detail why you use (these) three periods?

We chose different periods because of the different nature of the analysis. The PCA and k-means aimed at having the largest spatial density, so we chose the 30-year period with the largest number of stations. The long-term trend analysis aimed at a tradeoff between coverage of stations and as long as possible period. The moving-window short-term trend analysis will be removed in the revision (see responses to Ross Brown's review comments). We shall add an explanation also in the manuscript and provide more information in the data overview sections on these two periods and station subsets.

• **Which data (raw, gap-filled) was used for which analysis/plot?**

In the manuscript, I was sometimes a bit confused which data was used for which application. E.g. in section 2.4 you use the raw data without any gap filling, right? But for the subsequent analyses you always use the gap-filled data (as explained in appendix section A.3)? This point should be more explicitly stated in my opinion.

Thanks for pointing out this ambiguity. Actually, we used the gap filled data for all analyses. We now explicitly state this in all related method sections (overview as well as statistical analyses).

• **Gap filling method**

I struggled to follow the explanation of the gap filling method – particularly from line 641 to 654:

o  I do not understand what "crossing a calendar day window with a year window" means.

o  Shouldn't the "window of 31 days" be "window of gap length + 30 days"?

o  "mean of the daily values" -> does this refer to the daily climatology?

o  "and the weights were based on the vertical distance between candidate and reference station." àwhy are horizonal distances not considered?

o  Finally, I wondered if reconstructed values have a smaller temporal variability (on smaller scales) because you apply climatological values in your method. But this is not the case, right? Because you only compute the ratios from climatological values (daily means)?

We rewrote the description of the gap filling in an algorithmic way and also provide an additional explanatory figure. We hope that this helps to clarify our procedure. Regarding your specific comments:

- The horizontal distance was only considered for selecting candidate stations and not for the weighting, because we wanted to have univariate weights and not multivariate. In that case, we found the vertical distance to be more important than the horizontal distance.
- The temporal variability of the reconstructed series should not be affected. The climatological values are only used to calculate the ratios. The reconstructed values is then based on the daily value(s) of the neighbouring series. Consequently, the daily variability in the reconstructed series stems from the daily variability in the reference series'. The only loss of variability could occur because the reconstructed value is an average of up to 5 values from up to 5 reference stations.

**Point comments**

**L150-151:** I don't understand this sentence – could you rephrase it?

Done.

**L157-158:** this sentence is a bit odd: why do three different climate forcing zones create four main climate regions?

Yes, it is counterintuitive. However, the three forcings combined with the topographic effects result in gradients along the North-South and East-West directions. And these two gradients, if intersected, result in four regions. We modified the wording accordingly.

**L179-189:** I struggled to read this sentence (because it is so long). To increase readability, the providers from Italy could e.g. be listed with bullet points.

As suggested, we restructured the data providers of Italy as bullet points.

**L195-196:** "with a few expectations of monthly/seasonal data form the HZB and SMI." -> was the monthly/seasonal data also used in this study?

Actually, no. We removed this part of the sentence, to avoid misunderstandings.

**L196-199:** I'm confused by this sentence: automatic measurements are used both from France and the Aosta Valley, right? Then I would change "only for France or" to "only for France and". Anyway, I think the sentence should be rephrased to improve comprehensibility.

The sentence was split and rephrased.

**L205:** "(see also Fig. 2b)" should be "(see also Fig. 2c)", right?

True. Thanks for spotting the error.

**L245:** why is the data for Austria only available until 2016?

Because of the processing and quality checking performed by the Austrian Hydrographical Service, it takes some time until the data are published online (we added this information also in the revised manuscript). In our case, when we accessed the data (early 2020), only data until 2016 were available. Since a few months, the records have been updated to include 2017, however, we cannot manage to re-analyze this update in the revision and, furthermore, results are not expected to change significantly including a one year extension of the Austrian series.

**L253:** "The frequency by elevation (50 m bins were used to calculate to proportion)…" -> I don't understand this part.

We plotted so-called frequency polygons, which are basically histograms shown as lines and not bars. This makes it easier to compare different distributions. But, as for histograms, it is necessary to specify the bin width. We tried to improve the caption to make it more understandable.

**L257-258:** why is this criterion applied? Wouldn't e.g. a threshold of 50% make more sense?

Yes, a different threshold would also make sense. However, then, also the meaning of the figure would be altered. We chose this simple threshold to show availability of stations per year. Another option would be to show stations with the threshold you proposed, but then it also depends on which season the 50% apply (Dec-Feb, Nov-May, or Oct-Sep). Yet another option would have been to show only the stations used in the analysis. We decided to stick to our simple threshold, and tried to clarify the intent.

**L273:** did you consider to use the "elbow method" to find the optimal number of clusters?

We shall look at the elbow method for the revision. We also compared it to the average silhouette coefficients, which we used initially. The elbow method identifies 4 clusters as optimal if we apply the k-means on the scaled observations, and 5 clusters if we apply it on the PC matrix. The silhouette analysis identifies 4 clusters as optimal, if applied on the scaled observations. If applied on the PC matrix, then 2 clusters are optimal, followed closely by 5 clusters. We checked maps of the clustering results for all combinations (see figure below; we also shall add it in the supplementary material), and, to be honest, all choices of clusters make "sense". The different number of clusters and whether observations or PC results are the input, all highlight different aspects of the snow depths and their hierarchy (e.g. as seen by increasing the number of clusters). These are the elevation, North-South gradient, and East-West gradient. We finally decided to leave our analysis as it was with 5 clusters, because they agree best with our knowledge of the climatic and topographical drivers of snow depth in the Alps. This information on this process shall also be added in the revised methods section.

[Figure]

Figure R1: Results of k-means clustering. Rows show the number of clusters (order is arbitrary, so colors might not match within a row). Columns identify the input matrix for the clustering algorithm: For Obs scaled daily observations of snow depth were used and for PCA the PCA matrix. The two PCA columns stand for the standard PCA that does not allow missing values (no NA), which corresponds to the same station set as in Obs, and the second PCA column is for the modified PCA algorithm that allows missing values (with NA) and has a higher station coverage.

**L273-274:** what do you mean by "as well as clustering directly on the daily observations."?

We applied the k-means clustering on the PCA matrix with the principal components as input, and, as comparison, we also applied the k-means clustering directly on a (scaled) matrix of daily snow depth observation. We tried to make this clearer in the manuscript.

**L287-288:** remove "using assessed"

Done.

**L337:** "and is highly correlated to elevation up to 1000-1500 m, and mostly constant above." I don't understand this part.

What we meant is that the PC2 is related to elevations up to 1000-1500m. We removed the word "correlated" and rewrote the sentences to make it clearer that PC1 and PC2 are both elevation driven, but PC1 mostly >1000m and PC2 mostly <1000m..

**L348:** the clustering was performed on the 5-dimensional PCA loadings, right?

Yes. We rewrote the sentence to make it clearer.

**L367-368:** please provide a reference here for the HISTALP subregions

Done.

**L370:** what you do mean by "estimated data-driven"?

We meant that the clustering was performed automatically using snow depth data, and no manual re-assignment or modification was done on the clustering results. We modified the sentence in the manuscript accordingly.

**L377:** what do you mean by "unique stations"?

Unique in the sense that they have no similar neighbours in any cluster. We now added this information in the manuscript.

**L386:** I would not write "matches" here because there are 4 vs. 5 regions. I would rather write that the obtained regions are similar.

Rewrote according to your suggestion.

**L404:** does this statement refer to a specific month? Or the entire winter?

To the whole winter. Clarified it also in the manuscript.

**L408-409:** "mean North & high Alpine" is odd. Do you mean: "While in December, the mean negative trend was stronger in North & high Alpine"?

Thanks for pointing this out. We rewrote the sentence.

**L422:** "Points with lines indicate" -> "Lines indicate" (or are there points with no lines?)

No, you are right, all points have lines, even though they might be hardly visible. We rewrote the legend.

**L467-468:** "Moreover, we assume that most of this seasonal imbalance is because there is no or no significant snow cover in that month" I don't understand this part

We meant the seasonal imbalance of station observations, since some low elevation stations do not record outside the winter season. We rewrote the sentence accordingly.

**L493:** I guess "~100m" should be "~100 mm"

Yes, thank you. We also changed it to the $\approx$ `sign.`

**L515-516:** "because this implies less chances that precipitation falls at the "right" time." I don't understand this part

We replaced "right time" with "concurrent with low temperatures".

**L532-533:** I'm not sure if I understand this sentence correctly. Do you mean homogenization is not so important because such a large number of stations is used?

Partly. Homogenization is important. But given the extent of our dataset it was impossible to apply a common framework to all the data. We made this clearer in the manuscript.

**L589:** how do you define a network? A country?

By data source i.e. data provider. Clarified it in the manuscript.

**L612:** how is this surrounding band defined in terms of horizontal distance?

Horizontal distance was not considered here. We added this in the manuscript.

**L673-674:** I don't' understand this sentence.

We assume you refer to the explanation why our relative MAE is not a "true" MAE. In the standard way, the relative MAE is defined as $\frac{1}{n}\sum_{i=1}^{n} \left|\frac{y_i - x_i}{x_i}\right|$, while our modification is $\frac{1}{n}\sum_{i=1}^{n} |y_i - x_i|/|\underline{x}|$, where $\underline{x}$ is the average of all $x_i$. We added the formulas also in the manuscript to make it clearer.

**L686:** what is meant by "ablation scheme of the different stations"?

The local climatic and topographic characteristics that influence ablation. We modified the sentence accordingly.

**L701-702:** "gap filling snow depth series using simulations of the Crocus snow model for the French Alps" -> I'm a bit confused by this part. Does it state that gap filling was performed by running the Crocus snow model with meteorological forcing?

Yes. Actually, the Crocus simulations were performed independently of the gap-filling used for this study, but we found it interesting to compare the two approaches, since they were both available at the same sites. We added some more information on this in the manuscript.

**L721:** what is meant by "original observations"? Available observations?

The observations available before gap filling. Modified the sentence to make it clearer.

**Figure 1b:** this panel is hard to read. Could you enlarge it? To increase readability, station density could be plotted instead (i.e. the number of stations per a certain area).

We splitted the figure in sub panels (for available and used), and put the station density underneath the points for a 0.5*0.25 deg grid. We still show points, because we think they are important.

**Figure 2b:** how was the polygon for the DEM generated? With a convex hull?

No, a manual outlining along the stations, because a convex hull would include most of the Po valley in Italy, for which we do not have any stations.. We clarified this in the caption.

**Table 2:** there are typos in the first row (e.g. "(0,1000] m").

We modified the elevation intervals to say "Elevation: (0,1000] m" etc.

**Table 3:** the spacing between the columns should be improved (-> it is currently confusing that the columns "DJF #" and "MAM mean (min, max)" are so closed together)

We aligned all numeric columns to the right, so the spacing should be better now.

**Figure A1:** I guess a subset of stations was used to produce this figure, right?

Yes, it is explained in the text, and we added this information to the figure caption.

**Figure A2:** which statistical quantities (percentiles, outliers, etc.) do the points, lines and box edges represent?

We added this information in the figure caption.

**Figure B4:** how is the numerical quantity "silhouette width" computed?

With silhouette width we mean the silhouette value or coefficients, sometimes also called width. We modified the figure and provide the formulas for how the silhouette is calculated in the methods section.

**Figure C4:** For completeness, the table for MESCAN-SURFEX (March to May) should also be shown

Added.

**Stylistic comments and typos**

**L120:** "1960–1990. (Lejeune et al., 2019)." -> "1960–1990 (Lejeune et al., 2019)."

Modified.

**L144:** "while Section 4 concludes." -> "while Section 4 convers conclusions."

Modified to "Section 4 provides conclusions".

**L147:** "with their typical arc-shaped" -> "with their arc-shaped"

Modified.

**L231-232:** change to "Station numbers are shown for fresh snow (HN) and snow depth (HS) time series."

Modified.

**L279-280:** I would remove the line break here.

Thanks, but since we removed the moving window analysis in the revision, this comment resolved itself.

**L312:** "significantly to the" -> "significantly from the"

Modified.

**L348:** "There were" -> "This yielded"

Modified.

**L353-354:** "South of the main ridge, there were two regions:" -> "Two regions emerge south of the main ridge:"

Modified.

**L362:** "as has in the north" -> "analogue to the north"

Modified.

**L376:** "the station in common, and the same common stations" -> "the stations, and the same stations"

Modified.

**L400:** I would rather use present tense here (and in the following lines).

We noticed our inconsistent use of present and past tense in the results. We mostly prefer past tense for results, and adapt the complete results section accordingly.

**L469:** remove "supposed to be"

Modified.

**L561:** there seems to be a space in the word "scientific"

Yes, it looks like this in the PDF of the paper, but in the Word version everything is fine.

**L661:** "were useful" -> "are justified"

Modified.

**L664:** "has not been yet used" -> "has not yet been used

Modified.

---

## Author Comment (AC2) · 14 Dec 2020

General comments:

This paper provides an analysis of snow cover regional variability and trends over the European Alps based on a new in situ daily snow depth dataset developed through the collaboration of more than 20 institutions from six countries. The dataset covers the entire European Alps with more than 2000 surface snow depth observations, and represents an important contribution for research and development. The authors are to be congratulated for their efforts to develop this dataset and in particular, to make it available to the research community. The creation of a pan-Italian snow depth dataset from various agencies is a noteworthy achievement.

The paper presents the results of a PC and cluster analysis to characterize the regional snow climate, along with trend analysis to document trends by climate region and elevation over almost 50 years (1971-2019). The paper is in general well-written and clearly explained, and is close to publishable quality once some issues with overly-long sentence construction are rectified. I have three main comments concerning the methodology. First (comment #6 below), I question the need for the moving window analysis for trend variability, and recommend it be removed from the paper. Second (comment #10), the PC results reflect an uneven spatial distribution of stations with oversampling of elevations below 1000 m and undersampling of elevations above 2000 m. It is unclear to what extent this distorts the analysis results compared to those obtained based on a gridded representation of the station data that evenly samples the full spatial and elevation domain. Third, the paper provides no insights into interannual variability of snow cover and its relative magnitude compared to the long-term trend. The authors may feel this is beyond the scope of the current paper, but presenting trends without discussing the interannual and multi-decadal variability is a major oversight in my opinion.

I look forward to seeing the revised paper and congratulate the authors again for their significant contribution. Ross D. Brown, Canada (ross.brown@canada.ca)

We thank Ross D. Brown for the in-depth review of our manuscript, his positive appreciation of our work as well as the detailed and constructive suggestions for improvements, which have been very helpful in revising the manuscript. In order to address the 3 major comments, in the revised manuscript we plan to introduce the following modifications:

1) The moving window analysis will be removed.
2) The uneven distribution of stations across elevation is an issue. However, we think that interpolating the stations to a grid could introduce more uncertainty, given the complex topography in the European Alps. While it is certainly a good exercise, it goes beyond the scope of this paper. See also the detailed comments below for more on this issue
3) We shall add multiple analyses to show the interannual variability and how this is related to the trends. These include time series figures, ratios of the trend versus variability, and we also modified our statistical model to deal with the changes in variability across time. See the detailed answer below for more information.

Furthermore, we will shorten the sentences, which was suggested by another reviewer, too.

Detailed answers to the other comments can be found below.

Detailed comments:

1. Lines 74-75: Suggest rewording as "The main limitation … that their number decreases sharply with elevation, with few stations available above 3000 m in the European Alps."

Thank you. Done.

2. Lines 90-126: There is a lot of useful material presented here on published snow cover trends in the various countries, but it is difficult to read with very long sentences joined with rather unwieldy constructions like "which, however". I recommend you organize this material in a summary table, and provide a few lines that capture the common elements. This would lead very nicely into the paragraph starting on line 128.

Thanks for this very useful suggestion. We now provide this information in a summary table that will be included in the Appendix, given the length of the table itself (2-3 pages). In the introduction we will summarize the main elements, as suggested.

3. Lines 155-160: It would be instructive to show the main climatic divides on Figure 1.

Done.

4. Lines 204-206: consider rewording as "Many stations contain an important data gap for the 1981–1997 period that rendered a large fraction of the stations unusable for this study."

Done.

5. Line 279: Suggest deleting the following "The predicted variable was the mean monthly HS and the only predictor the year (shifted to 0)" and replacing the previous sentence with "Linear trends in monthly mean HS were computed separately for each month from November to May for stations with complete data in the period."

Done.

6. Line 280: "The second approach was a moving window approach that aims at identifying the short-term changes in trends." I think it would be clearer to say "A second moving window approach was used to examine the variability in 30-year trends over the period from 1961". I would consider removing this analysis for the following reasons: (1) the lack of a clear rationale for the analysis, (2) the inconsistency introduced by the different start period (1961 vs 1971), (3) the fact that overlapping windows are not independent, (4) the use of what is essentially an arbitrary 30-year period for the trend, and (5) the fact that the network density changes over time. I think it would be more instructive to look at the signal-to-noise properties of the 1971-2019 trend, by breaking it up into the amount of variance explained by the trend vs the amount of variance explained by interannual variability. Mapping the two quantities would highlight areas where trend was stronger relative to natural variability and vice versa.

Thanks a lot, we clearly see your point and agree to a large extent. We therefore removed the moving window approach from the manuscript. Instead we introduced the analysis of interannual variability (see also your comment no. 11 below). This analysis presents the time series plots of the mean monthly HS (averaged over 500 m elevation bands because of the number of stations) and a short discussion of the fraction of variance explained by the trend.

Looking at the time series figures, we noticed some changes in the interannual variability across time. The most prominent is the decline in variability at the end of the season associated with the decrease in

mean HS, which approaches zero. We decided to account for these changes in the variability in our linear model by including a time coefficient for the error variance (=interannual variability). This results in replacing the standard OLS model with a GLS (generalized least squares) model. For our regression formula $y_t = \beta_0 + \beta_1 t + \epsilon_t$, OLS has a constant error variance $Var(\epsilon_t) = \sigma^2$, while with a GLS we can allow the error variance to depend on time: $Var(\epsilon_t) = \sigma^2 * exp(2 * \gamma * t)$, where $\gamma$ is another estimated coefficient that indicates the change in variance associated to $t$.

The trends in mean HS are not affected, but, with the new model structure, we were able to account for changes in the variance. More details on the model specifics can be found in the revised method section. We shall also discuss the results in the new section on interannual variability.

7. Line 297: Not clear what you mean here … the homogeneity of the data used in a gridded dataset is the key issue. Several reanalyses have well documented discontinuities related to changes in data input streams.

We completely agree. Actually we meant the spatial homogeneity, but this should be captured by the previous part of the sentence. So we removed this part of the sentence that evidently created a misinterpretation.

8. Line 317: There is no season dedicated to the snow-cover onset period (Nov-Dec?), but there is one (March-May) for the spring season. Any reason for this? In my work documenting snow cover variability over Bulgaria (Brown and Petkova, 2007, Int. J. Climatol. 27, 1215–1229) and Quebec (Brown, 2010, Hydrol. Process., 24, 1929–1954) we found different trends in the fall and spring periods as well as different modes of atmospheric variability influencing snow cover variability in each season.

Our idea was to use the single months as an alternative to seasonal aggregates. However, in the conclusion we aggregated by season (cf. your comment 11.), so this was not very consistent.

Furthermore, we decided to include analyses of mean seasonal values of HS, maximum HS, and SCD in the manuscript. However, we will place most of these in the appendix, because the paper is already quite long. The most important results were retained for the conclusion table.

9. Line 400: Can you please include the variable(s) the trend is computed for to remind the reader what the results refer to.

Good idea. We modified the sentence accordingly.

10. Lines 473-474: "In relative terms, the elevations of the stations used in this study oversample the elevations up to 1000 m, are similar from 1000 to 2000 m, significantly underrepresent 2000 to 3000 m, and do not cover elevations above 3000 m". This begs the question of why you did not attempt to transform the observations to an equal area grid (e.g. by kriging, or pseudo obs from modelling) to force the spatial coverage to be representative prior to the PC analysis?

This is a good point. But not a trivial issue. Because of the complex topography and strong elevational gradients in the Alps it is difficult to choose an appropriate resolution for the interpolation onto a grid. This would require a balanced number of stations both horizontally and vertically - a condition which is not met across the whole domain with our station set. While the transformation of the observations into an equal area grid would be interesting, we think it goes beyond the scope of the manuscript.

The main intent of this section is to give an overview of the confidence that can be expected from our assessments with respect to spatial coverage and elevation. We clarified it in the manuscript.

11. Conclusions: This study provided useful new insights into snow-climate regions and trends for the European Alps, but did not look at interannual variability in snow cover e.g. PC analysis of annual time series of snow cover duration and maximum accumulation. Is there a particular reason why you chose to ignore this? Documenting and understanding interannual variability is a key component of interpreting long-term changes (e.g. the signal-to-noise ratio of climate heating induced changes).

We followed your suggestion and included a section on interannual variability, as well as analyses on SCD and maximum HS; see also our replies to comments 6. and 8.

---

## Author Comment (AC3) · 14 Dec 2020

General comment. I was delighted to see this compilation and analysis of snow records from the whole span of the European alps. Previous country-based studies have used different methods that prevented aggregate conclusions, and the efforts the authors have undertaken to compile this comprehensive dataset represents an important breakthrough that paves the way for a much improved understanding of the consequences of warming for snow in the European alps. Having assembled three datasets (with more similarities than those here) from different jurisdictions for some of my work, I can appreciate the magnitude of the task.

Thank you for this positive assessment which we highly appreciate. Please find below a detailed account of our changes to the manuscript as well as responses to your remaining comments.

Two referees have provided some technical corrections, to which I add the following.

Abstract - lines 49-51 are an attempt to represent much of the information in table 3 in a line of text, but the result is insufficiently specified and confusing. I suggest reducing the amount of detail and focusing on the key numerical message, and delivering it clearly. Perhaps one number for the DJF all-station average and one for the MAM all-station average. The next level of detail would be to list the average trends by elevation bands, but it's less confusing to put the elevation band first: "for 0-1000m, -1.1cm/decade; for 1000-200m, …" Including the ranges is too much detail for an abstract, and places undue emphasis on outliers.

We adopted your suggestion to give all-station averages without ranges.

Regarding the elevational detail, we agree that we tried to condense too much information in too little space. We therefore remove numeric results from the abstract and now we only provide indicative remarks.

IPCC 2019 - follow the citation convention specified at the beginning of the report

Actually, we already took the citation as specified in the report. The IPCC 2019 citation refers to the SPM (summary for policymakers) part, and the citation has only been adapted to journal rules.

There almost seems to be a straight line through the loadings of PC2-5 (Fig 3) at about 47.5°N, straighter than the topography would suggest. It's suspiciously close to the Germany-Austria border. Can you convince me that it's not a data artefact?

Yes, this impression can arise. And it is possibly also strengthened by our choice of the color scale. But our analysis indicates that this is not a data artefact. The issue is rather that the border between Germany and Austria is tightly linked to a strong topographic divide. It is also the case e.g. for France-Italy, where also PC3-5 can give such an impression. We added country borders in the topography map Fig 1(a), to make it easier to see that the change is linked to topography rather than national borders.

Also, we created the PC figure splitted by country, in order to provide a clearer overview on the different data sources, and to better highlight the fact that the gradient is not a border effect (here only the subset for Switzerland, Germany and Austria for PC2):

**PC2**

[Figure]

We shall add the full figure with all PCs and countries in the supplement, and discuss this issue in the manuscript as well.

Fig B1 is very important for the interpretation of the loadings; I strongly suggest moving it to the main paper

Done.

Line 401: state the p-value of significance

Done.

Section 3.3 - I see no real reason to shorten the record and present 30-year trends, except to calibrate the variability of shorter trends. I see another reviewer provided extensive comments on this.

We remove this section from the analysis (see also the detailed information on the response to Ross D. Brown review comments).

---

## Referee Report (RR1)

I would like to thank the authors for the additional effort they put in the revision of the manuscript. Readability and completeness were much improved. I particularly like section A.3 which is now very detailed and comprehensible. I have only some minor additional comments (the page and line number refer to the latest manuscript version with track changes).

**Point comments**

L55: I find "after accounting for elevation" a bit odd. Do you mean trends of the same elevation bin differ amongst regions?

L145-146: I don't understand this sentence. Why is synthesizing studies into a common Alpine view relevant for providing snow cover information at the regional scale?

L331-332: I'm not sure if I understand this correctly: you tested the clustering with 2 – 8 PCs, right? It's a bit confusing because a couple of lines above, you write that you only retain the first five PCs.

L373-374: Why do you only filter time series based on the months April and May?

L390-391: You state that GLS performed better than OLS (but only for a small model fraction). But this would already justify the application of GLS – or not?

L540: Trends were not discussed yet → I would rephrase the beginning of this sentence

L547: The term "station" might be ambiguous here. It refers to snow stations, right? And not meteorological stations that measure e.g. air temperature and precipitation.

L568-571: I'm still struggling with this sentence. Also because trends have not been discussed.

L701-703: Do you have an idea what could cause this difference?

L704-705: I'm confused by this sentence. Do you mean that in terms of absolute values, your study and the one from Bach et al. (2018) do not agree?

L718: I would replace "spatial and elevation" by "horizontal and elevation"

L727: I would be careful with the statement "their significance is limited for hydrological applications" because higher elevations typically store more snow than lower elevations (hence the 0.7% can be misleading)

L794: Are depth of snowfall (HN) not used at all in this study? Somehow, I had the feeling they were used for quality control of the data. Or were they only processed for the harmonized data set?

L892: I guess some horizontal distance was used to select appropriate stations, right?

L965: What does "halving distance 250 m" mean?

L994: What does "non-zero true values" mean?

L1000: This sentence is oddly stated: How can the gap filling be unbiased with an overall non-zero daily bias?

**Stylistic comments and typos**

L228: "merged to" → "merged with"?

L235: "used for the" → "used from the"

L283: "some sources ended some date in between" → "some sourced ended in between"?

L360: "made sense" → "are meaningful"

L491-493: This sentence is still a bit hard to read. Maybe one could write: "and see if an additional third cluster would emerge (as in the north)…"

L770: "gridded area" → "gridded product"

L786: remove "one of those" or write "SCD and maxHS are amongst indicators least affected"

L976: "Visualization of some steps of the gap filling algorithm."

L1226-1227: change to "which includes scripts for the following tasks: reading in different data sources, performing all data pre-processing, quality checking, gap filling and statistical analysis."

**Figures and tables**

Figure B1: The figure caption is maybe missing

Table B2 → caption: correct "Empty cells no stations with…"

---

## Author Response (AR2)

**Cover letter**

Dear Guillaume Chambon,

We want to thank the reviewers for carefully checking the revised manuscript. We addressed all comments, which helped to further improve the manuscript and remove inconsistencies.

Below you can find our answers to the reviewers comments.

Looking forward to your evaluation,

Best,

Michael Matiu

**Referee 1: Ross Brown**

Many thanks to the authors for their diligent and comprehensive responses to my and the other reviewers' comments. The manuscript is significantly improved and I recommend the paper for publication after the authors have attended to the following minor comments:

Thank you for the positive feedback and for carefully checking the revised manuscript.

1. The regression analysis contains a couple of loose ends, namely the use of different methods (OLR and GLR) for different variables, and the use of bounded definitions for SCD that the authors indicate has some influence on trend results at low and high elevations. The use of a standard non-parametric trend analysis method such as Mann-Kendall would solve the first issue, while less restrictive definitions of SCD would solve the latter e.g. SCD computed over entire snow year and separately for the first and second halves of the snow year.

Thank you for pointing out these aspects. Concerning the SCD issue, indeed we already included during the first revision of the manuscript separate calculations for three different periods: first half was November to February, second half was March to May, and the full period November to May. However, we noted that SCD splitting does not solve the bounded nature of the metric. In the case of the higher elevation sites, the issue could be solved by increasing the length of the snow season to the full year (as we only used November to May), thus capturing the start and end of the snow cover. However, not all station records covered the entire year.

Instead, the solution we identified is to include in the analysis a probability distribution that is suitable for count data and that accounts for overdispersion, such as Negative Binomial. This is also able to handle the many zeros of SCD at lower elevation. We then compared the results of the Negative Binomial model to the other trend models (see below).

Since you mentioned the non-parametric MK test (with Theil-Sen slope estimates, we presume), we systematically evaluated the influence of the trend model on the results. We compared the estimates of GLS to OLS, to non-parametric MK with Theil-Sen, and for the SCD variables also to a Negative Binomial model. Both trend magnitudes and significance values do not differ substantially by choice of trend model. Trend magnitudes are highly correlated (on average > 0.95), and the trend significance agrees on average in 88% of the cases. We added details of all comparisons in the supplementary material as well as a short summary of these new results in the manuscript at Section 2.5 (methods: trend analysis).

Given these results, we decided to consistently use the GLS model throughout the manuscript and remove OLS.

2. Figure 8 is a useful addition, but did you consider computing regional series as standardized anomalies? This would highlight common responses and may allow the construction of a pan-regional series for each elevation range.

We created the suggested figure and provided it in the appendix. It seems that the SCD anomalies are highly correlated within regions, so a pan-regional series might work there. However, for the snow depth indices, this is not the case. Consequently, we were reluctant to create pan-regional series at this stage.

But as you implied, the anomalies highlight the common responses (or deviations) better, so we think it's a useful addition. We also added a sentence on this in the conclusion.

3. I flagged two places where I think some improvements could be made in the wording:

Section 3.7: the material on the homogenization of the series would fit better with the discussion of data homogeneity starting in line 180.

We moved this paragraph as suggested.

Line 523-524: Suggest removing the sentence "Such low R2 values…" as it does not add anything useful.

Done.

4. The fact that trend significance increases with elevation and is stronger in the spring period are important points that you should consider highlighting in the conclusions and abstract, as they are consistent with enhanced albedo feedbacks in mountain regions. I'm currently unable to access my collection of snow literature, but I recall there being several papers published on elevation-dependent albedo feedbacks that you can cite as part of this discussion.

Thank you for this suggestion. We highlighted this aspect in the abstract and conclusion, also citing relevant literature.

**Referee 2: Anonymous**

I would like to thank the authors for the additional effort they put in the revision of the manuscript. Readability and completeness were much improved. I particularly like section A.3 which is now very detailed and comprehensible. I have only some minor additional comments (the page and line number refer to the latest manuscript version with track changes).

We want to thank you for carefully evaluating the revised manuscript and for your valuable comments, which helped to identify many inconsistencies.

**Point comments**

L55: I find "after accounting for elevation" a bit odd. Do you mean trends of the same elevation bin differ amongst regions?

Exactly. We modified this to "at the same elevation".

L145-146: I don't understand this sentence. Why is synthesizing studies into a common Alpine view relevant for providing snow cover information at the regional scale?

Sorry for the confusion. Regional is somewhat ambiguous, and we meant the mountain range or larger extents, so we modified the sentence accordingly.

L331-332: I'm not sure if I understand this correctly: you tested the clustering with 2 – 8 PCs, right? It's a bit confusing because a couple of lines above, you write that you only retain the first five PCs.

Thank you for spotting this inconsistency. We forgot to remove the sentence in the previous lines during the revision.

L373-374: Why do you only filter time series based on the months April and May?

We only removed the respective months and not the other months. We made this clearer in the manuscript.

L390-391: You state that GLS performed better than OLS (but only for a small model fraction). But this would already justify the application of GLS – or not?

We followed a parsimonious approach, where we tried to keep the simpler models. However, also the other referee mentioned this inconsistency. We systematically evaluated differences between various trend models, and since they were marginal, we opted for GLS model for all analyses. We added explanations and results of the trend model comparison in the methods and supplement.

L540: Trends were not discussed yet à I would rephrase the beginning of this sentence

True, an artifact of the manuscript restructuring. We changed the linking sentence to match the previous section.

L547: The term "station" might be ambiguous here. It refers to snow stations, right? And not meteorological

stations that measure e.g. air temperature and precipitation.

True. We changed "station" to "site" in this section, when it refers to temperature and precipitation.

L568-571: I'm still struggling with this sentence. Also because trends have not been discussed.

Yes, this does not make sense at this point. We removed the part from this section, and added instead a sentence in Sec. 3.7 (Outlook), where we already have discussed detection and attribution.

L701-703: Do you have an idea what could cause this difference?

We tried to reproduce the results from Klein et al 2016, and found the following: For maxHS, the differences are caused by the different periods (1970-2015 vs 1971-2019). For the SCD variable, this is also because of the different period, and in addition also because of the different season definition. We used November to May, while Klein et al looked at the whole year (this affects especially the higher elevation sites). We added some sentences on this also in the manuscript, since this gives some evidence to our introductory statement why the synthesis is challenging.

L704-705: I'm confused by this sentence. Do you mean that in terms of absolute values, your study and the one

from Bach et al. (2018) do not agree?

Exactly. We tried to make this clearer.

L718: I would replace "spatial and elevation" by "horizontal and elevation"

Done.

L727: I would be careful with the statement "their significance is limited for hydrological applications" because

higher elevations typically store more snow than lower elevations (hence the 0.7% can be misleading)

True. We modified the statement to: "While the elevations above 3000 m only cover a minimal area (0.7% of the area studied here, see Fig. 2(c)), they store large amounts of snow: Figure 6(a) gives an indication of the expected increase in HS with elevation. Long-term monitoring is extremely challenging at elevations above 3000 m, and the snow cover at these elevations is relevant for hydrology, mountain ecosystems, glacier dynamics and mountain (ski) tourism. "

L794: Are depth of snowfall (HN) not used at all in this study? Somehow, I had the feeling they were used for

quality control of the data. Or were they only processed for the harmonized data set?

Thanks for spotting this. Yes, we used HN to same extent in the quality checking, but have not analysed it all otherwise. We made this clearer.

L892: I guess some horizontal distance was used to select appropriate stations, right?

For this step of the quality check, no. Given the climatological nature of the screening and the stronger dependency on elevation, we found horizontal distance here less important. We tried to make this clearer in the manuscript.

L965: What does "halving distance 250 m" mean?

The weights (for the average) are based on an exponential decay, where the weights are halved every 250m. This is a more interpretable transformation of the decay parameter (lambda). We added more clarification in the manuscript.

L994: What does "non-zero true values" mean?

This was a bad choice of formulation. We meant the non-zero held-out values. This has been changed in the manuscript.

L1000: This sentence is oddly stated: How can the gap filling be unbiased with an overall non-zero daily bias?

We changed "is unbiased" to "has extremely little bias".

**Stylistic comments and typos**

L228: "merged to" à "merged with"?

Done.

L235: "used for the" à "used from the"

Done.

L283: "some sources ended some date in between" à "some sourced ended in between"?

Done.

L360: "made sense" à "are meaningful"

Done.

L491-493: This sentence is still a bit hard to read. Maybe one could write: "and see if an additional third cluster would emerge (as in the north)…"

Done.

L770: "gridded area" à "gridded product"

Done.

L786: remove "one of those" or write "SCD and maxHS are amongst indicators least affected"

Done the latter.

L976: "Visualization of some steps of the gap filling algorithm."

Done.

L1226-1227: change to "which includes scripts for the following tasks: reading in different data sources, performing all data pre-processing, quality checking, gap filling and statistical analysis."

Done.

**Figures and tables**

Figure B1: The figure caption is maybe missing

The version with track changes looks so, but in the clean version it is there.

Table B2 à caption: correct "Empty cells no stations with…"

Done.